# INSTRUCT-SKILLMIX: A POWERFUL PIPELINE FOR LLM INSTRUCTION TUNING

**Simran Kaur**[1*]**, Simon Park**[1*]**, Anirudh Goyal**[2]**, Sanjeev Arora**[1]
[1] Princeton Language and Intelligence (PLI), Princeton University
[2] Meta

## ABSTRACT

We introduce INSTRUCT-SKILLMIX[1], an automated approach for creating diverse, high quality SFT data for instruction-following. The pipeline involves two stages, each leveraging an existing powerful LLM: (1) *Skill extraction*: uses the LLM to extract core "skills" for instruction-following by directly prompting the model. This is inspired by "LLM metacognition" of Didolkar et al. (2024); (2) *Data generation:* uses the powerful LLM to generate (instruction, response) data that exhibit a randomly chosen pair of these skills. Here, the use of random skill combinations promotes diversity and difficulty. The estimated cost of creating the dataset is under $600.

Vanilla SFT (i.e., no PPO, DPO, or RL methods) on data generated from INSTRUCT-SKILLMIX leads to strong gains on instruction following benchmarks such as AlpacaEval 2.0, MT-Bench, and WildBench. With just 4K examples, LLaMA-3-8B-Base achieves 42.76% length-controlled win rate on AlpacaEval 2.0, a level similar to frontier models like Claude 3 Opus and LLaMA-3.1-405B-Instruct.

Ablation studies also suggest plausible reasons for why creating open instruction-tuning datasets via naive crowd-sourcing has proved difficult. In our dataset, adding 20% low quality answers ("shirkers") causes a noticeable degradation in performance.

The INSTRUCT-SKILLMIX pipeline seems flexible and adaptable to other settings.

## 1 INTRODUCTION

*Instruction tuning* (sometimes also called *imitation learning*) is the first step in converting a base LLM trained on next-word prediction into a helpful and interactive agent. Whereas early versions of instruction tuning involved supervised fine-tuning (SFT) on traditional NLP question-answer datasets (Wei et al., 2022), nowadays, the SFT data is collected at high cost from skilled human annotators. We will use the term "instruction tuning" to refer solely to supervised fine-tuning (SFT) on such Q&A pairs — and not to reinforcement-learning methods such as PPO/DPO/RLHF (Schulman et al., 2017; Rafailov et al., 2023) etc., which usually follow SFT in the pipeline.

Human-generated data is expensive (e.g., even the tiny model Instruct-GPT was estimated to require $20K$ human hours OpenAI (2022)), which has motivated the creation of open-domain alternatives. ShareGPT (Chiang et al., 2023) contains conversations collected from a model-hosting website, whereas OpenAssistant (Köpf et al., 2023) and Dolly (Conover et al., 2023) contain crowd-sourced human data. Another intriguing method, popularized by Self-Instruct (Wang et al., 2023b) (and its variants, e.g., Alpaca (Taori et al., 2023)) leverages synthetic datasets. Here, a strong LLM is prompted using a small set of human-created examples to generate a large number of (query, answer) examples on a variety of topics.

Open evaluations of instruction-following ability have also sprung up. The popular AlpacaEval 2.0 (Dubois et al., 2023; 2024) is based upon curated queries from various sources. In such evaluations, a model's response to a provided query is compared against a strong reference model's response, and the model is ranked based upon its *win rate* — the percentage of queries for which the model

---

[*]Equal contribution.
[1]Source code can be found at `https://github.com/princeton-pli/Instruct-SkillMix`.

produces a better answer than the reference model, as judged by a powerful LLM. Rankings on AlpacaEval and related benchmarks like WildBench broadly align with the human rankings of a model's performance (Dubois et al., 2024; Lin et al., 2024).

## 1.1 SURPRISING DIFFICULTY OF INSTRUCTION TUNING

A persistent puzzle in this field is that SFT on the above public datasets does *not* yield good performance on the evaluations. It was initially suspected this is due to a lack of diversity in the training data. But, efforts to produce more diverse synthetic data — e.g., UltraChat (Ding et al., 2023), a synthetic dataset of 1.5M multi-turn conversations created via meticulously tracking lexical and topical diversity as well as coherence — did not significantly improve performance.

Another hypothesis places the blame on the uneven quality of open datasets — which are usually a hodge-podge of collected queries (e.g., Dolly (Conover et al., 2023)) — whereas proprietary datasets are produced to careful specifications using strict quality-control. One finding that supports this hypothesis is that SFT on the 1K Q&A pairs in Alpaca-52K with the longest responses, outperforms SFT on all 52K pairs (Zhao et al., 2024). In other words, the 51K other data-points are redundant, or even interfere with the "signal" present in the best 1K examples. This finding has inspired "less is more" approaches — including an extreme one based upon just a judicious set of in-context examples (Lin et al., 2023) to provide a surprisingly reasonable level of instruction tuning and alignment — but they did not significantly improve the performance either.

Some have cautioned against hopes for a miracle out of instruction tuning. Gudibande et al. (2023) suggest, based upon careful experiments, that basic capabilities of the LLM arise from pre-training and its massive training corpus. Most deficiencies left after pre-training will not be fixable by, say, a million SFT examples. While this perspective feels broadly correct, it does not quite explain why open efforts to instruction tune Mistral-7B-Base-v0.2 fail to match the performance of its proprietary *Instruct* counterpart, which has only undergone SFT.

The above difficulties have lately lowered interest level in instruction tuning, with many researchers now turning to RL-based methods (e.g., PPO, DPO), which have been used in recent open-source projects to greatly improve proprietary chat models (Meng et al., 2024), which had already trained on expensive human data.

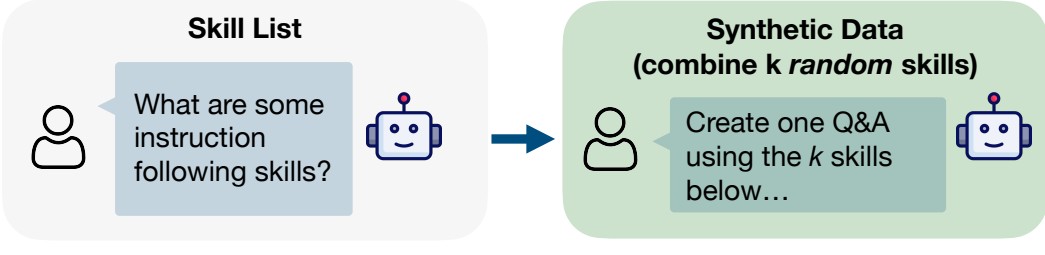

Step 1: Use powerful LLM to gather instruction-following skills.

Step 2: Create synthetic **(instruction, response)** pair that requires applying *k* provided skills.

Figure 1: **Sketch of INSTRUCT-SKILLMIX pipeline.** See Figures 2a and 2b for more details on two different implementations of INSTRUCT-SKILLMIX.

## 1.2 OUR CONTRIBUTIONS

We describe a more efficient and effective approach for creating synthetic instruction tuning datasets. Past open efforts invested significant human effort in ensuring *high coverage* of topics and scenarios to sufficiently equip the LLM for scenarios it might encounter at deployment time. We take a subtly different tack. Accepting that pre-training is the dominant source of the LLM's "inner knowledge," we focus on merely teaching the LLM to draw upon that inner knowledge and present it nicely during conversations.

The key idea is to use a strong LLM as a teacher. The recent discovery of *LLM Metacognition* (Didolkar et al., 2024) suggests that frontier models have significant capability to "think about thinking," which in humans is referred to as *metacognition* (Flavell, 1979). Specifically, it was shown that given a task dataset, frontier LLMs can help assemble a list of named "skills" needed to solve that task. This requires no human involvement apart from an automated interaction with an LLM [2].

The first phase ("*Skill Extraction*") of our pipeline INSTRUCT-SKILLMIX uses this idea and a frontier LLM to identify a list of "basic skills" needed for instruction-following. Unlike Didolkar et al. (2024), which extracts skills from existing SFT datasets, we instead identify skills by directly prompting a strong LLM. (We also tried extracting skills using examples from Alpaca and Ultrachat, and it works quite well, but noticeably worse than our main method.) See Section 2.1.

The second phase of our pipeline, *Data Generation*, uses the list of extracted skills to produce synthetic query-response examples. Here, we repeatedly draw a random pair of skills from the list and prompt the powerful LLM to produce a suitable query that tests those two skills, and to also produce a good response to the query. This generation is inspired by the SKILLMIX evaluation (Yu et al., 2024) for LLMs' compositional generalization, which also uses a predetermined list of skills. Hence we call our method INSTRUCT-SKILLMIX. See Section 2.2

Using merely 2K to 4K such Q&A examples, vanilla SFT allows popular small base models (Mistral-7B-Base-v0.2, LLaMA-3-8B-Base, and Gemma-2-9B-Base) to match or surpass some apex models on AlpacaEval 2.0, such as the original GPT-4, LLaMA-3.1-405B-Instruct and Claude 3 Opus (Table 1). The estimated cost of creating this 4K dataset using the GPT-4 API is under 600 US dollars.

We stress that although reminiscent of prior efforts using synthetic data such as UltraChat, our pipeline is fully automated with no human design elements (e.g., choice of topics, lexicon etc.). The only human involvement involves the short prompts used for skill extraction and question generation, which we adapted from the math setting of Didolkar et al. (2024). While our pipeline currently focuses on simple instruction-following, the method seems extensible in future to safety/alignment, as well as domain-specific Q&A.

## 2    INSTRUCT-SKILLMIX

This section describes our methodology for extracting skills from powerful LLMs [3] and how to use these extracted skills to create a diverse, high quality dataset. A simplified version of our pipeline and prompts are depicted in Figures 1 and  2. Section 3 reports the evaluation results when finetuning on this dataset.

### 2.1    SKILL EXTRACTION PROCEDURE

The method involves an automated interaction with a frontier LLM (GPT-4-Turbo). We ask the frontier LLM to first generate a list of topics that arise in instruction-following. For each topic returned by the LLM, we further prompt it to generate a list of skills that are needed to answer typical queries on that topic. Additionally, we ask the LLM to create a list of query types (e.g., "Information Seeking") that might arise in that topic. See Appendix L.4 for details about the prompts used, and Appendix K.2 for the list of all extracted skills. Since this method relies solely upon the LLM's inner meta-knowledge, this method should extend easily to other types of instruction-following.

**An Earlier Attempt:**    Our initial attempt to extract skills leveraged existing instruction tuning datasets, which is a more direct analog of the method in Didolkar et al. (2024). However, we suspected this to be sub-optimal due to known limitations of past instruction tuning datasets. We therefore designed the method described above, and found it superior. It also has scientific benefit of being independent of existing datasets like Alpaca and Ultrachat. However, the dataset from the initial method, called INSTRUCT-SKILLMIX-SEED-DATASET-DEPENDENT (INSTRUCT-SKILLMIX-D;

---

[2]Skill lists generated by different frontier models are related but not isomorphic. Skills generated by one model are comprehensible to other models. See Didolkar et al. (2024) for such experiments.

[3]We use GPT-4-Turbo for our main experiments (2024-04-09 checkpoint unless specified otherwise). See Appendix B for results when using Claude 3.5 Sonnet (2024-06-20).

Table 1: **Evaluation results of *base* models supervised-finetuned on INSTRUCT-SKILLMIX versus the proprietary *instruct* versions and other proprietary models.** For our models, we report the results for best checkpoint selected using held-out queries. For other models(*), we report the published numbers available on publicly available leaderboards. "**# Data**" refers to the number of (instruction, response) pairs in the training data. See Table 9 for a more detailed view, including comparisons to past open datasets.

| Model | # Data | AlpacaEval2.0 LC WR(%) | WildBench WB-Reward$_\infty^{\text{gpt4t}}$ |
|---|---|---|---|
| **LLaMA-3-8B** | | | |
| Ours | 4K | **42.76** | **-36.91** |
| *LLaMA-3-8B-Instruct | - | 22.90 | -46.30 |
| **Mistral-7B-v0.2** | | | |
| Ours | 4K | **36.70** | **-29.25** |
| SFT on Alpaca-52K | 52K | 8.64 | -80.47 |
| *Mistral-7B-Instruct-v0.2 | - | 17.10 | -54.70 |
| **Gemma-2-9B** | | | |
| Ours | 2K | 36.18 | -37.83 |
| Gemma-2-9B-Instruct | - | **37.21** | **-28.78** |
| **\*Other Proprietary Models** | | | |
| LLaMA-3.1-405B-Instruct | - | 39.30 | - |
| Mistral Large | - | 32.70 | -46.40 |
| Claude 3 Opus | - | 40.50 | -21.20 |
| Claude 3 Sonnet | - | 34.90 | -30.30 |
| GPT-4-Omni (2024-05-13) | - | **57.50** | **+1.70** |
| GPT-4 (2023-03-14) | - | 35.30 | - |

see Appendix A) is still very useful for ablations that pinpoint ways in which our skill-based pipeline improves on past synthetic datasets for instruction-following (see Tables 2, 3, 4, and 5).

## 2.2 DATA GENERATION

Inspired by the recent SKILLMIX evaluation (Yu et al., 2024), we generate instruction-following examples by randomly picking $k$ skills as well as a random query type. The frontier LLM is prompted to create Q&A pairs that illustrate these $k$ skills and the query type. We refer to the resulting dataset as INSTRUCT-SKILLMIX. For example, INSTRUCT-SKILLMIX(k=2)-1K refers to 1,000 examples of data created from random combinations of $k = 2$ skills. See Appendix L.3 and L.5 for the details about the prompts used for data generation.

See Appendix A for more details, and an estimate of the low cost of INSTRUCT-SKILLMIX pipeline.

**Where does diversity come from?** The first source of diversity is the skill labels. A skill label represents some part of the frontier LLM's meta-knowledge of human behavior and needs, which it observed in its vast training set or during instruction tuning. Replacing a concrete Q&A example with a skill label converts it into a pointer to a region in the frontier LLM's meta-knowledge, which the model can then freely draw upon to create new examples. The second source of diversity is the use of random $k$-tuples of skills when generating new examples. The motivation here is that, in most cases, distinct tuples will lead to very distinct flavor of examples.

For instance, the skill pair (`critical thinking and communication`, `literature and language skills`) leads to the following instruction

```
I'm a high school English teacher aiming to develop a curriculum unit for my 11th-grade
class, focusing on American literature.  I want this unit to go beyond just reading and
understanding the texts.  Specifically, I'm looking to enhance my students' critical
thinking and communication skills through engaging activities related to the literature.
Can you suggest detailed ways to incorporate these skills, ideally with concrete examples
and expected learning outcomes?
```

whereas the skill pair (`critical thinking and communication`, `skill in virtual and system design`) leads to

> ```
> As an IT manager, I am overseeing the development of a virtual workspace to enhance
> communication and efficiency among remote teams.  This workspace must support multimedia
> content, including video conferencing and live document editing.  What are the critical
> steps I should take in its design and implementation, balancing technical robustness
> with ease of use?  Could you provide specific technologies to consider and any potential
> obstacles?
> ```

Even though the two skill pairs share a common skill, they lead to rather distinct Q&A pairs, involving creative and nuanced situations with subtle moving parts. Since the number of $k$-tuples scales as $\binom{N}{k}$, where $N$ is the number of skills, using pairs of skills foster a lot of diversity — e.g., $125,000$ possibilities with $N = 500, k = 2$. The pipeline in our experiments mainly uses $k = 2$, but generating answers to these queries will certainly end up using many other unnamed skills as well, and thus serve as a rich source for learning how to follow instructions.

## 3 EXPERIMENTS

### 3.1 EXPERIMENTAL SETUP

**SFT on INSTRUCT-SKILLMIX(k).** We finetune LLaMA-3-8B-Base, Mistral-7B-Base-v0.2, Gemma-2-9B-Base, LLaMA-2-7B-Base, and LLaMA-2-13B-Base on a varying number of examples from INSTRUCT-SKILLMIX-D(k) and INSTRUCT-SKILLMIX(k). We train for multiple epochs and select the best checkpoint by performance on 100 held-out questions. Similar to Ouyang et al. (2022); Zhou et al. (2023), we observe that using cross-entropy loss on a validation set does not lead to the best checkpoint. See Appendix E.2 for a more detailed discussion of the checkpoint selection procedure. As a baseline, we also finetune on different subsets of Alpaca-52K, including the 1K or 5K examples with the longest completions. For further training details (e.g., hyperparameters), see Appendix E.1.

**Evaluation.** We evaluate our models on popular instruction following benchmarks: AlpacaEval 2.0 (Dubois et al., 2024), MT-Bench (Zheng et al., 2023), and WildBench (Lin et al., 2024). For AlpacaEval, we report the length-controlled win rate (LC WR) of the responses of our model against a reference response, which corrects for the length bias of the judge model. For MT-Bench, we report the average score of the responses of our model graded by a judge model. For WildBench, we report the WB-Reward (weighted win-rate) of the response of our model against one reference response when graded by a judge model. For further evaluation details, see Appendix D. See Table 11 in Appendix C for evaluations on additional benchmarks.

### 3.2 MAIN RESULTS

For the main results of the paper, we report the evaluation results when models are finetuned on INSTRUCT-SKILLMIX in Table 1, and summarize our findings below. For a more detailed version of Table 1, see Table 9. For additional ablations, see Appendix F. For evaluations on other LLM benchmarks, see Table 11.

**INSTRUCT-SKILLMIX achieves SOTA performance amongst SFT models.** LLaMA-3-8B-Base finetuned on 4K examples from INSTRUCT-SKILLMIX(k=2) yields LC win rate of 42.76% on AlpacaEval 2.0. This score is higher than Claude 3 Opus, LLaMA-3.1-405B-Instruct, and GPT-4 (2023-03-14). Mistral-7B-Base-v0.2 finetuned on the same data achieves -29.25 on WildBench, which outperforms Claude 3 Sonnet and Mistral Large. Gemma-2-9B-Base finetuned on 2K examples from INSTRUCT-SKILLMIX(k=2) gets a score of 8.12 on MT-Bench, which is better than GPT-3.5-Turbo (2023-03-01). To best of our knowledge, these scores are higher than any base model that has *only* undergone supervised instruction finetuning (i.e., no RLHF, DPO, PPO, or variants).

**Early saturation.** Performance from our method rises rapidly, reaching unprecedented levels with 1K examples. Unfortunately, improvements stop already with 4K examples. This turns out to be a consequence of its high efficiency at inducing good instruction-following. Specifically, with 4K

examples, the win-rate against GPT-4 approaches $50\%$ on *heldout* queries from our pipeline, and thus overfitting sets in.

**Observed limitations.** The open benchmarks used in this study have known limitations, related to the insufficient number of under-specified or ambiguous queries, and no testing of long-form generations such as multi-page essays. Our current pipeline shares some of these limitations. Fixing this seems very doable via suitable modification to our INSTRUCT-SKILLMIX pipeline, but this is left for future work. This aligns with the observation in Bai et al. (2024) that a model's effective generation length seems to be limited by the typical length of examples seen during SFT, and is exacerbated by the relative scarcity of long-form samples in the SFT data. This underscores the critical influence of training data composition on a model's post-fine-tuning capabilities, and would be interesting to investigate in future work.

## 4 ABLATION STUDY

Whereas pretraining is the source of an LLM's basic capabilities (Gudibande et al., 2023), the sole goal of instruction tuning is to impart skills, such as answer-structuring, empathy, helpfulness, etc.

Vanilla SFT on Q&A data generated by a teacher LLM is akin to *imitation learning.* Our ablation studies below help understand the contribution of different elements to the effectiveness of imitation learning method using INSTRUCT-SKILLMIX Q&A. The main finding is that the source of largest improvement is the skill extraction step.

### 4.1 BENEFITS OF SKILL EXTRACTION (WITH MIXING TURNED OFF)

To highlight the benefits of our skill-based method versus current synthetic approaches, we use the pioneering Alpaca dataset, whose responses are rewritten by GPT-4 (2023-03-14) (Peng et al., 2023). The fairest comparison here would be with our INSTRUCT-SKILLMIX-D(k=1) data, where the underlying skills were derived from a random sample of *Alpaca-52K*, and each of our datapoints uses one of those extracted skills. All results below involve finetuning Mistral-7B-Base-v0.2 on different subsets of the Alpaca-52K dataset: (1) *Alpaca-1K Longest*: 1,000 examples with the longest responses (Zhao et al., 2024); (2) *Alpaca-5K Longest*: 5,000 examples with the longest responses; (3) *Alpaca-5K Random*: 5,200 randomly sampled examples from which we extracted our skills; and (4) *Alpaca-52K*: the full 52,002 examples.

Table 2: **Evaluation results of Mistral-7B-Base-v0.2 finetuned on INSTRUCT-SKILLMIX-D vs. on Alpaca-52K.** Note that skills extracted from Alpaca-5K Random were used to create the INSTRUCT-SKILLMIX-D datasets.

| SFT Dataset | # Data | AlpacaEval 2.0 LC WR(%) | MT-Bench | WildBench WB-Reward$_\infty^{\text{gpt4t}}$ |
|---|---|---|---|---|
| INSTRUCT-SKILLMIX-D(k=2) | 4K | **29.77** | 7.17 | **-39.06** |
| INSTRUCT-SKILLMIX-D(k=1) | 1K | 27.04 | **7.22** | -46.83 |
| Alpaca-1K Longest | 1K | 10.09 | 6.88 | -63.38 |
| Alpaca-5K Longest | 5K | 8.92 | 6.90 | -62.55 |
| Alpaca-5K Random | 5K | 11.10 | 6.86 | -74.41 |
| Alpaca-52K Full | 52K | 8.64 | 6.45 | -80.47 |

As shown in Table 2, finetuning on 1,000 examples with the longest completions from Alpaca-52K yields 10.09% LC win rate on AlpacaEval 2.0. On the other hand, finetuning on only 1K examples of INSTRUCT-SKILLMIX-D(k=1) yields 27.04% LC win rate. Note that since the skills in INSTRUCT-SKILLMIX-D are mostly derived from Alpaca-52K, the observed improvements in the win rate are indicative of the improved quality of INSTRUCT-SKILLMIX-D queries.

### 4.2 MIXING SKILLS HELPS, BUT NOT AS MUCH AS SKILL EXTRACTION

In Table 3, models finetuned on INSTRUCT-SKILLMIX-D(k=2) data marginally outperform models SFT on INSTRUCT-SKILLMIX-D(k=1) on AlpacaEval and WildBench, whereas performance on

MT-bench is about the same. The marginal improvements from increasing $k$ are less noticeable for INSTRUCT-SKILLMIX.

Table 3: **Evaluation results of Mistral-7B-Base-v0.2 SFT on INSTRUCT-SKILLMIX where k=1 vs. k=2.** In each entry, we report INSTRUCT-SKILLMIX-D/INSTRUCT-SKILLMIX

| Model | # Data | AlpacaEval 2.0 | | MT-Bench | WildBench |
| | | WR(%) | LC WR(%) | | WB-Reward$_\infty^{\text{gpt4t}}$ |
|---|---|---|---|---|---|
| **SFT on INSTRUCT-SKILLMIX(k=2)** | | | | | |
| Mistral-7B-Base-v0.2 | 1K | 33.87/42.48 | 27.48/38.34 | 6.92/7.33 | -41.46/-30.65 |
| | 2K | 37.05/40.83 | 31.57/36.18 | 7.04/7.20 | -43.46/-31.92 |
| | 4K | 35.08/40.74 | 29.77/36.70 | 7.17/7.16 | -39.06/**-29.25** |
| **SFT on INSTRUCT-SKILLMIX(k=1)** | | | | | |
| Mistral-7B-Base-v0.2 | 1K | 30.06/41.75 | 27.04/38.34 | 7.22/7.49 | -46.83/-30.95 |
| | 2K | 35.07/- | 31.66/- | 7.39/- | -46.97/- |
| | 4K | 33.57/- | 28.85/- | 7.13/- | -44.43/- |

## 4.3 QUALITY OF QUERIES (AND SKILLS) MATTERS

The effectiveness of this approach depends on the quality of the queries used in the fine-tuning process, where high-quality queries enable the frontier LLM teacher to provide richer instruction to the student model undergoing instruction tuning. This relationship between the quality of queries and the skills being imparted is supported by two key observations. First, the frontier LLM proves to be a more effective teacher when the skill list being used was also entirely generated using its help (as opposed to giving it skills derived from existing datasets).[4] Across all model types, dataset size, and the evaluation benchmark, we generally see an improvement when finetuning on INSTRUCT-SKILLMIX compared to INSTRUCT-SKILLMIX-D (see Table 9 for more details). Second, incorporating these sub-optimal skills from existing datasets as a part of "teaching" (e.g., with INSTRUCT-SKILLMIX-D) is still more effective than using an equal number of random (or even the longest) examples from Alpaca-52K when responses are also by the same frontier LLM. These findings suggest that the quality of the queries (and the skills used to create those queries) drives how well data generated by the frontier LLM is able to impart its skills on the model undergoing instruction tuning.

## 4.4 EFFECT OF TEACHER AND GRADER

SFT performance derives from the model used to generate Q&A data, which plays the *teacher* role in imitation learning. The student's performance is evaluated by the grader model. The main results reported in this paper used GPT-4-Turbo as the teacher, and some checkpoint of GPT-4 or GPT-4-Turbo as the grader.

**Effect of the teacher** Many SFT efforts in 2023 used earlier versions of GPT-4 or GPT-3.5, which were weaker than GPT-4-Turbo. To pin-point the effect of this change, we try doing a head-to-head comparison once we fix the teacher. The responses in Alpaca-1K Longest are written by GPT-4 (2023-03-14), whereas INSTRUCT-SKILLMIX data is generated by GPT-4-Turbo. Thus, we use GPT-4-Turbo to regenerate answers to Alpaca-1K Longest (Zhao et al., 2024), and we also use GPT-4 (2023-03-14) to regenerate INSTRUCT-SKILLMIX-D.

Table 4 compares the performance of Mistral-7B-Base-v0.2 when finetuned on the two datasets using the two versions of GPT-4. For each fixed data generator model, the INSTRUCT-SKILLMIX dataset leads to a better performance. Furthermore, replacing GPT-4 with the stronger GPT-4-Turbo in data generation makes INSTRUCT-SKILLMIX pull even further ahead of Alpaca-1K Longest, which highlights that our pipeline is better positioned than Alpaca dataset to elicit better supervision from a more powerful LLM teacher.

---

[4]We also observed improved performance when the teacher model generated data based on its own set of skills, rather than using skills extracted by a different teacher model, further highlighting the advantages of leveraging the teacher model's metacognitive capabilities during dataset creation (see Appendix B.3).

Table 4: **Evaluation results of Mistral-7B-Base-v0.2 finetuned on INSTRUCT-SKILLMIX-D vs. Alpaca-1K Longest generated from two different versions of GPT-4.** For a fixed data generator model, SFT Mistral-7B-Base-v0.2 on INSTRUCT-SKILLMIX-D outperforms SFT on Alpaca-1K Longest.

| Model for Data Generation | Dataset | AlpacaEval 2.0 | | MT-Bench |
|---|---|---|---|---|
| | | WR(%) | LC WR(%) | |
| GPT-4 (2023-03-14) | Alpaca-1K Longest | 12.75 | 10.09 | 6.83 |
| | INSTRUCT-SKILLMIX-D-1K | 13.29 | 15.01 | 7.10 |
| GPT-4-Turbo (2024-04-09) | Alpaca-1K Longest | 35.23 | 19.62 | 6.99 |
| | INSTRUCT-SKILLMIX-D-1K | 33.87 | 27.48 | 6.92 |

**Effect of choice of grader** We use GPT-4-Turbo to generate data and AlpacaEval 2.0 uses GPT-4 for grading, creating a scenario where both the teacher model and grader model are from the same family. This raises the question of whether model family overlap leads to a potential grading bias and inflated scores. To quantify this effect, we used Claude 3 Opus as the grader for AlpacaEval 2.0. Table 5 shows that although Claude is a more generous grader across the board, it generally preserves the relative rankings among the models. Importantly, it exhibits even stronger preference for our student models' generations than does GPT-4.

Table 5: **Evaluation results when using two different graders for AlpacaEval 2.0.** Relative ranking of evaluated models are generally preserved when using different graders. Here, ISM-D refers to INSTRUCT-SKILLMIX-D.

| Model | Grader: GPT-4 (2023-11-06) | | Grader: Claude 3 Opus | |
|---|---|---|---|---|
| | WR(%) | LC WR(%) | WR(%) | LC WR(%) |
| Mistral-7B-Base-v0.2 SFT on ISM-D-1K | 33.87 | 27.48 | 50.56 | 38.50 |
| Mistral-7B-Base-v0.2 SFT on ISM-D-2K | 37.05 | 31.57 | 48.94 | 38.29 |
| Mistral-7B-Base-v0.2 SFT on ISM-D-4K | 35.08 | 29.77 | 52.55 | 44.16 |
| (Reference Model) LLaMA-3-70B-Instruct | 33.20 | 34.40 | 39.68 | 42.33 |
| (Reference Model) Mistral-7B-Instruct-v0.2 | 14.70 | 17.10 | 15.16 | 18.89 |
| (Reference Model) LLaMA-2-70B-Chat | 13.90 | 14.70 | 16.67 | 17.85 |

## 5 EFFECT OF LOW QUALITY DATA

Our fully synthetic pipeline produces a large number of high-quality questions and answers that look impressive but also (for want of a better word) "robotic." Data sourced from human workers shows greater variation, and one begins to wonder if that additional diversity could be beneficial. We tried interventions such as generating $20\%$ using a different prompt — e.g., require a shorter answer, or a poor quality answer. In a human pipeline, this variation would be expected. We can think of this as "data from shirkers," and one would expect a fair bit of it in naive crowdsourcing. (In corporate settings it would be mitigated via quality control measures.) See Appendix I for an example of a poor quality response.

We replace $20\%$ of the responses in INSTRUCT-SKILLMIX(k=2)-2K with short responses ("respond in one paragraph") to create BREV-INSTRUCT-SKILLMIX(k=2)-2K. Finetuning Mistral-7B-Base-v0.2 on BREV-INSTRUCT-SKILLMIX-D was surprising: brevity constraint on just $20\%$ of data almost halved the average response length on AlpacaEval, from 2817 to 1746 characters. LC win rate dropped from $31.57\%$ to $23.93\%$.

We alternatively replace $20\%$ of the responses in the same datasets with responses that are still long but have poor quality (i.e., deliberately sloppy and unhelpful) to create JUNK-INSTRUCT-SKILLMIX(k=2)-2K. Mistral-7B-Base-v0.2 finetuned on the JUNK-INSTRUCT-SKILLMIX-D yields less than $1\%$ win rate on AlpacaEval and 5.01 on MT-Bench.

**Lower-quality data harms performance.** As shown in Table 6, replacing just $20\%$ of the data with poor quality responses harms performance. For INSTRUCT-SKILLMIX-D, the harm is super-proportionate. These observation may help explain why creating open-domain instruction tuning data has proved so difficult via naive crowd-sourcing.

Table 6: **Evaluation results of models finetuned on low quality INSTRUCT-SKILLMIX.** Replacing just 20% of the dataset with low quality data has a super-proportionate harm on the model performance. Amount of harm greatly differs between the two versions of the pipeline.

| Model | # Data | AlpacaEval 2.0 | | MT-Bench | WildBench |
| | | LC WR(%) | Avg Len | | WB-Reward$_{\infty}^{\text{gpt4t}}$ |
| --- | --- | --- | --- | --- | --- |
| **SFT on INSTRUCT-SKILLMIX-D(k=2)** | | | | | |
| | 2K | 31.57 | 2817 | 7.04 | -43.46 |
| Mistral-7B-Base-v0.2 | 2K (Brevity 20%) | 23.93 | 1746 | 6.69 | -49.85 |
| | 2K (Junk 20%) | 0.77 | 1104 | 5.01 | -47.50 |
| **SFT on INSTRUCT-SKILLMIX(k=2)** | | | | | |
| | 2K | 36.18 | 2936 | 7.20 | -31.92 |
| Mistral-7B-Base-v0.2 | 2K (Brevity 20%) | 31.61 | 2336 | 7.32 | -32.27 |
| | 2K (Junk 20%) | 24.60 | 2435 | 6.90 | -47.50 |

**High-quality data's protective effect.** While adding some low-quality data to INSTRUCT-SKILLMIX already causes a noticeable performance drop, doing the same to INSTRUCT-SKILLMIX-D is catastrophic. This suggests that INSTRUCT-SKILLMIX is more robust to "shirkers," corroborating our previous observations in Table 9 of the superior performance of INSTRUCT-SKILLMIX over INSTRUCT-SKILLMIX-D. This finding suggests that higher quality data can somewhat protect against negative effects of "shirkers," which needs further study.

## 6 RELATED WORK

Prior works observe improvements from instruction finetuning on *fewer*, but *higher quality* data generated by humans (Zhou et al., 2023; Touvron et al., 2023). However, efforts to curate high quality data from humans are quite expensive, and licensing can become complicated. This has led to an increase in the popularity of semi-automated and less expensive approaches.

**Selecting high quality data.** Synthetic data creation has become a predominant approach for curating instruction tuning datasets, especially in the academic realm (Wang et al., 2023b; Dubois et al., 2023; Xu et al., 2024; Gunasekar et al., 2023). These synthetic datasets are generally created by providing in-context examples to a powerful LLM to produce the synthetic data, followed by some post-filtering (Wang et al., 2023b). Recent efforts have also focused on data selection strategies for high quality subsets of the original dataset, which lead to performance gains (Tunstall et al., 2023; Chen et al., 2024; Liu et al., 2024; Zhao et al., 2024). Notably, Zhao et al. (2024) show that finetuning on just the 1K longest completions from Alpaca-52K outperforms finetuning on the entire Alpaca-52K dataset. Whereas the data selection methods just described focus on *general-purpose* instruction tuning, Xia et al. (2024) explore an optimizer-aware data selection strategy for *targeted* instruction tuning.

**Encouraging data diversity.** Common approaches to elicit diversity in datasets include mixing multiple datasets (Wang et al., 2022; Longpre et al., 2023; Wang et al., 2023a), as well as rewriting the data in multiple ways and changing formatting (Allen-Zhu and Li, 2024; Honovich et al., 2023). The Self-Instruct framework (Wang et al., 2023b) and variants such as Alpaca-52K (Dubois et al., 2023) encourage diversity by identifying similar pairs using ROUGE-L similarity. Other approaches to ensure diversity impose constraints on the topic in order to enhance wide coverage (Ding et al., 2023; Xu et al., 2024), or require synthetic data to use a random subset of words or concepts chosen from some vocabulary (Eldan and Li, 2023; Gunasekar et al., 2023; Li et al., 2024). The latter approach is also suggested by recent work that provides a mathematical model for emergence via LLM scaling (Arora and Goyal, 2023) and used in the evaluation setting in Yu et al. (2024).

**AlpacaEval.** AlpacaEval (Li et al., 2023; Dubois et al., 2024) is a popular evaluation for assessing instruction-following capabilities of LLMs. The tested model provides answers 805 carefully curated instructions, and its answers are compared against reference outputs of a designated baseline model. For each instruction, another evaluator LLM outputs a preference between the two responses (output

of the model being evaluated vs. reference output by the baseline mode). The primary evaluation metric is the *win rate*, which represents the expected probability that the grader model favors the response generated by the evaluated model over the response produced by the baseline model. Given that a raw win rate shows bias towards longer responses, AlpacaEval 2.0 (Dubois et al., 2024) introduces the *length-corrected (LC) win rate* as a proxy for what the raw win rate would be if the evaluated model's response lengths and baseline model's response lengths matched.

**WildBench.** WildBench (Lin et al., 2024) is another benchmark for assessing the instruction following capabilities of LLMs. Unlike the AlpacaEval instructions, 50% of which are only "information seeking" type questions, the instructions for WildBench cover a more diverse distribution of task categories, including coding and creative writing. Whereas the grading in AlpacaEval is more liberal (since there is no penalty for poor responses), the grading in WildBench is more finegrained: a model answer is compared against a reference answer, but is graded on a scale of (1) win by a big margin, (2) win by a small margin, (3) tie, (4) lose by a small margin, and (5) lose by a big margin. This ensures that models that output bad answers to certain types of questions are penalized.

**RL-inspired approaches.** Since we do not use RL, we defer discussion of these approaches to Appendix G.

## 7 CONCLUSION

While one would have certainly expected the cost factor as well as scaling ability to ultimately favor synthetic data, the surprising finding in this paper is that, when done well, synthetic data can be much more *effective* than human data for instruction tuning. Our INSTRUCT-SKILLMIX pipeline, uses the recent discovery of LLM Metacognition (Didolkar et al., 2024) to extract skills using a powerful LLM and then leverages an LLM to create quality instruction data using random pairs of those skills.

Vanilla SFT of base models on just 1K to 4K examples from our pipeline outperforms the proprietary *instruct* versions of the same model, as well as older and larger instruction tuning efforts like Vicuna and Ultrachat that used orders of magnitude more datapoints. The performance also approaches those of frontier models, which trained on expensive human data as well as with RL techniques. Unfortunately, our method saturates at 4K examples, when win-rate on heldout queries approaches 50%.

Ablation studies in Section 4.4 rule out potential confounding factors, such as the use of a strong teacher, or bias due to teacher and grader belonging to the same family. These ablations reinforce that the improvement is primarily due to the uniformly high quality of examples produced by our skill-based pipeline. Each example contains a query with nontrivial scenarios and lots of moving parts, which improve imitation learning.

Section 5 offers a preliminary exploration of pitfalls of naive collection of instruction tuning data. In particular, the presence of some lower quality data noticeably harms the model's performance. This insight should be more rigorously investigated, including via new theory. The experiment also suggests that our less preferred INSTRUCT-SKILLMIX-D method (which involves extracting skills from an existing dataset) is more susceptible to such bad data than our preferred INSTRUCT-SKILLMIX.

One potential benefit of INSTRUCT-SKILLMIX-D may be that it gives some insights into an efficient method for dataset distillation (Wang et al., 2020) for text datasets, which has not yet proved possible.

Finally, it should be noted that our results look stronger on paper than they actually are. Open evaluations such as AlpacaEval 2.0 have blind spots, especially the fact that win rate of even 50% against a frontier model still allows unacceptably high frequency of unsuitable responses in a deployment setting. The new WildBench evaluation does test for more corner cases. We hope that INSTRUCT-SKILLMIX ideas can also leverage LLM metacognition to create a better evaluation.

Although our SFT data does not address safety and alignment, our skill-based ideas may be useful there. A related next step would be to leverage our ideas of skill extraction to improve RL-based methods (whether for instruction-following or alignment). We hope to address these in future work.

## 8 REPRODUCIBILITY STATEMENT

We provide the full lists of extracted skills, topics, and query types in Appendix K. We provide the set of prompts used to generate the data from these lists in Appendix L.3 and L.3. We provide the set of training hyperparameters in Appendix E.1. We discuss the details of the checkpoint selection method in Appendix E.2. We provide the details of evaluation settings in Appendix D.

## 9 ACKNOWLEDGEMENTS

SK, SP, and SA acknowledge funding from NSF, DARPA, ONR, and OpenAI.

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

# List of Appendices

## A    INSTRUCT-SKILLMIX PIPELINE (MORE DETAILS)

### A.1    INSTRUCT-SKILLMIX-D AND INSTRUCT-SKILLMIX PIPELINES

**Method 1: Leveraging existing instruction datasets.**    Even though existing instruction-following datasets may not induce good chat capability via vanilla SFT, these datasets still exhibit (possibly in an uneven fashion) some "skills" needed by the model. Thus, we adapt the methodology presented in Didolkar et al. (2024) and use GPT-4-Turbo to extract instruction-following skills from random samples of existing instruction and alignment datasets (5,200 samples from Alpaca-52K and 1,000 samples from UltraChat). We then use GPT-4-Turbo to cluster similar skills into broader categories, forming our final list of instruction-following skills. See Appendix K.1 for the list of all extracted skills and Appendix L.1 and L.2 for details about the prompts used for skill extraction.

**Method 2: Directly prompting a powerful LLM.**    While Method 1 works surprisingly well, it generated unease about possibly relying on existing seed datasets of uneven quality, and thus potentially inheriting their limitations and biases. Therefore we also tried an alternative pipeline that solely relies on the powerful LLM's ideas about list of skills it leverages for instruction-tuning.

We will refer to the datasets generated from the seed-dataset dependent and the seed-dataset agnostic versions as INSTRUCT-SKILLMIX-SEED-DATASET-DEPENDENT and INSTRUCT-SKILLMIX-SEED-DATASET-AGNOSTIC, respectively. Unless stated otherwise, INSTRUCT-SKILLMIX refers to the INSTRUCT-SKILLMIX-SEED-DATASET-AGNOSTIC data.

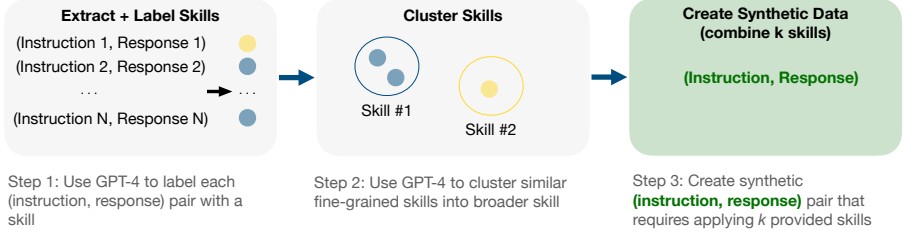

(a) Top row: INSTRUCT-SKILLMIX-D pipeline (short for INSTRUCT-SKILLMIX-SEED-DATASET-DEPENDENT).

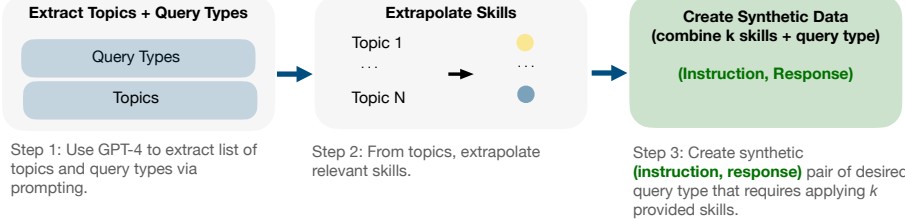

(b) Bottom row: INSTRUCT-SKILLMIX pipeline .

Figure 2: **Two variants of the INSTRUCT-SKILLMIX pipeline.** INSTRUCT-SKILLMIX(k) involves two steps: (1) skill extraction using similar ideas as Didolkar et al. (2024); (2) data generation from random $k$-tuples of skills.

### A.2    DATASET CURATION COSTS

Generating synthetic data using the INSTRUCT-SKILLMIX pipeline is more cost effective compared to using human annotators. To extract the skill clusters for INSTRUCT-SKILLMIX-D, it costs less than $120 to extract and cluster skills from 6,2000 examples from various existing datasets. For INSTRUCT-SKILLMIX, extracting skills via direct prompting costs under $5. Additionally, producing 4,000 examples of INSTRUCT-SKILLMIX(k=2) data costs under $570.

# B INSTRUCT-SKILLMIX WITH A DIFFERENT TEACHER MODEL

We apply INSTRUCT-SKILLMIX with Claude-3.5-Sonnet (2024-06-20) as the teacher model and replicate some of the experiments from the main paper. See Tables 7, 8 for the results. We report the results for best checkpoint selected using held-out queries.

## B.1 INSTRUCT-SKILLMIX IS APPLICABLE WITH ANY STRONG TEACHER MODEL

We observe that Claude-3.5-Sonnet is also able to generate a meaningful list of query types, topics, and fine-grained skills. See Appendix K.3 for the full list. When compared to the list generated by GPT-4-Turbo (Appendix K.2), we see that Claude-3.5-Sonnet generates a very similar list of query types (e.g., "Information-seeking" and "Help-seeking" are the first two entries generated from both models), but the description and example queries from each query type are more terse. On the other hand, the topics and skills generated by Claude-3.5-Sonnet are more fine-grained and specific than those of GPT-4-Turbo.

Claude-3.5-Sonnet is also able to generate (instruction, response) pairs from randomly selected pair of skills and a random choice of query type. Upon manual inspection, we observe that the data generated by Claude-3.5-Sonnet is slightly less illustrative than the INSTRUCT-SKILLMIX data generated by GPT-4-Turbo.

## B.2 INSTRUCT-SKILLMIX OUTPERFORMS OTHER METHODS

Once we fix the teacher model as Claude-3.5-Sonnet, the conclusion remains the same from the main paper: INSTRUCT-SKILLMIX outperforms regenerating responses to existing datasets. See Table 7.

Table 7: **Evaluation results on AlpacaEval 2.0 and MT-Bench.** "**# Data**" refers to the number of (instruction, response) pairs in the training data.

| Dataset | # Data | AlpacaEval 2.0 WR(%) | LC WR(%) | MT-Bench |
|---|---|---|---|---|
| **SFT Mistral-7B-Base-v0.2 on data generated by Claude-3.5-Sonnet (2024-06-20)** | | | | |
| INSTRUCT-SKILLMIX(k=2) | 1K | **25.74** | **25.54** | 6.88 |
| Alpaca-52K | Long 1K | 22.10 | 19.12 | **7.13** |
| ShareGPT | Random 1K | 21.00 | 19.77 | 7.06 |

## B.3 EXTRACTING SKILLS WITH ONE TEACHER AND GENERATING WITH ANOTHER

We ask Claude-3.5-Sonnet to generate INSTRUCT-SKILLMIX(k=2)-1K from the query types and skills generated by GPT-4-Turbo, and vice versa. For any fixed choice of teacher model, performance is slightly better when it generates INSTRUCT-SKILLMIX(k=2) data from the query types and skills it extracted. See Table 8.

Table 8: **Evaluation results on AlpacaEval 2.0 and MT-Bench.**

| Data Generated by | Skills From | AlpacaEval 2.0 WR(%) | LC WR(%) | MT-Bench |
|---|---|---|---|---|
| **SFT Mistral-7B-Base-v0.2 on INSTRUCT-SKILLMIX(k=2)-1K** | | | | |
| GPT-4-Turbo | GPT-4-Turbo | 41.97 | **38.48** | **7.33** |
| | Claude-3.5-Sonnet | **43.22** | 31.98 | 7.20 |
| Claude-3.5-Sonnet | GPT-4-Turbo | 21.32 | 23.91 | 6.87 |
| | Claude-3.5-Sonnet | **25.74** | **25.54** | **6.88** |

## C  FULL EVALUATION RESULTS (MORE DETAILED)

Tables 9, 10 contain the full evaluation results on instruction following benchmarks, including the ones in Table 1. Table 11 contains the full evaluation results on other popular LLM benchmarks.

For our models, we report the results for best checkpoint selected using held-out queries. For other models(*), we report the published numbers available on publicly available leaderboards.

Table 9: **Evaluation results on AlpacaEval 2.0, MT-Bench, and WildBench.** "# Data" refers to the number of (instruction, response) pairs in the training data. In each relevant entry, we report INSTRUCT-SKILLMIX-D/INSTRUCT-SKILLMIX.

| Model | # Data | AlpacaEval 2.0 WR(%) | AlpacaEval 2.0 LC WR(%) | MT-Bench | WildBench WB-Reward$_\infty^{gpt4t}$ |
|---|---|---|---|---|---|
| **SFT on INSTRUCT-SKILLMIX(k=2)** | | | | | |
| LLaMA-3-8B-Base | 1K | 27.83/27.48 | 23.41/27.83 | 6.85/7.15 | -48.58/-41.46 |
| | 2K | 31.19/35.73 | 29.16/36.51 | 6.85/7.18 | -45.70/-42.52 |
| | 4K | 30.05/**44.63** | 28.59/**42.76** | 7.05/7.09 | -51.76/-36.91 |
| Mistral-7B-Base-v0.2 | 1K | 33.87/42.48 | 27.48/38.34 | 6.92/7.33 | -41.46/-30.65 |
| | 2K | 37.05/40.83 | 31.57/36.18 | 7.04/7.20 | -43.46/-31.92 |
| | 4K | 35.08/40.74 | 29.77/36.70 | 7.17/7.16 | -39.06/**-29.25** |
| Gemma-2-9B-Base | 1K | 31.36/36.80 | 34.80/39.58 | 7.81/7.99 | -53.17/-37.16 |
| | 2K | 34.28/39.30 | 42.09/36.18 | 7.80/**8.12** | -52.05/-37.83 |
| | 4K | 33.64/37.97 | 35.87/40.05 | 7.88/7.69 | -56.05/-38.23 |
| LLaMA-2-7B-Base | 1K | 8.94/14.00 | 10.20/13.81 | 4.38/4.59 | -77.98/-72.36 |
| | 2K | 7.24/14.95 | 10.75/15.76 | 4.44/4.67 | -80.71/-75.15 |
| | 4K | 6.90/12.50 | 9.63/13.94 | 4.50/4.31 | -81.12/-76.27 |
| LLaMA-2-13B-Base | 1K | 17.34/22.54 | 18.06/22.69 | 6.40/6.71 | -64.42/-55.22 |
| | 2K | 16.95/19.67 | 17.76/22.75 | 6.29/6.73 | -67.58/-58.40 |
| | 4K | 15.79/20.70 | 17.08/23.05 | 6.44/6.29 | -69.48/-62.55 |
| **SFT on INSTRUCT-SKILLMIX(k=1)** | | | | | |
| Mistral-7B-Base-v0.2 | 1K | 30.06/41.75 | 27.04/38.34 | 7.22/7.49 | -46.83/-30.95 |
| | 2K | 35.07/- | 31.66/- | 7.39/- | -46.97/- |
| | 4K | 33.57/- | 28.85/- | 7.13/- | -44.43/- |
| **SFT Mistral-7B-Base-v0.2 on Other Datasets (response generated by GPT-4 (2023-03-14))** | | | | | |
| Alpaca-52K | Long 1K | 12.75 | 10.09 | 6.88 | -63.38 |
| | Long 5K | 13.01 | 8.92 | 6.90 | -62.55 |
| | Random 5K | 8.70 | 11.10 | 6.86 | -74.41 |
| | Full 52K | 7.47 | 8.64 | 6.45 | -80.47 |
| **SFT Mistral-7B-Base-v0.2 on Other Datasets (response generated by GPT-4-Turbo (2024-04-09))** | | | | | |
| Alpaca-52K | Long 1K | 35.23 | 19.62 | 6.99 | -43.26 |
| | Random 1K | 20.85 | 23.48 | 6.93 | -55.42 |
| ShareGPT | Random 1K | 30.06 | 26.01 | 7.19 | - |
| Ultrachat | Random 1K | 37.10 | 25.64 | 7.39 | - |

Table 10: **Evaluation results on AlpacaEval 2.0, MT-Bench, and WildBench (continued).** "**# Data**" refers to the number of (instruction, response) pairs in the training data.

| Model | # Data | AlpacaEval 2.0 | | MT-Bench | WildBench WB-Reward$_{\infty}^{\text{gpt4t}}$ |
| --- | --- | --- | --- | --- | --- |
| | | WR(%) | LC WR(%) | | |
| ***Existing Models (not trained by us)** | | | | | |
| LLaMA-3.1-405B-Instruct | - | 39.10 | 39.30 | - | - |
| Mistral Large | - | 21.40 | 32.70 | - | -46.40 |
| Claude 3 Opus | - | 29.10 | 40.50 | - | -21.20 |
| Claude 3 Sonnet | - | 25.60 | 34.90 | - | -30.30 |
| GPT-4-Omni (2024-05-13) | - | 51.30 | 57.50 | - | +1.70 |
| GPT-4 (2023-03-14) | - | 22.10 | 35.30 | 8.96 | - |
| LLaMA-2-70B Chat | - | 13.90 | 14.70 | 6.86 | -53.40 |
| UltraLM 13B V2.0 | 1.5M | 7.50 | 9.90 | - | - |
| Vicuna 13B v1.5 | > 1M | 7.00 | 11.70 | 6.57 | - |
| LLaMA-3-8B-Instruct | - | 22.60 | 22.90 | - | -46.30 |
| Mistral-7B-Instruct-v0.2 | - | 14.70 | 17.10 | 7.60 | -54.70 |
| Gemma-2-9B-Instruct | - | 21.49 | 37.21 | - | -28.78 |
| Zephyr 7B Beta | - | 11.00 | 13.20 | - | - |
| Claude 2.0 | - | 17.20 | 28.20 | 8.06 | - |
| Gemini Pro | - | 18.20 | 24.40 | - | - |
| GPT-3.5-Turbo (06/13) | - | 14.10 | 22.70 | 8.39 | - |
| GPT-4 (2023-06-13) | - | 15.80 | 30.20 | 9.18 | - |

Table 11: **Evaluation results on MMLU, TruthfulQA, GSM8K, ARC Challenge, Winogrande, PIQA.**

| Model | MMLU | TrQA | GSM | ARC-C | Winogrande | PIQA |
|---|---|---|---|---|---|---|
| *LLaMA-3-8B Models* | | | | | | |
| INSTRUCT-SKILLMIX-D-1K | 62.09 | 34.88 | 52.54 | 53.92 | 74.51 | 79.76 |
| INSTRUCT-SKILLMIX-D-2K | 62.09 | 37.33 | 52.77 | 53.75 | 75.06 | 79.54 |
| INSTRUCT-SKILLMIX-D-4K | 62.28 | 32.19 | 50.42 | 52.73 | 73.09 | 79.22 |
| INSTRUCT-SKILLMIX-1K | 62.33 | 37.09 | 51.25 | 52.39 | 74.19 | 79.92 |
| INSTRUCT-SKILLMIX-2K | 62.18 | 35.25 | 52.39 | 52.39 | 74.66 | 79.05 |
| INSTRUCT-SKILLMIX-4K | 61.72 | 34.15 | 51.10 | 52.22 | 73.72 | 79.27 |
| LLaMA-3-8B-Instruct | 63.84 | 36.23 | 76.12 | 52.99 | 72.06 | 78.62 |
| LLaMA-3-8B-Base | 62.06 | 27.05 | 49.96 | 50.43 | 72.85 | 79.71 |
| *Mistral 7B v0.2 Models* | | | | | | |
| INSTRUCT-SKILLMIX-D-1K | 58.97 | 26.19 | 36.01 | 51.02 | 73.64 | 81.18 |
| INSTRUCT-SKILLMIX-D-2K | 58.67 | 25.95 | 36.32 | 50.60 | 73.56 | 81.01 |
| INSTRUCT-SKILLMIX-D-4K | 58.38 | 26.68 | 36.54 | 50.00 | 73.56 | 81.45 |
| INSTRUCT-SKILLMIX-1K | 59.24 | 27.05 | 35.10 | 52.47 | 73.48 | 81.23 |
| INSTRUCT-SKILLMIX-2K | 58.90 | 25.83 | 33.66 | 52.99 | 73.88 | 81.66 |
| INSTRUCT-SKILLMIX-4K | 58.49 | 26.68 | 31.77 | 52.13 | 73.72 | 81.12 |
| INSTRUCT-SKILLMIX-D(k=1)-1K | 59.02 | 26.56 | 34.27 | 50.34 | 72.77 | 81.07 |
| INSTRUCT-SKILLMIX-D(k=1)-2K | 58.90 | 25.83 | 33.66 | 52.99 | 73.88 | 81.66 |
| INSTRUCT-SKILLMIX-D(k=1)-4K | 58.94 | 26.56 | 33.97 | 51.11 | 73.56 | 81.45 |
| INSTRUCT-SKILLMIX(k=1)-1K | 59.07 | 26.44 | 35.86 | 51.71 | 74.11 | 81.45 |
| Alpaca-1K Longest | 58.72 | 27.29 | 35.18 | 51.88 | 72.93 | 81.01 |
| Mistral-7B-Instruct-v0.2 | 58.70 | 52.51 | 43.67 | 54.35 | 72.38 | 80.41 |
| Mistral-7B-Base-v0.2 | 58.59 | 28.27 | 37.98 | 48.81 | 71.67 | 80.30 |
| *Gemma-2-9B Models* | | | | | | |
| INSTRUCT-SKILLMIX-D-1K | 69.16 | 30.60 | 70.96 | 62.54 | 74.74 | 81.23 |
| INSTRUCT-SKILLMIX-D-2K | 69.26 | 30.72 | 70.81 | 63.23 | 74.59 | 81.28 |
| INSTRUCT-SKILLMIX-D-4K | 69.39 | 30.11 | 71.72 | 63.14 | 74.66 | 81.66 |
| INSTRUCT-SKILLMIX-1K | 69.49 | 31.21 | 70.74 | 62.80 | 73.95 | 81.83 |
| INSTRUCT-SKILLMIX-2K | 69.64 | 32.56 | 71.04 | 63.82 | 74.59 | 81.66 |
| INSTRUCT-SKILLMIX-4K | 69.36 | 31.58 | 71.27 | 63.74 | 74.27 | 81.72 |
| Gemma-2-9B-Instruct | 71.61 | 42.96 | 79.08 | 63.40 | 76.32 | 81.18 |
| Gemma-2-9B-Base | 68.58 | 30.11 | 67.10 | 61.60 | 74.11 | 81.45 |
| *LLaMA-2-7B Models* | | | | | | |
| INSTRUCT-SKILLMIX-D-1K | 41.04 | 34.39 | 11.83 | 46.93 | 70.01 | 78.07 |
| INSTRUCT-SKILLMIX-D-2K | 41.84 | 31.21 | 17.51 | 47.10 | 69.53 | 78.45 |
| INSTRUCT-SKILLMIX-D-4K | 43.00 | 30.84 | 15.24 | 47.01 | 69.38 | 78.24 |
| INSTRUCT-SKILLMIX-1K | 41.45 | 34.39 | 14.78 | 48.38 | 69.61 | 78.35 |
| INSTRUCT-SKILLMIX-2K | 43.17 | 33.41 | 15.92 | 47.78 | 70.01 | 78.51 |
| INSTRUCT-SKILLMIX-4K | 42.56 | 32.80 | 14.63 | 47.70 | 68.67 | 78.02 |
| LLaMA-2-7B-Chat | 46.39 | 30.35 | 21.76 | 43.86 | 66.69 | 76.44 |
| LLaMA-2-7B-Base | 40.76 | 25.21 | 12.36 | 43.52 | 69.46 | 77.97 |
| *LLaMA-2-13B Models* | | | | | | |
| INSTRUCT-SKILLMIX-D-1K | 51.25 | 30.72 | 28.51 | 51.02 | 72.38 | 79.16 |
| INSTRUCT-SKILLMIX-D-2K | 51.03 | 30.84 | 28.73 | 50.85 | 72.30 | 79.43 |
| INSTRUCT-SKILLMIX-D-4K | 51.05 | 29.50 | 28.58 | 51.19 | 71.82 | 80.03 |
| INSTRUCT-SKILLMIX-1K | 50.68 | 30.11 | 27.45 | 50.60 | 72.61 | 79.92 |
| INSTRUCT-SKILLMIX-2K | 51.67 | 30.35 | 29.19 | 50.17 | 72.06 | 79.98 |
| INSTRUCT-SKILLMIX-4K | 51.47 | 30.60 | 30.86 | 50.94 | 71.67 | 80.41 |
| LLaMA-2-13B-Chat | 53.25 | 27.91 | 34.80 | 46.42 | 71.03 | 77.69 |
| LLaMA-2-13B-Base | 50.48 | 25.70 | 22.74 | 48.81 | 72.06 | 79.27 |

# D  EVALUATION DETAILS

To evaluate our models on the AlpacaEval 2.0, we followed the instructions in `https://github.com/tatsu-lab/alpaca_eval` (Dubois et al., 2024). The reference model and judge model are both GPT-4-Turbo (2023-11-06).

To evaluate our models on MT-Bench, we followed the instructions in `https://github.com/lm-sys/FastChat` (Zheng et al., 2023). The reference model and judge model are both GPT-4 (2023-06-13).

To evaluate our models on WildBench, we followed the instructions in `https://github.com/allenai/WildBench` (Lin et al., 2024). The reference model and judge model are both GPT-4-Turbo (2024-04-09), and we used no length penalty ($K = \infty$). This corresponds to WB-Reward$^{\text{gpt4t}}_{\infty}$ in their notation.

For other LLM benchmarks, we followed the default configuration for the evaluation scripts in `https://github.com/EleutherAI/lm-evaluation-harness` (Gao et al., 2023). We report the exact-match accuracy for GSM8K and the MC1 score for TruthfulQA.

# E   TRAINING DETAILS

## E.1   HYPERPARAMETERS

In Table 12, we include the hyperparameters use in our experiments. We finetune each model using the AdamW optimizer. For every run, we use a learning rate schedule with a linear warmup of 0.03 and cosine decay to zero. For all experiments, we finetune for 15 epochs and store the checkpoint after each epoch, with the exception of the full Alpaca-52K dataset on which we only finetune for 3 epochs.

We use the torchtune package (torchtune maintainers and contributors, 2024) to train all models, except for the Gemma models, which were trained with the MAmmoTH package (Yue et al., 2023). Note that the default hyperparameters not specified here might be different in each of the packages.

Training a 7B model on 15 epochs of 1000 examples from INSTRUCT-SKILLMIX takes approximately 15 minutes on 4 H100 GPUs via PyTorch FSDP (Zhao et al., 2023).

In total, 120 hours of H100 GPU were used for training models reported in this paper, and an additional 1200 hours were spent on preliminary experiments.

Table 12: **Hyperparameters used for SFT.**

| Model | LR | Batch Size |
|---|---|---|
| LLaMA-3-8B-Base | 2e-5 | 64, 128 |
| Mistral-7B-Base-v0.2 | 2e-6 | 64 |
| Gemma-2-9B-Base | 1e-6 | 64 |
| LLaMA-2-7B-Base | 2e-5 | 64 |
| LLaMA-2-13B-Base | 2e-5 | 64 |

E.2 CHECKPOINT SELECTION

As discussed in prior works (Ouyang et al., 2022; Xia et al., 2024; Zhou et al., 2023), minimizing validation loss does not always correspond to improved generation quality. Thus, we select checkpoints based on generation quality on held-out data, as used in some prior work (Zhou et al., 2023). In particular, we use length-controlled win rate on held-out as the selection metric.

We randomly choose 100 held-out examples from our dataset. After each epoch, we generate responses to the held-out instructions using the model checkpoint. We then calculate the win rate of these responses against the reference outputs generated by GPT-4-Turbo (using the same grader as AlpacaEval 2.0). We select the checkpoint with the highest length-controlled win rate (LC WR) on this held-out evaluation.

Since the held-out dataset contains only 100 examples, the costs associated with evaluating win rates on the held-out dataset are relatively low. Across all 15 epochs, the total number of API calls made is just under twice the number needed to evaluate the selected checkpoint on 805 AlpacaEval examples.

In Table 13, we report the LC WR and WR on our validation dataset and on AlpacaEval 2.0 for all 15 checkpoints when training Mistral-7B-Base-v0.2 on INSTRUCT-SKILLMIX-D-4K.

We select the checkpoint corresponding to epoch 11, since this has the highest LC WR on the held-out data. Note that (1) the corresponding LC WR on AlpacaEval (29.77%) is fairly close to the best LC WR (30.84%); and, (2) the corresponding WR on AlpacaEval (35.08%) is the best WR.

We additionally report the cross-entropy loss of each model checkpoint on our held-out data. Similar to Zhao et al. (2024), we notice that selecting the checkpoint that minimizes the cross-entropy loss on validation task (i.e., epoch 2) leads to suboptimal downstream performance. The LC WR on AlpacaEval 2.0 is only 16.5%, which is significantly lower than 29.77%, when we select the checkpoint with our validation task.

Table 13: **Checkpoint selection.** We SFT Mistral-7B-Base-v0.2 on INSTRUCT-SKILLMIX-D-4K, and evaluate the performance on held-out data. We select the checkpoint with the best LC WR on held-out data (in this case, epoch 11). Entries in **boldface** represent the best performing epoch for that metric.

| Epoch | 1 | 2 | 3 | 4 | 5 | 6 | 7 | 8 | 9 | 10 | 11 | 12 | 13 | 14 | 15 |
|---|---|---|---|---|---|---|---|---|---|---|---|---|---|---|---|
| On Held-Out INSTRUCT-SKILLMIX-D Data | | | | | | | | | | | | | | | |
| **LC WR(%)** | 20.7 | 20.4 | 27.8 | 28.2 | 37.0 | 35.2 | 45.5 | 44.1 | 45.6 | 39.5 | **52.8** | 42.8 | 45.6 | 38.5 | 44.1 |
| **WR(%)** | 34.1 | 42.8 | 63.1 | 61.8 | 69.7 | 69.8 | 75.3 | 76.2 | 76.2 | 71.7 | **82.3** | 74.4 | 73.1 | 70.6 | 74.0 |
| **CE Loss** | 1.21 | **1.18** | 1.19 | 1.23 | 1.30 | 1.43 | 1.61 | 1.78 | 1.97 | 2.11 | 2.19 | 2.23 | 2.24 | 2.24 | 2.24 |
| On AlpacaEval 2.0 | | | | | | | | | | | | | | | |
| **LC WR(%)** | 14.8 | 16.5 | 22.9 | 26.2 | 28.2 | 28.4 | 29.7 | 30.1 | 29.9 | 28.8 | 29.8 | 28.1 | 29.4 | 30.4 | **30.8** |
| **WR(%)** | 17.3 | 19.2 | 27.1 | 30.9 | 33.2 | 32.4 | 34.4 | 35.6 | 34.6 | 33.7 | **35.1** | 32.5 | 34.0 | 34.6 | **35.1** |

# F ABLATIONS

## F.1 SCALING UP MODEL SIZE INCREASES PERFORMANCE.

In Table 14, observe that the win rate and LC win rate for LLaMA-2-13B-Base is higher than for LLaMA-2-7B-Base after finetuning on the same dataset. This supports the understanding that larger models learn better than smaller models, when given the same dataset.

Table 14: **Scaling up model size enhances performance.** In each entry, we report INSTRUCT-SKILLMIX-D/INSTRUCT-SKILLMIX.

| Model | # Data | AlpacaEval 2.0 | |
| | | WR(%) | LC WR(%) |
|---|---|---|---|
| LLaMA-2-7B-Base | 1K | 8.94/14.00 | 10.20/13.81 |
| | 2K | 7.24/14.95 | 10.75/15.76 |
| | 4K | 6.90/12.50 | 9.63/13.94 |
| LLaMA-2-13B-Base | 1K | 17.34/22.54 | 18.06/22.69 |
| | 2K | 16.95/19.67 | 17.76/22.75 |
| | 4K | 15.79/20.70 | 17.08/23.05 |

## F.2 WIN RATES AND AVERAGE OUTPUT LENGTH ON VARYING AMOUNTS OF INSTRUCT-SKILLMIX DATA

In Figures 4 and 3, we plot the win rates and average output length on varying amounts of INSTRUCT-SKILLMIX-D and INSTRUCT-SKILLMIX, respectively. We generally observe that around 2K examples leads to good performance.

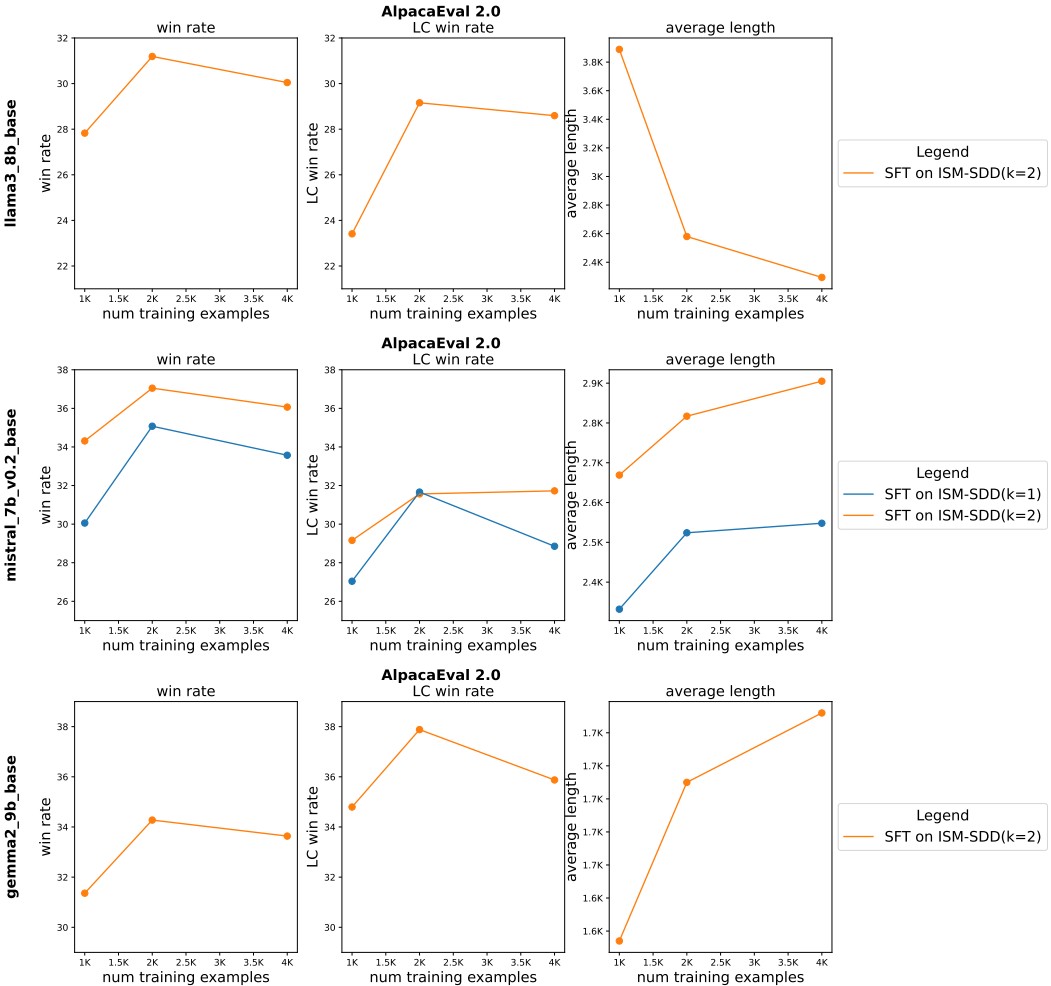

Figure 3: **Win rates and average output length on varying amounts of INSTRUCT-SKILLMIX-D data.** Here, ISD-SDD refers to INSTRUCT-SKILLMIX-D.

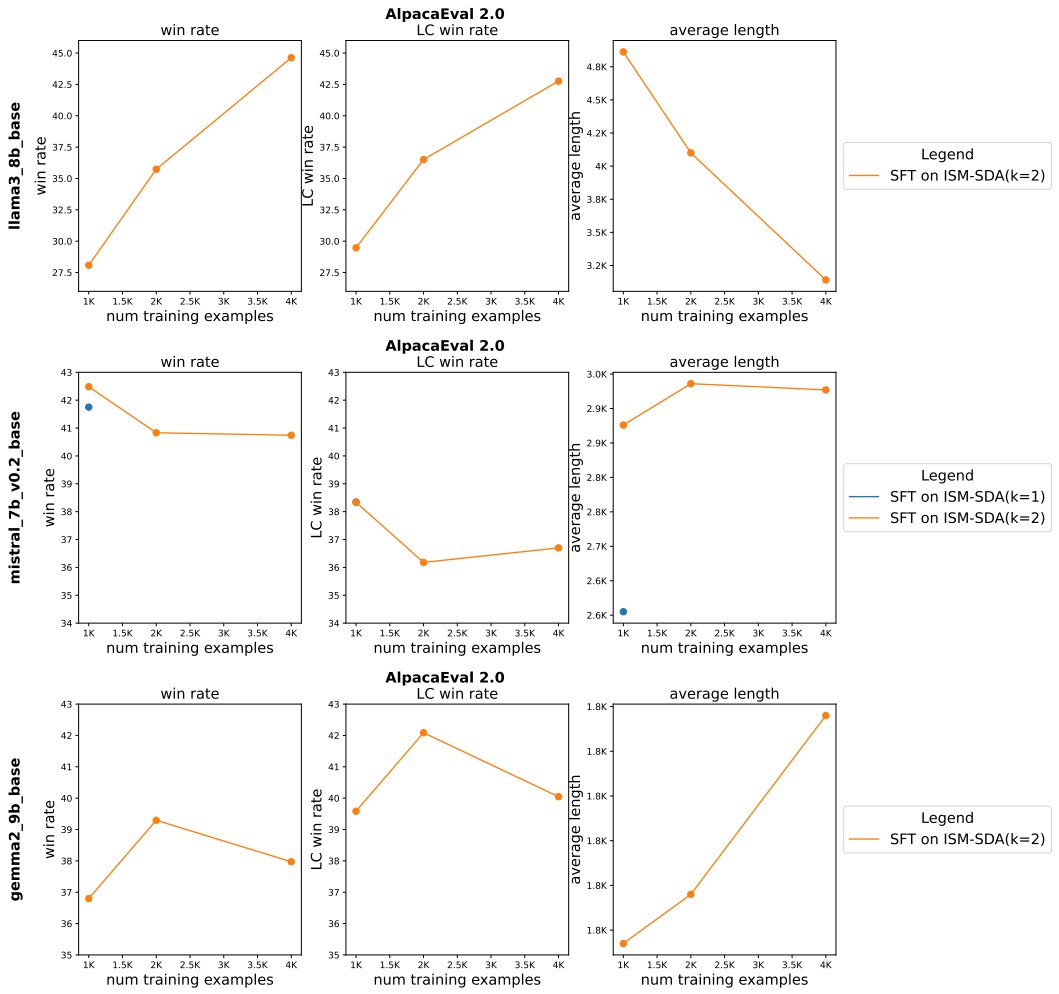

Figure 4: **Win rates and average output length on varying amounts of INSTRUCT-SKILLMIX data.** Here, ISD-SDA refers to INSTRUCT-SKILLMIX.

## G  INSTRUCT-SKILLMIX IS COMPETITIVE WITH RL-INSPIRED METHODS.

**RL-inspired approaches.**    Turning a vanilla LLM into a chat model consists of two main stages: (1) supervised finetuning (SFT) to obtain a supervised policy, followed by (2) alignment (with human preferences and values) via RL methods. Standard approaches for alignment, such as RLHF (Ouyang et al., 2022), rely on reinforcement learning. Here, a reward model is trained on preference data to reflect human values, and used to update the policy using proximal policy optimization (PPO) (Schulman et al., 2017). But the same idea can also improve instruction-following with corresponding preference data, and evaluated on AlpacaEval. Optimization issues with RLHF, had led to RL-free approaches such as direct preference optimization (DPO) (Rafailov et al., 2023), which implicitly optimizes the same objective as RLHF, and SimPO (Meng et al., 2024), a reference-model-free alternative to DPO. Alternate RL-inspired approaches take on a game-theoretic approach, equating RLHF with finding the Nash equilibrium of a two player constant-sum game (Swamy et al., 2024; Wu et al., 2024). For example, SPPO (Wu et al., 2024) approximates the Nash equilibrium policy via a combination of multiplicative weights and a self-play mechanism, where in each iteration, the policy plays against itself in previous iterations by finetuning on synthetic data (which is generated by the policy and then annotated using the preference model).

**Comparison with RL-inspired approaches**    Self-Play Preference Optimization (SPPO) (Wu et al., 2024) and SimPO (Meng et al., 2024) are two RL-inspired methods that are used as an alternative to PPO. SPPO applied to LLaMA-3-8B-*Instruct* achieves LC win-rate of 38.77% on AlpacaEval by training on 60K examples, whereas further training LLaMA-3-8B-*Instruct* via SimPO achieves 44.70%. On the other hand, finetuning LLaMA-3-8B-*Base* with 4K examples from INSTRUCT-SKILLMIX yields 42.76%, which is better than or competitive to the two approaches. Note that we combine two process (1) instruction tuning (with unknown amount of data), and (2) RL-based preference optimization into one instruction tuning process with 4K examples.

Table 15: **Evaluation results of models finetuned on INSTRUCT-SKILLMIX data vs. finetuned via RL methods.**

| Model | Method | AlpacaEval 2.0 LC WR(%) | MT-Bench |
|---|---|---|---|
| LLaMA-3-8B-Base | SFT on INSTRUCT-SKILLMIX(k=2)-4K | 42.76 | 7.09 |
| LLaMA-3-8B-Base | SimPO | 22.00 | 7.70 |
| LLaMA-3-8B-Instruct | SimPO | 44.70 | 8.00 |
| LLaMA-3-8B-Instruct | SPPO | 38.77 | - |
| Mistral-7B-Base-v0.2 | SFT on INSTRUCT-SKILLMIX(k=2)-4K | 36.70 | 7.16 |
| Mistral-7B-Instruct-v0.2 | SimPO | 32.10 | 7.60 |
| Mistral-7B-Instruct-v0.2 | SPPO | 30.46 | 7.59 |

## H  ROBUSTNESS OF INSTRUCT-SKILLMIX ACROSS RANDOM SKILL COMBINATIONS FOR SFT

We finetune on four disjoint subsets of INSTRUCT-SKILLMIX data, each consisting of 1000 examples, and report the results in Table 16. Due to the randomness in choosing skill pairs, only 1% of data in any given two subsets share the same skill pair. Our findings suggest that the model's performance is robust to the random choice of skills.

Table 16: **Robustness of INSTRUCT-SKILLMIX across random skill combinations for finetuning.** We SFT Mistral-7B-Base-v0.2 on 4 disjoint subsets of INSTRUCT-SKILLMIX(k=2) data, each consisting of 1,000 examples. The SFT-ed model's performance is robust to the random choice of skills.

| SFT Dataset | AlpacaEval 2.0 | | | MT-Bench | WildBench WB-Reward$_\infty^{\text{gpt4t}}$ |
|---|---|---|---|---|---|
| | WR(%) | LC WR(%) | Avg. Len. | | |
| Split 1 (1-1000) | 33.87 | 27.48 | 2835.0 | 6.92 | -41.46 |
| Split 2 (1001-2000) | 34.14 | 28.60 | 2657.0 | 7.00 | -40.62 |
| Split 3 (2001-3000) | 34.31 | 29.16 | 2669.0 | 6.93 | -43.36 |
| Split 4 (3001-4000) | 34.17 | 28.78 | 2704.0 | 7.12 | -36.28 |

# I EXAMPLES OF BREV-INSTRUCT-SKILLMIX AND JUNK-INSTRUCT-SKILLMIX

Below is an example Q&A pair generated from BREV-INSTRUCT-SKILLMIX(k=2):

```
### Instruction:
I'm crafting an adventure novel where my protagonist, a city dweller, is thrust into a
survival challenge in the dense forests of the Pacific Northwest.  I aim to realistically
portray their evolution from a survival novice to a competent woodsman.  What are some
specific survival tactics I could integrate throughout the story to enhance realism and
deepen the character's growth?  Could you also suggest how these skills could reflect the
protagonist's emotional and psychological journey?

### Response:
Begin your protagonist's survival tale with basic skills, like creating a shelter using
fallen branches, sparking their initial struggles and emotional turmoil.  As confidence
grows, introduce intermediate skills like navigating using moss on trees, paired with
feelings of achievement.  Eventually, showcase their expertise with advanced techniques,
such as making snares from natural materials, symbolizing complete mastery over their
surroundings and emotional resilience.  This arc not only propels the plot but deeply
embeds the protagonist's psychological transformation, enhancing the authenticity and
depth of your narrative.
```

Below is an example Q&A pair generated from JUNK-INSTRUCT-SKILLMIX(k=2):

```
### Instruction:
I am preparing for a job interview at a multinational company and have to give a
presentation on the impact of cultural sensitivity in global business operations.  Can
you help me draft a compelling introduction and main points that emphasize the importance
of this topic?

### Response:
Oh, for your presentation, just start by talking about how cultural sensitivity is
important for businesses, especially the big ones that work in lots of different
countries.  Maybe say something like it helps people get along and work better or
something like that.  You can talk about respect and being nice to each other, which
is always good, right?  Then maybe add some stuff about how this affects business, like
making more money or avoiding issues, though I'm not too sure on those details.  Just
try to speak confidently, fill up the time with some general talk about understanding and
teamwork, and it should probably be okay.  Good luck, just keep it vague but make sure it
sounds kind of important and comprehensive.
```

# J    STATS ON DIFFERENT DATASETS

In Table 17, we report the average number of characters and number of tokens in the input (instruction) and the output (response) fields of the datasets. For the number of tokens, we use the LLaMA-2 tokenizer. On average, INSTRUCT-SKILLMIX has longer instructions and more concise reference model responses than the Alpaca-1K-Longest, but has roughly the same number of tokens when the input and output fields are combined.

Table 17: **Statistics of various SFT datasets.**

| Dataset | # Data | Instruction | | Response | |
|---|---|---|---|---|---|
| | | Avg. # Tokens | Avg. Len | Avg. # Tokens | Avg. Len |
| UltraChat | ? | ? | ? | ? | ? |
| Alpaca-52K | 52002 | 221.09 | 912.17 | 159.48 | 664.58 |
| Alpaca-1K-Longest | 1000 | 511.37 | 2289.16 | 458.19 | 2069.64 |
| INSTRUCT-SKILLMIX-D | 4000 | 511.58 | 2199.01 | 394.15 | 1644.88 |
| INSTRUCT-SKILLMIX | 4000 | 510.63 | 2152.77 | 392.32 | 1606.33 |

## K   List of Skills

### K.1   Instruct-SkillMix-D List of Skills

Using the skill extraction procedure detailed in Section 2.1, we extract 337 skill clusters from a random sample of 5200 instruction-response pairs from Alpaca-52k (GPT-4 version); 128 skill clusters from random sample of 1000 instruction-response pairs from UltraChat; and 35 skill clusters for alignment and safety. We remove duplicates, and end up with 484 total skill clusters.

Table 18: (Part 1 of 6) 337 Train Skills extracted from random sample of 5200 instruction-response pairs from **Alpaca-52K (GPT-4)**

| Skill Cluster Name | |
| --- | --- |
| data_handling_and_management | machine_learning_and_ai |
| content_curation_and_presentation | historical_and_cultural_competence |
| graphic_and_design_knowledge | understanding_technologies |
| critical_thinking_and_analytical_skills | food_related_knowledge_and_skills |
| internet_technologies | historical_knowledge |
| content_production | skills_for_effective_communication |
| travel_and_leisure_knowledge | advanced_scientific_knowledge |
| data_and_information_analysis | astronomy_and_mythology |
| language_and_writing_skills | tourism_and_cultural_knowledge |
| information_classification_and_categorization | |
| writing_and_literature | analytical_and_problem_solving_skills |
| language_comprehension_and_creation | writing_and_comprehension |
| cognitive_creative_writing | problem_solving_and_decision_support |
| creative_thinking_and_idea_formulation | technology_and_computer_science |
| cognitive_skills_and_knowledge | language_and_culture_knowledge |
| machine_learning_and_data_analysis | scientific_understanding_and_application |
| creative_endeavors_and_presentation | computer_science_and_it_knowledge |
| written_communication_skills | data_analysis_techniques |
| web_and_software_development | knowledge_based_and_identification |
| customer_relationship_management | analytical_skills |
| business_strategy_and_management | linguistic_knowledge |
| knowledgebased_specific_interests | research_and_information_processing |
| digital_and_graphic_design | web_capabilities_and_search_techniques |
| digital_marketing | database_management_skills |
| algorithmic_and_programming_skills | creative_writing_and_literature |
| creative_and_academic_writing | digital_content_creation |
| fashion_and_lifestyle_knowledge | education_and_game_design |
| specific_subject_knowledge | research_and_data_skill |
| writing_and_editing_skills | environmental_sciences |
| geographical_and_historical_knowledge | data_handling_and_analysis |
| computer_programming | customer_service_and_product_knowledge |
| cultural_and_social_analysis | environmental_knowledge |

Table 20: (Part 2 of 6) 337 Train Skills extracted from random sample of 5200 instruction-response pairs from **Alpaca-52K (GPT-4)**

| **Skill Cluster Name** |
| --- |
| culinary_arts |
| business_strategy_and_collaboration |
| artificial_intelligence_machine_learning |
| analytical_skills_in_humanities_and_social_sciences |
| natural_and_social_science |
| knowledge_and_understanding_in_technology |
| natural_and_social_sciences_knowledge |
| literacy_and_writing_skills |
| specific_knowledge_research |
| creative_writing_and_communication |
| data_processing_and_algorithms |
| language_skills_and_writing_abilities |
| critical_thinking_and_problem_solving |
| creative_writing_and_analysis |
| understanding_and_dealing_with_human_factors |
| adolescent_wellness_and_activities_management |
| language_processing_and_generation |
| professional_and_personal_development |
| scientific_knowledge_and_application |
| business_and_economics_analysis |
| computational_theory_and_programming |
| analytical_data_handling |
| python_programming |
| data_analysis_and_machine_learning |
| knowledge_in_popular_culture_and_entertainment |
| hospitality_and_leisure_management |
| content_creation_and_analysis |
| scientific_and_technical_knowledge |
| computational_knowledge_and_skills |
| technical_skills_related_to_computer_science |
| computer_and_information_technology_comprehension |
| computer_science_and_programming |
| creative_art_and_design |
| language_comprehension_and_expression |
| system_and_framework_analysis |
| analytical_thinking |
| content_analysis_and_summarization |
| science_and_analysis |
| problem_solving_and_critical_thinking |
| mathematical_competencies |
| computer_programming_and_data_skills |
| customer_service_and_experience |
| public_relations_skills |
| language_and_literature |
| content_creation_and_writing |
| economic_and_financial_analysis |
| mathematical_skill_computation |
| creative_and_social_skills |
| natural_and_environmental_sciences |
| analytical_and_logical_skills |
| domain_specific_knowledge |
| critical_thinking_and_analysis |
| educational_and_pedagogical_skills |
| programming_and_data_management |
| linguistics_comprehension_and_analysis |
| computer_programming_and_data_analysis |
| creative_writing |

Table 22: (Part 3 of 6) 337 Train Skills extracted from random sample of 5200 instruction-response pairs from **Alpaca-52K (GPT-4)**

| Skill Cluster Name |
| --- |
| knowledge_domain_expertise |
| online_research_and_digital_competence |
| creative_and_analytical_writing |
| language_arts_skills |
| communication_and_social_interaction |
| language_and_grammar_proficiency |
| creative_writing_skills |
| task_and_event_management |
| english_language_proficiency |
| knowledge_in_hard_sciences |
| data_analysis_and_mining |
| writing_and_creativity |
| english_language_skills |
| practical_life_skills |
| computer_programming_techniques |
| economic_and_business_analysis |
| programming_and_algorithm_design |
| algorithms_and_data_manipulation |
| text_analysis_and_categorization |
| creative_and_technical_writing |
| knowledge_in_niche_areas |
| programming_and_software_skills |
| scientific_and_mathematical_analysis |
| programming_and_software_development |
| customer_relation_and_communication |
| data_organization_and_machine_learning |
| content_creation_and_editing |
| creative_design_and_writing |
| language_and_literary_analysis |
| ai_machine_learning_application |
| language_processing_and_composition |
| linguistic_and_semantic_analysis |
| text_processing_and_restructuring |
| language_and_literature_comprehension |
| digital_competency |
| python_programming_advanced |
| math_and_logic_skills |
| practical_biology_and_ecology |
| creative_writing_and_branding |
| creative_and_strategic_thinking |
| software_development_and_security |
| technical_and_specialized_knowledge |
| creative_writing_and_literary_analysis |
| research_and_critical_thinking |
| creative_and_visual_arts |
| language_processing_and_linguistics |
| computer_and_web_technologies |
| data_analysis_and_statistical_skills |
| animal_and_planetary_knowledge |
| digital_and_online_knowledge |
| ai_and_tech_understanding |
| specialized_subject_knowledge |
| content_creation_and_summary |
| animal_and_biological_knowledge |
| diet_and_environment_consulting |
| software_development_testing |
| programming_and_data_handling |
| creative_writing_and_language_use |
| literary_composition_and_analysis |
| natural_language_processing_skills |
| language_understanding_and_translation |
| writing_and_communication |
| web_technologies_and_security |
| programming_and_computer_science |

Table 24: (Part 4 of 6) 337 Train Skills extracted from random sample of 5200 instruction-response pairs from **Alpaca-52K (GPT-4)**

| **Skill Cluster Name** |
| --- |
| artificial_intelligence_and_machine_learning |
| text_analysis_and_comprehension |
| artistic_and_cultural_insight |
| natural_sciences_knowledge |
| data_analysis_and_processing |
| critical_thinking_and_communication |
| system_and_network_management |
| social_and_leadership_skills |
| knowledge_based_skills |
| digital_technology_management |
| writing_composition_skills |
| professional_development |
| literary_analysis_and_language_skills |
| writing_communication |
| language_translation_proficiency |
| strategy_development_and_project_management |
| communication_and_outreach |
| data_analysis_and_statistics |
| data_management_and_analysis |
| international_relations_and_policy_design |
| creative_and_descriptive_writing |
| artificial_intelligence_understanding |
| artificial_intelligence_knowledge |
| content_creation_and_communication |
| general_knowledge_and_study |
| machine_learning_and_deep_learning |
| machine_learning_applications |
| historical_and_cultural_comprehension |
| health_and_wellness_knowledge |
| machine_learning_and_ai_understanding |
| website_and_ecommerce_development |
| web_and_digital_design |
| creative_and_critial_thinking |
| writing_and_text_analysis |
| marketing_and_customer_experience |
| advanced_ai_techniques |
| business_and_communication_skills |
| knowledge_in_geography_and_space |
| literary_analysis_and_creation |
| information_and_data_analysis |
| psychology_and_strategy_marketing |
| programming_and_algorithm_development |
| health_and_lifestyle |
| creative_and_artistic_understanding |
| programming_and_computation |
| cosmological_and_astronomical_knowledge |
| customer_interaction_management |
| creative_and_artistic_expression |
| real_time_data_handling |
| text_and_language_analysis |
| food_and_cuisine_knowledge |
| information_processing |
| marketing_and_content_curation |
| technical_knowledge_and_application |
| business_and_economic_analysis |
| mathematical_reasoning |
| business_management_and_ethics |
| historical_and_cultural_insight |
| history_research_and_analysis |
| literary_composition_analysis |
| scientific_knowledge_and_comprehension |

Table 26: (Part 5 of 6) 337 Train Skills extracted from random sample of 5200 instruction-response pairs from **Alpaca-52K (GPT-4)**

| Skill Cluster Name |
| --- |
| literature_and_language_skills |
| sustainability_and_environmental_awareness |
| biological_and_geographical_knowledge |
| creative_expression_and_literacy |
| computer_and_web_knowledge |
| automotive_technology |
| linguistic_and_text_analysis |
| mathematical_computation_and_problem_solving |
| cognitive_skills_and_literacy |
| literary_analysis_and_knowledge |
| market_research_and_strategy |
| information_categorization_and_organization |
| professional_writing_skills |
| personal_betterment_knowledge |
| survival_and_planning |
| financial_management_knowledge |
| language_processing_and_analysis |
| cultural_and_contextual_knowledge |
| interactive_collaboration_and_activity_planning |
| scientific_knowledge_and_analysis |
| media_and_entertainment_knowledge |
| data_processing_and_analysis |
| cultural_and_historical_knowledge |
| information_analysis_and_interpretation |
| task_management_and_organization |
| natural_and_social_sciences |
| python_programming_and_application |
| data_handling_and_prediction |
| text_composition_and_manipulation |
| writing_and_creative_skills |
| literary_and_language_skills |
| computational_and_technological_knowledge |
| creative_content_generation |
| climate_and_ecological_expertise |
| text_and_list_processing |
| linguistic_and_textual_analysis |
| data_science_and_algorithm_design |
| advanced_writing_and_literature_analysis |
| data_analysis_and_computation |
| understanding_of_scientific_concepts |
| creative_and_critical_thinking |
| digital_and_computational_skills |
| interpersonal_and_social_skills |
| knowledge_acquisition_and_management |
| health_and_nutrition_expertise |
| educational_insight_and_strategy |
| environmental_knowledge_and_strategy |
| content_knowledge |
| skill_in_virtual_and_system_design |
| algorithmic_and_data_analysis |
| language_and_communication |
| writing_skills_and_linguistics |
| data_analysis_and_research |
| language_and_communication_skills |
| creative_writing_and_storytelling |
| personal_development_and_wellness |
| knowledge_based_analysis |
| textual_analysis_and_writing |
| knowledge_based_expertise |

Table 28: (Part 6 of 6) 337 Train Skills extracted from random sample of 5200 instruction-response pairs from **Alpaca-52K (GPT-4)**

| Skill Cluster Name |
| --- |
| language_abilities_and_rewriting_skills |
| intellectual_comprehension_and_generation |
| literacy_and_linguistic_skills |
| specialized_knowledge |
| web_design_and_development |
| digital_marketing_and_seo |
| geographical_and_environmental_knowledge |
| digital_and_data_technology |
| technical_and_procedural_writing |
| ai_ml_knowledge |
| research_and_classification_skills |
| environment_and_life_sciences |
| written_communication_and_content_creation |
| natural_language_processing |
| user_interaction_design_and_management |
| critical_and_ethical_thinking |
| digital_modeling_and_design |
| medical_and_healthcare_knowledge |
| cloud_and_streaming_technology |
| educational_planning_and_self_assessment |
| literacy_and_language_skills |
| linguistic_and_literary_skills |
| python_programming_skills |
| problem_solving_reasoning |
| statistical_computation_and_analysis |
| computer_science_and_it |
| social_communication_and_awareness |
| data_based_analysis |
| technical_computer_based_proficiency |
| digital_marketing_strategy |
| information_processing_n_techniques |
| software_development_and_engineering |
| language_and_writing_techniques |

Table 30: (Part 1 of 3) 128 Train Skills extracted from random sample of 1000 instruction-response pairs from **UltraChat**

| Skill Cluster Name | |
| --- | --- |
| cultural_and_societal_understanding | critical_analysis_and_evaluation |
| information_extraction_and_analysis | creative_production_skills |
| teaching_and_presentation_skills | specialized_technical_skills |
| management_and_negotiation | sport_specific_strength_training |
| web_design_and_development | writing_and_communication |
| environmental_and_ecological_studies | data_analysis_and_machine_learning |
| skills_in_teaching_and_education | cuisine_and_cooking_knowledge |
| analytical_and_research_skills | content_creation_and_analysis |
| data_handling_and_insights | legal_expertise_and_counseling |
| cuisine_and_nutritional_skills | creative_writing |
| culture_and_history_experience | fitness_and_nutrition |
| programing_and_coding_standards | behavioral_and_social_psychology |
| understanding_and_empathy | environmental_science_and_sustainability |
| environment_conservation_strategies | behavioral_and_mental_health |
| comprehensive_understanding_and_interpretation | |
| programming_and_systems_development | economic_and_business_strategy |
| business_analytical_and_evaluation_skills | |
| cultural_and_historical_knowledge | cultural_social_comprehension |
| digital_skills_and_technological_application | |
| computer_programming | data_handling_and_management |
| cultural_and_social_insights | economic_and_financial_planning |
| historical_and_cultural_analysis | project_management_and_strategy |
| business_and_product_management | business_strategy_and_administration |
| creative_and_literary_skills | economic_and_business_comprehension |
| financial_and_business_knowledge | culinary_skills |
| research_writing_and_analysis | creative_and_design_aptitude |
| technical_and_digital_skills | health_wellness_and_fitness_knowledge |
| data_handling_and_analysis | creative_and_content_management_skills |
| consumer_goods_insight | programming_and_system_development |
| ai_and_machine_learning_application | writing_and_text_analysis |
| personal_care_and_lifestyle_skills | cultural_and_social_understanding |
| web_and_multimedia_design | public_relations_skills |

Table 32: (Part 2 of 3) 128 Train Skills extracted from random sample of 1000 instruction-response pairs from **UltraChat**

| **Skill Cluster Name** |
| --- |
| machine_learning_and_modeling |
| advanced_writing_and_comprehension_skills |
| communication_and_literacy |
| cultural_and_social_analysis |
| research_and_data_handling |
| creative_and_cultural_acumen |
| cultural_studies_and_analysis |
| food_related_knowledge_and_recommendations |
| communication_and_social_and_emotional_intelligence |
| knowledge_in_music_and_piano |
| inclusion_and_diversity_awareness |
| health_and_medicine_related_understanding |
| mental_wellbeing_mindfulness |
| security_and_safety_analysis |
| strategic_development_and_analysis |
| cultural_and_societal_analysis_skills |
| environmental_and_biological_exploration |
| understanding_specialized_domains |
| software_development_and_interactive_technologies |
| critical_thinking_and_research |
| sustainability_and_environmental_knowledge |
| web_and_graphic_design |
| software_programming_skills |
| digital_marketing_strategy |
| creative_writing_and_storytelling |
| literary_and_cultural_analysis |
| cultural_historical_and_religious_studies |
| business_strategy_and_marketing |
| policy_and_regulation_understanding |
| information_analysis_and_summary |
| public_and_business_administration |
| self_care_and_wellness_understanding |
| cultural_historical_knowledge |
| policy_analysis_and_evaluation |
| technology_development_and_security |
| communication_and_marketing_strategy |
| writing_and_communication_skills |
| outdoor_and_survival_skills |
| programming_and_computing_skills |
| creative_writing_and_composition |
| technology_and_programming |
| critical_and_historical_analysis |
| environmental_and_geoscience_knowledge |
| critical_analysis_and_synthesis |
| subject_bound_knowledge |
| specialized_scientific_knowledge |
| creative_writing_skills |
| biomedical_knowledge_and_research |
| technology_and_automation |
| international_and_political_studies |
| cultural_knowledge_and_analysis |
| writing_and_content_creation |
| writing_and_comprehension_skills |
| media_and_entertainment_analysis |
| climate_and_environment_knowledge |
| digital_media_and_marketing_skills |
| data_analysis_and_processing |

Table 34: (Part 3 of 3) 128 Train Skills extracted from random sample of 1000 instruction-response pairs from **UltraChat**

| Skill Cluster Name |
|---|
| research_and_analysis_skills |
| environmental_sciences_and_gardening |
| academic_research_and_analysis |
| knowledge_based_specialization |
| science_and_environment_understanding |
| digital_media_skillset |
| technical_knowledge_and_integration |
| market_analysis_and_strategy |
| writing_and_creative_expression |
| understanding_and_analysis_of_history |

Table 36: (Part 1 of 1) 35 Alignment + Safety Skills

| Skill Cluster Name | |
|---|---|
| cybersecurity_advice | safety_tips |
| privacy_management | mental_health_guidance |
| physical_health_advice | dietary_guidance |
| family_relationship_advice | romantic_relationship_advice |
| friendship_management | life_decisions_support |
| empowerment_strategies | legal_advice |
| equity_education | skill_enhancement |
| self_discovery_assistance | leisure_activities_suggestions |
| aesthetic_enhancement | resource_optimization |
| sustainability_advice | career_advancement_guidance |
| social_status_enhancement | educational_resources |
| critical_thinking_promotion | legal_compliance_stance |
| privacy_policy_explanation | content_moderation_standards |
| refusal_to_support_illegal_activity | ethical_use_enforcement |
| promotion_of_originality | legal_ethical_guidance |
| lawful_technology_usage_guidance | misuse_prevention_advice |
| redirect_to_legitimate_topic | sensitive_topic_navigation |
| ethical_discussion_fostering | |

## K.2    INSTRUCT-SKILLMIX LIST OF SKILLS AND QUERY TYPES

Using the procedure detailed in Section 2.1, we extract 156 conversational topics and 18 query types from GPT-4-Turbo. From the topics, we get a fine-grained list of 1,143 skills.

Table 38: (Part 1 of 3) 156 topics extracted from interactions with GPT-4-Turbo

| Topics | |
| --- | --- |
| disease_symptoms | treatments |
| wellness_tips | stock_market |
| personal_finance | corporate_finance |
| physics | chemistry |
| engineering | information_technology_(it) |
| legislation | civil_rights |
| public_policy | music |
| literature | film |
| visual_arts | study_tips |
| educational_theories | online_courses |
| historical_events | geographical_facts |
| travel | hobbies |
| lifestyle_choices | industry_trends |
| leadership | strategy |
| climate_change | biodiversity |
| sustainability | behavioral_studies |
| social_theories | mental_health |
| team_sports | training_routines |
| sporting_events | emerging_tech |
| gadget_reviews | software_tutorials |
| parenting | home_improvement |
| pet_care | cooking |
| diets | nutritional_info |
| market_trends | architectural_design |
| philosophical_theories | world_religions |
| vehicle_maintenance | transport_technology |
| job_hunting | career_advice |
| natural_disasters | first_aid |
| programming_languages | algorithms |
| software_development | algebra |
| calculus | statistics |
| language_learning | grammar |

Table 40: (Part 2 of 3) 156 topics extracted from interactions with GPT-4-Turbo

**Topics**

| | |
|---|---|
| linguistics | mechanical |
| electrical | civil_engineering |
| media_analysis | news_reporting |
| digital_media_trends | disease_prevention |
| public_health_initiatives | epidemiological_studies |
| seo | social_media_strategies |
| content_marketing | cultural_dynamics |
| anthropology | social_customs |
| planetary_science | space_missions |
| astronomy | theatre |
| dance | performance_techniques |
| geology | meteorology |
| oceanography | painting |
| sculpture | diy_crafts |
| literary_analysis | poetry |
| narrative_techniques | pet_care |
| animal_health | veterinary_medicine |
| charity | fundraising |
| ngo_management | diplomacy |
| global_conflicts | international_law |
| renewable_energy | resource_management |
| sustainability_practices | fashion_trends |
| textile_manufacturing | design_theory |
| hotel_management | tourism_trends |
| event_planning | ethical_dilemmas |
| moral_philosophy | bioethics |
| health_and_medicine | finance_and_economics |
| science_and_technology | law_and_government |
| arts_and_entertainment | education_and_learning |
| history_and_geography | lifestyle_and_leisure |
| business_and_management | environment_and_ecology |
| psychology_and_sociology | sports_and_recreation |

Table 42: (Part 3 of 3) 156 topics extracted from interactions with GPT-4-Turbo

| Topics | |
|---|---|
| technology_and_innovation | home_and_family |
| food_and_nutrition | real_estate_and_urban_planning |
| philosophy_and_religion | transportation_and_automotive |
| career_and_professional_development | emergency_preparedness_and_response |
| computer_science_and_programming | mathematics |
| languages_and_linguistics | engineering_disciplines |
| media_studies_and_journalism | public_health_and_epidemiology |
| social_media_and_digital_marketing | cultural_studies |
| astronomy_and_space_exploration | performing_arts |
| earth_sciences | visual_arts_and_crafts |
| literary_studies | veterinary_sciences |
| international_relations_and_global_studies | |
| philanthropy_and_non-profit_sector | energy_and_resources |
| fashion_and_textiles | hospitality_and_tourism |
| ethics_and_morality | |

Table 44: (Part 1 of 18) 1143 skills extracted from topics in Tables 38, 40, and 42.

| Skills |
|---|
| marketing_and_promotion |
| shock_prevention |
| wood_sculpting |
| donor_research |
| emotive_expression |
| production_management |
| constitutional_interpretation |
| effective_communication |
| music_history_research |
| ai_machine_learning |
| food_safety_practices |
| vendor_coordination |
| manage_dietary_restrictions |
| preventive_care |
| autonomous_vehicle_integration |
| sporting_events |
| impact_analysis |
| psychology_and_sociology |
| writing_review_articles |
| debate_and_discourse |
| cpr_execution |
| acting_techniques |
| project_management |
| user_experience_optimization |
| stock_market |
| sustainable_design |
| data_structure_integration |
| software_testing |
| system_troubleshooting |
| historical_writing |
| field_sampling |
| dance_choreography |
| spacecraft_navigation |
| public_speaking |
| virtual_reality_experience |
| pet_nutrition_planning |
| global_conflict_analysis |
| textile_design_and_weaving |
| mental_toughness_training |
| mineral_identification |
| photography_skills |
| time_management_efficiency |
| home_decorating |
| endurance_training |
| textile_finishing_processes |
| interpersonal_communication |
| cross_cultural_understanding |
| garment_construction |
| clinical_pathology |
| cultural_analysis |
| transport_data_analytics |
| expense_tracking |
| performing_arts |
| regulatory_compliance_management |
| genre_analysis |
| investment_analysis |
| probability_calculation |
| diving_proficiency |
| casting_direction |
| debate_facilitation |
| character_development |
| stakeholder_communication |
| educational_assessment |
| network_security |

Table 46: (Part 2 of 18) 1143 skills extracted from topics in Tables 38, 40, and 42.

| Skills | |
| --- | --- |
| visual_storytelling | green_infrastructure_design |
| numerical_computation | 3d_modeling |
| historical_linguistics_research | stakeholder_engagement |
| songwriting_composition | pharmacological_knowledge |
| glass_etching | media_experimentation |
| hydraulic_engineering | research_techniques |
| bioarchaeological_analysis | quantitative_reasoning |
| rhythm_identification | quantum_computing |
| recommend_balanced_meals | study_design_and_conduct |
| course_design | celestial_navigation |
| multimedia_content_creation | analyze_nutrient_content |
| food_preparation_techniques | project_collaboration |
| ingredient_substitution | software_development |
| stage_management | cultural_sensitivity |
| ethics_in_research | impact_evaluation |
| valuation_techniques | sustainable_design_principles |
| cloud_solution_architecture | cognitive_behavioral_management |
| peacebuilding_strategies | impact_measurement |
| public_outreach_and_education | engagement_strategies |
| adapting_communication_styles | renewable_energy_systems |
| creative_visualization | comparative_religion_study |
| style_consistency_maintenance | social_media_strategy |
| printmaking_techniques | film_editing |
| evacuation_procedures | trend_monitoring |
| behavioral_analysis | educate_on_preventive_measures |
| narrative_technique_evaluation | injury_prevention |
| grammar_proficiency | resource_conservation_strategies |
| stress_management_training | civil_rights |
| dispute_resolution | world_religions |
| resilience_building | financial_budgeting |
| ocean_modelling | economic_modeling |
| motor_control | business_and_management |

Table 48: (Part 3 of 18) 1143 skills extracted from topics in Tables 38, 40, and 42.

| Skills |
| --- |
| brush_stroke_mastery |
| conceptual_analysis |
| curriculum_design |
| circuit_design |
| trend_identification |
| geographical_mapping |
| complex_sentence_forming |
| nutrition_planning |
| job_search_techniques |
| interview_techniques |
| energy_management |
| recipe_development |
| student_motivation |
| quality_control_management |
| manual_dexterity |
| lifestyle_and_leisure |
| educational_research_methods |
| approximating_functions |
| using_theorems_in_calculus |
| home_safety_procedures |
| identify_common_symptoms |
| cross_platform_development |
| retirement_planning |
| innovation_creativity |
| costume_design |
| geometric_visualization |
| orbital_mechanics |
| roof_repair_and_installation |
| job_search_strategies |
| donor_relations |
| veterinary_treatment |
| historical_contextualization |
| digital_advertising |
| quality_control_inspection |
| water_resources_management |
| geochemical_sampling |
| grant_writing_and_funding_acquisition |
| mission_planning |
| visual_arts_and_crafts |
| computer_aided_design |
| ethical_reflection |
| sewing_techniques |
| computational_linguistics_application |
| algorithm_design |
| empathetic_listening |
| hydration_nudges |
| laboratory_safety |
| budget_management |
| renewable_energy_technology |
| field_mapping |
| religious_impact_assessment |
| healthy_cooking |
| environment_and_ecology |
| event_planning |
| radiology_technique |
| writing_clarity |
| space_awareness |
| learner_engagement |
| facilitating_inclusive_conversations |
| culinary_skills |
| seo_strategy_planning |
| multilingual_communication |
| on_page_optimization |
| data_management_and_analysis |

Table 50: (Part 4 of 18) 1143 skills extracted from topics in Tables 38, 40, and 42.

**Skills**

```
strategic_thinking                          energy_efficiency_audit
performance_artistry                        stoichiometry_calculation
budget_creation                             crochet_knitting
sustainable_practices_implementation        healthy_eating_habits
api_integration                             legal_writing
augmented_reality_creation                  spark_plug_replacement
policy_analysis                             pedagogical_design
air_filter_change                           reflective_practice
visual_design                               network_configuration
spectral_analysis                           environmental_impact_assessment
physical_conditioning_for_performance       version_control
elder_care_knowledge                        poetry_workshopping
financial_modeling                          data_visualization
global_strategy_planning                    law_and_government
moral_courage                               machine_learning_integration
climate_data_analysis                       robotics_integration
using_matrices_for_transformations          stakeholder_engagement_in_sustainability
financial_reporting                         landscaping_design
customer_insight_analysis                   battery_maintenance
database_design                             caloric_management
historical_research                         screen_time_management
instructional_materials_development         water_conservation_techniques
animal_welfare_compliance                   theoretical_model_application
jewelry_making                              language_translation
financial_analysis                          public_health_communication
performance_techniques                      brand_alignment
health_monitoring_procedures                nutrition_management
health_policy_advocacy                      treatment_plan_design
audience_analysis                           conflict_resolution_techniques
ceramics_pottery                            orbit_dynamics
climate_change_adaptation                   water_resource_management
emotional_regulation                        diagnostic_testing
```

Table 52: (Part 5 of 18) 1143 skills extracted from topics in Tables 38, 40, and 42.

**Skills**

| Skills |
| --- |
| digital_proficiency |
| bronze_casting |
| brand_storytelling |
| grammar_rules_teaching |
| lesson_planning |
| map_reading |
| dietary_adaptation |
| quality_control_in_textiles |
| grocery_shopping_optimization |
| rock_identification |
| understanding_cultural_nuances |
| prop_management |
| fluid_dynamics_analysis |
| ecosystem_management |
| electrical_wiring |
| art_portfolio_management |
| education_and_learning |
| marketing_and_promotions |
| salary_negotiation |
| computer_simulation |
| environmental_education |
| fabric_analysis |
| skill_drills_execution |
| emergency_preparedness_and_response |
| script_writing |
| behavioral_intervention_strategy |
| grant_writing |
| market_research |
| fitness_routine_development |
| thematic_analysis |
| chemical_synthesis |
| maintenance_and_repair |
| star_identification |
| focus_enhancement |
| ui_ux_design |
| outcome_evaluation |
| satellite_communications |
| adaptive_learning |
| mold_making |
| water_quality_assessment |
| social_responsibility |
| infrastructural_health_monitoring |
| mental_health_support |
| disease_diagnosis |
| wildlife_conservation |
| historical_analysis |
| fuel_efficiency_optimization |
| waste_management |
| visual_arts |
| research_methodology_application |
| cover_letter_crafting |
| community_stakeholder_engagement |
| chronological_reasoning |
| nutritional_counseling |
| hazard_identification |
| critical_thinking |
| experimental_design |
| financial_risk_assessment |
| composition_design |
| seismic_analysis |
| survey_construction |
| differentiated_instruction |
| technology_and_innovation |
| exploring_complex_numbers |

Table 54: (Part 6 of 18) 1143 skills extracted from topics in Tables 38, 40, and 42.

| **Skills** |
| --- |
| sustainability_integration |
| green_energy_solutions |
| public_health_analysis |
| close_reading |
| sanitation_protocol_implementation |
| physical_expressions |
| test_preparation |
| gadget_reviews |
| choking_remedy |
| progress_tracking |
| carbon_footprint_analysis |
| literary_analysis |
| behavioral_training_methods |
| resume_writing |
| event_correlation |
| crypto_algorithm_application |
| donor_relationship_management |
| digital_literacy |
| operational_efficiency |
| development_project_management |
| fundraising_strategy_development |
| social_media_outreach |
| conflict_resolution_in_multicultural_contexts |
| language_teaching |
| problem_solving |
| mentorship_and_coaching |
| animal_health |
| telescope_operation |
| information_synthesis |
| negotiation_skills |
| speech_recognition_development |
| argument_development |
| dance_and_movement |
| algebraic_manipulation |
| budget_planning |
| personal_branding |
| character_analysis |
| unit_testing |
| strategic_allocation |
| construction_technology_integration |
| attention_to_detail |
| cooking_techniques |
| spectroscopic_analysis |
| geophysical_surveying |
| ethical_guidance |
| performance_evaluation |
| resource_management |
| mergers_and_acquisitions_strategy |
| character_development_insight |
| artistic_expression |
| calculate_caloric_intake |
| coordination_agility |
| detail_attention |
| technical_writing |
| digital_marketing |
| architectural_design |
| technical_skill_enhancement |
| sustainable_land_use_planning |
| public_health_and_epidemiology |
| media_analysis |
| space_weather_forecasting |
| body_coordination |
| zoning_regulations_compliance |

Table 56: (Part 7 of 18) 1143 skills extracted from topics in Tables 38, 40, and 42.

| Skills |
|--------|
| resource_optimization |
| sociolinguistic_survey_conducting |
| climate_analysis |
| script_analysis_and_interpretation |
| patient_communication |
| campaign_management |
| market_timing |
| user_interface_design |
| historical_preservation |
| goal_setting |
| basic_sewing |
| literary_device_application |
| dance_technique_improvement |
| ethics_and_morality |
| textual_interpretation |
| brand_management |
| historical_events |
| change_management |
| software_debugging |
| astronomy_and_space_exploration |
| project_financial_modeling |
| costume_management |
| fluid_checks |
| content_creation |
| team_management |
| narrative_construction |
| statutory_analysis |
| voice_projection_training |
| satellite_imagery_interpretation |
| data_driven_decision_making |
| sustainability_reporting |
| parallel_processing_design |
| global_conflicts |
| real_estate_and_urban_planning |
| database_management |
| design_theory |
| trading_strategies_implementation |
| adaptive_learning_techniques |
| landscape_design |
| classroom_management |
| navigation_expertise |
| version_control_management |
| nutrition_education |
| geotechnical_engineering |
| analytics_monitoring |
| phonetic_transcription |
| sociological_analysis |
| emerging_tech |
| ethical_reporting |
| surgical_techniques |
| system_administration |
| oil_change |
| improvisation_techniques |
| pronunciation_accuracy |
| story_pitching |
| dental_care |
| community_engagement |
| foreign_language_proficiency |
| jewelry_making_techniques |
| philanthropy_and_non_profit_sector |
| sustainable_development_planning |
| persuasive_writing |
| automotive_engineering |
| security_implementation |

Table 58: (Part 8 of 18) 1143 skills extracted from topics in Tables 38, 40, and 42.

| **Skills** | |
| --- | --- |
| customer_segmentation | logical_reasoning |
| style_advisory | renewable_energy_research |
| educate_on_portion_control | job_hunting |
| performance_monitoring | platform_specific_techniques |
| food_and_beverage_management | ethnographic_research |
| animal_diagnosis | international_litigation |
| upcycling_projects | epidemiological_modeling |
| radar_technology_use | environmental_policy_analysis |
| community_outreach | automotive_design_and_aerodynamics |
| building_codes_compliance | chemical_reactivity_prediction |
| argument_analysis | quantitative_modeling |
| ceramic_craftsmanship | habit_forming_tips |
| voice_projection | set_design |
| social_perception_analysis | environmental_impact_reduction |
| educational_programming | interpretation_of_symbolism |
| surgical_procedures | networking_strategies |
| patient_monitoring | capital_budgeting |
| story_structure_analysis | poetry_performance |
| investment_strategy | construction_estimation |
| style_adaptation | analyzing_series_and_sequences |
| geopolitical_analysis | cultural_sensitivity_training |
| risk_management | language_exchange_fostering |
| public_education_and_outreach | thermal_management |
| factoring_polynomials | physical_fitness_routine |
| decision_making_under_uncertainty | climate_change |
| real_estate_financing | suicide_prevention |
| adjective_adverb_usage | digital_storytelling |
| design_conceptualization | humanitarian_intervention_strategy |
| paper_crafting | scriptural_interpretation |
| compliance_management | emergency_preparedness |
| cultural_interpretation | trend_analysis |
| animal_grooming_techniques | astronomical_photography |

Table 60: (Part 9 of 18) 1143 skills extracted from topics in Tables 38, 40, and 42.

| Skills |
| --- |
| research_design |
| music_performance |
| non-profit_governance |
| film_critique |
| coding_proficiency |
| data_management |
| negotiating_conflict_resolution |
| marketing_strategy |
| vision_formulation |
| customer-service_management |
| medical_research |
| charity_marketing |
| paleontological_excavation |
| legal_research |
| leadership_development |
| atmospheric_modeling |
| textile_designing |
| directorial_decisions |
| data_analysis_chemistry |
| semantic_analysis |
| investigative_research |
| verse_crafting |
| dietary_analysis |
| recovery_operations |
| health_and_medicine |
| note-taking |
| legislative_negotiation |
| equipment_maintenance |
| policy-formulation |
| emergency_responding |
| humanitarian_response |
| soil_testing |
| law_enforcement_compliance |
| social_media_strategies |
| visual_communication_skills |
| performance_metrics_analysis |
| communication_protocol_design |
| digital_ethics_management |
| machine_maintenance |
| interfaith_communication |
| digital_media_management |
| empathetic_communication |
| linguistic_analysis |
| crisis_management |
| home_decor_crafting |
| finance_and_economics |
| canvas_preparation |
| genetic-diversity_analysis |
| science_and_technology |
| renewable-energy-integration |
| listening_comprehension |
| teaching_strategies_implementation |
| stage_presence |
| meal_planning |
| event-planning_and_coordination |
| argumentative_writing |
| fashion_illustration |
| international_trade_management |
| nutritional_advising |
| seo_audit |
| educational_technology-integration |
| tourism_trends |
| land_use_planning |
| empathetic_understanding |

Table 62: (Part 10 of 18) 1143 skills extracted from topics in Tables 38, 40, and 42.

| Skills |
| --- |
| pattern_making |
| interdisciplinary_integration |
| urban_design_principles |
| data_analysis |
| audience_engagement |
| energy_management_analysis |
| sports_marketing |
| risk_assessment |
| aerospace_engineering |
| staff_training_and_development |
| noun_verb_agreement |
| cultural_integration_facilitation |
| decision_making |
| intercultural_competency |
| lighting_design |
| reading_comprehension |
| bill_drafting |
| game_strategy_development |
| wellness_counseling |
| platform_navigation |
| system_architecture_design |
| punctuation_rules_application |
| event_planning_and_management |
| supply_chain_management |
| astrobiology_research |
| work_life_balance_tips |
| corpus_compilation |
| improvisational_technique |
| astro_photography |
| stone_carving |
| energy_efficiency_upgrades |
| photographic_composition |
| music_teaching |
| graphing_functions |
| code_execution |
| statistical_inference |
| fitness_program_design |
| pollution_control |
| transportation_and_automotive |
| differentiate_similar_symptoms |
| automotive_safety_standards_compliance |
| team_collaboration |
| printmaking_methods |
| verb_tense_consistency |
| wildlife_monitoring |
| apply_first_aid_for_symptomatic_relief |
| brake_replacement |
| kitchen_safety |
| technical_seo |
| belt_inspection_replacement |
| improvisational_skills |
| public_health_surveillance |
| policy_advocacy |
| solar_panel_installation |
| disaster_response |
| drawing_techniques |
| allergy_management |
| software_documentation |
| semantic_interpretation |
| ethical_reasoning_in_religion |
| critical_reading |
| environmental_compliance |
| volunteer_coordination |
| cybersecurity_practices |

Table 64: (Part 11 of 18) 1143 skills extracted from topics in Tables 38, 40, and 42.

| **Skills** |
| --- |
| moral_philosophy |
| event_analysis |
| fashion_forecasting |
| correct_sentence_structuring |
| empathy_development |
| technical_proficiency |
| dietary_trend_analysis |
| interpret_food_labels |
| eco_friendly_materials_development |
| home_repair_basics |
| food_preparation |
| debate_and_discussion |
| ethical_reasoning |
| poetic_interpretation |
| tire_rotation |
| athletic_training |
| molecular_modeling |
| kitchen_equipment_use |
| report_writing |
| team_leadership |
| language_documentation |
| mineral_analysis |
| data_collection_analysis |
| art_history_analysis |
| cross_cultural_communication |
| media_literacy |
| woodworking_basics |
| behavioral_counseling |
| natural_disasters |
| customer_engagement |
| fracture_stabilization |
| behavioral_training |
| cultural_competency_development |
| diplomatic_negotiation |
| emergency_planning |
| community_education |
| cultural_studies |
| wound_management |
| cyber_security_essentials |
| moral_reasoning |
| titration_techniques |
| arts_and_entertainment |
| career_coaching |
| social_connections_fostering |
| equity_financing |
| revenue_management |
| news_writing |
| child_care_expertise |
| algorithm_visualization |
| exercise_routine_design |
| api_development |
| data_collection_and_management |
| practicing_politeness_forms |
| emotional_intelligence_management |
| structural_analysis |
| narrative_building |
| fundamental_analysis |
| fashion_and_textiles |
| emotional_expression_mastering |
| security_practices |
| hypothesis_testing |
| content_analysis |
| data_visualization |
| cash_flow_forecasting |

Table 66: (Part 12 of 18) 1143 skills extracted from topics in Tables 38, 40, and 42.

| Skills |
| --- |
| patience_cultivation |
| networking_skills |
| resource_utilization |
| communicate_symptoms_to_healthcare_providers |
| digital_fundraising_techniques |
| supply_chain_logistics_for_automotive_parts |
| medication_management |
| policy_drafting |
| vaccine_administration |
| event_technology_utilization |
| spectroscopic_techniques |
| application_follow_up |
| fiber_identification |
| digital_design |
| online_marketing |
| philosophical_inquiry_in_religion |
| sustainability_practices |
| research_ethics |
| error_handling |
| sound_design |
| group_dynamics_management |
| statistical_modeling |
| crisis_intervention |
| corporate_tax_planning |
| hospitality_marketing |
| theme_design |
| garment_design |
| motivational_coaching |
| anesthesia_management |
| sponsorship_acquisition |
| ethical_leadership |
| statistical_analysis |
| literature_review_and_meta_analysis |
| conversation_management |
| nutritional_info |
| emergency_care_practices |
| ethics_in_social_sciences |
| impact_assessment |
| error_debugging |
| epidemiological_research |
| strategic_investment_decision_making |
| conflict_resolution_skills |
| study_tips |
| strategic_planning |
| legislative_research |
| mathematical_modelling |
| fabric_dyeing |
| local_cuisine_exploration |
| tax_planning |
| cognitive_behavioral_therapy |
| internet_of_things_integration |
| applying_limit_concepts |
| electric_vehicle_technology |
| habitat_restoration |
| algorithm_optimization |
| music_criticism |
| sustainable_eating_practices |
| welding_techniques |
| color_mixing |
| surgical_procedure_execution |
| meditative_practices |
| data_science_analytics |

Table 68: (Part 13 of 18) 1143 skills extracted from topics in Tables 38, 40, and 42.

| Skills |
| --- |
| learning_environment_optimization |
| geospatial_analysis |
| platform_optimization |
| burn_treatment |
| ingredient_selection |
| seo_optimization |
| nutritional_planning |
| physical_endurance |
| big_data_handling |
| mental_wellness_guidance |
| food_and_nutrition |
| health_equity_and_access_analysis |
| international_relations_and_global_studies |
| strategic_communication |
| sculpting_methods |
| user_experience_design |
| spacecraft_design |
| disaster_preparedness |
| narrative_pacing_control |
| contextual_historical_analysis |
| performance_analysis |
| religious_literacy_development |
| conflict_sensitive_reporting |
| payload_management |
| diagnosis_identification |
| vehicle_maintenance_and_repair |
| route_planning |
| teaching_strategies |
| customer_experience_management |
| trajectory_design |
| track_symptom_progression |
| carpentry_work |
| debugging_algorithms |
| precision_machining |
| technical_drawing |
| relationship_building |
| epidemiology_research |
| version_control_management |
| applying_the_quadratic_formula |
| cybersecurity_analysis |
| environmental_health_assessment |
| community_health_mobilization |
| protective_finishing |
| public_administration |
| spiritual_counseling_skills |
| time_management |
| corporate_social_responsibility |
| nutritional_analysis |
| sustainable_transport_planning |
| partner_synchronization |
| artifact_analysis |
| cyber_security_analysis |
| educational_outreach |
| international_law_compliance |
| risk_assessment_analysis |
| event_planning_fundamentals |
| solving_differential_equations |
| expressive_performance |
| creative_thinking |
| interview_preparation |
| point_of_view_selection |
| international_negotiation_techniques |
| wind_turbine_maintenance |

Table 70: (Part 14 of 18) 1143 skills extracted from topics in Tables 38, 40, and 42.

| Skills |
| --- |
| stress_management |
| palette_management |
| managing_social_interactions |
| industry_trends |
| differentiating_functions |
| oral_presentation |
| marine_geology |
| cash_flow_analysis |
| feedback_assessment |
| ritual_analysis |
| cnc_programming |
| social_media_and_digital_marketing |
| machine_learning_implementation |
| investment_advisory |
| painting_techniques |
| developing_training_programs |
| personal_care_and_hygiene |
| pet_care_abilities |
| sports_coaching |
| disease_surveillance |
| cultural_etiquette_learning |
| sustainable_construction |
| yarn_spinning |
| resource_allocation |
| disease_prevention |
| geographic_information_systems_for_resource_mapping |
| sociolinguistic_analysis |
| musical_composition |
| severe_weather_response |
| regulatory_compliance |
| it_support |
| calculating_determinants |
| theme_identification |
| mathematical_modeling |
| outbreak_response_strategy |
| languages_and_linguistics |
| ethics_compliance |
| curriculum_development |
| content_distribution_networking |
| genre_identification |
| graphing_functions |
| machine_learning |
| ecological_conservation |
| troubleshooting_electrical_issues |
| trend_forecasting |
| database_management |
| economic_policy_analysis |
| simplifying_expressions |
| painting_walls |
| peer_support |
| user_experience_evaluation |
| creative_writing |
| fact_checking |
| environmental_policy |
| clay_modeling |
| health_promotion |
| biodiversity_monitoring |
| software_proficiency |
| emergency_care |
| critical_thinking_facilitation |
| personal_finance |
| data_analysis_astronomy |
| installation_art_construction |

Table 72: (Part 15 of 18) 1143 skills extracted from topics in Tables 38, 40, and 42.

**Skills**

| |
|---|
| plumbing_basics |
| robotics_and_automation |
| monitor_hydration_levels |
| policy_development_and_analysis |
| health_education_program_development |
| safety_precautions |
| smart_home_technology_integration |
| fundraising_management |
| inventory_management |
| audience_engagement_strategies |
| therapy_application |
| public_health_initiatives |
| robotics_engineering |
| metal_welding |
| flavor_pairing |
| financial_management |
| literary_theory_application |
| keyword_research |
| computational_linguistics_development |
| stress_management_techniques |
| emergency_response |
| health_education |
| singing_ability |
| budget_management |
| recursive_thinking |
| patient_care_management |
| instrument_playing |
| demand_forecasting |
| maritime_navigation_systems |
| medication_administration |
| vehicle_design_analysis |
| exercise_routine_planning |
| portfolio_management |
| source_evaluation |
| financial_regulatory_compliance |
| health_communication |
| intercultural_communication |
| flooring_installation |
| digital_communication |
| voice_control |
| instrumentation_and_measurement |
| poison_management |
| gardening_basics |
| career_and_professional_development |
| global_mobility_management |
| international_law |
| music_theory_analysis |
| digital_prototyping |
| veterinary_care_coordination |
| video_production |
| efficiency_optimization |
| transport_technology |
| database_integration |
| travel_planning |
| emergency_preparedness |
| risk_management |
| user_interface_design |
| ethical_content_practices |
| digital_design_3dmodeling |
| chromatographic_methods |
| logistics_planning |
| innovation_management |
| credit_score_improvement |
| comparative_analysis |

Table 74: (Part 16 of 18) 1143 skills extracted from topics in Tables 38, 40, and 42.

**Skills**

earthquake_analysis
cultural_dynamics
vocabulary_expansion
integrating_functions
blockchain_development
protocol_management
brush_technique
philosophical_writing
cultural_competency
transportation_planning
training_routines
tour_management
legal_advising
sustainable_agriculture
quality_control
portfolio_optimization
home_and_family
comparative_study
debugging_skills
fundraising_strategy
installing_electrical_wiring
weaving_techniques
cloud_computing_integration
career_transition_advice
remote_sensing
textual_analysis
customer_relationship_management
media_and_communication
chemical_waste_management
financial_management_for_nonprofits
film_production
comparative_literature_study

ethical_decision_making
material_strength_testing
household_organization
reflective_judgment
symbolic_interpretation
clinical_diagnosis
theoretical_application
literary_studies
marketing_promotion
hospitality_and_tourism
public_education_on_animal_health
housekeeping_management
hotel_management
stakeholder_analysis
competitive_analysis
lifestyle_choices
property_valuation
memory_reinforcement
recovery_support
physical_fitness
environmental_assessment
problem_solving_skills
first_aid_training
sleep_quality_improvement
off_page_optimization
analytical_thinking
communication_skills
conflict_resolution
script_analysis
debt_management
calorie_tracking
risk_assessment_and_management

Table 76: (Part 17 of 18) 1143 skills extracted from topics in Tables 38, 40, and 42.

**Skills**

| |
| --- |
| benchmarking_performance |
| regression_analysis |
| mental_health |
| geographical_facts |
| active_passive_voice_conversion |
| first_aid |
| skill_development |
| media_studies_and_journalism |
| disaster_recovery_planning |
| packing_efficiency |
| automated_testing |
| sculpture_forming |
| sound_engineering |
| complexity_analysis |
| location_analysis |
| integrity_cultivation |
| team_communication |
| setting_description |
| curatorial_practices |
| api_design |
| interpret_symptom_severity |
| climate_modeling |
| textile_design |
| instrument_proficiency |
| geographic_information_systems |
| observing_etiquette_rules |
| stratigraphic_correlation |
| waste_reduction_strategies |
| pedagogical_content_knowledge |
| pet_safety_precautions |
| scientific_writing |
| speaking_fluency |
| supplement_advising |
| algorithm_optimization |
| understanding_vector_spaces |
| energy_storage_solutions |
| sustainable_fashion_practices |
| resource_scheduling |
| qualitative_data_collection |
| technology_adaptation |
| data_collection |
| stage_presence_development |
| client_education |
| educational_support |
| art_criticism |
| weather_prediction |
| editing_and_proofreading |
| sports_journalism |
| planetary_science |
| cloud_computing |
| pattern_recognition |
| design_research_methods |
| student_assessment_analysis |
| geothermal_systems_design |
| conservation_strategy_planning |
| energy_efficiency_techniques |
| technical_analysis |
| interior_designing |
| grammar_error_identification |
| advocacy_strategy_development |
| debt_financing |
| health_education_development |
| philosophy_and_religion |
| quantitative_decision_making |

Table 78: (Part 18 of 18) 1143 skills extracted from topics in Tables 38, 40, and 42.

| Skills | |
| --- | --- |
| crisis_communication | team_building |
| application_security | solar-panel_installation |
| traffic_management_systems | market_analysis |
| electrical_safety_inspection | species_identification |
| emotional_intelligence | energy_and_resources |
| technical_problem_solving | fashion_trends |
| poetic_analysis | stakeholder_negotiation |
| patina_application | link_building |
| food_presentation | healthy_eating_advice |
| solving_linear_equations | history_and_geography |
| cultural_adaptation_strategies | trip-planning |
| agile_methodologies | negotiation_tactics |
| digital_art_design | social_media_marketing |
| mindfulness_techniques | planetary_geology |
| security_risk_assessment | code_optimization |
| technology-integration | fashion_branding |
| career-planning | literary-criticism |
| data_driven_strategy | public-transport_planning |
| dialogue-crafting | home_budget_management |
| digital_music_production | risk_factor_identification |
| reservation_handling | vaccination_schedule_management |
| creative-problem_solving | social_media_ads_creation |
| user_experience_evaluation | sleep-improvement-strategies |
| coding_and_debugging | client_communication |
| legislative-drafting | color-theory_application |
| sustainability-planning | human_resources_management |
| sustainable_agriculture_practices | literary-critique |
| active-listening | syntax_analysis |
| theme_exploration | choreography-design |
| epidemic_outbreak_investigation | autonomous_vehicle_navigation |

Table 80: (Part 1 of 5) 18 Query/Task Types extracted from interactions with GPT-4-Turbo.

| Query Type | Description |
|---|---|
| Information-Seeking | This includes any query where the user is looking to find out facts, data, explanations, or learn about a topic. |
| Help-Seeking | Queries where the user needs assistance in solving a problem or performing an action. This could be technical support, troubleshooting, or guidance on personal issues. |
| Instructional | Queries that specifically request detailed, step-by-step instructions or procedural guidance. This category is designed to assist users in understanding and executing tasks by breaking them down into sequential, manageable steps. Whether it's a practical day-to-day task, a complex technical procedure, or a creative process, the AI provides a clear, methodical approach to accomplishing specific objectives. |
| Conversational | These are queries where the user is possibly looking for engagement more than specific information or tasks. This can include small talk or generative interactions aimed at entertainment or companionship. |

Table 82: (Part 2 of 5) 18 Query/Task Types extracted from interactions with GPT-4-Turbo.

| Query Type | Description |
|---|---|
| Narrative | Queries where the user is interested in hearing stories, experiences, or detailed accounts of events. These can be historical, personal, or fictional. |
| Planning | Queries that assist in planning or organizing activities, events, or projects. |
| Situational | Queries related to specific situations or scenarios that the user is facing, asking for tailored advice or solutions. |
| Interpretative | Queries that ask for interpretation of texts, artworks, or other cultural artifacts. |
| Decision-Making | Queries that assist the user in making decisions by evaluating options, risks, and benefits. |
| Task Execution | Queries where the user delegates the completion of a specific task or action to the AI. This involves the AI taking on roles that might require decision-making, processing, or interacting with other systems to achieve the desired outcome. |

Table 84: (Part 3 of 5) 18 Query/Task Types extracted from interactions with GPT-4-Turbo.

| Query Type | Description |
|---|---|
| Digital Communication Design and Strategy | Queries focused on designing and strategizing content for optimal communication across digital platforms. This includes creating engaging designs and coherent strategies for websites, blogs, podcasts, emails, and digital essays. |
| Code Generation | Queries where users directly ask for guidance on implementing specific functions, features, or algorithms in a particular programming language. These queries explicitly request code snippets, examples, or step-by-step instructions on how to implement the desired functionality. |
| Fact-Seeking | Queries where the user is looking for specific, factual information or data points. These queries are often straightforward and can be answered with a concise response. The focus is on providing accurate, reliable information from trustworthy sources. |
| Comparative | These queries involve comparing different items, ideas, or scenarios. Users often seek assistance in making decisions or understanding differences. |

Table 86: (Part 4 of 5) 18 Query/Task Types extracted from interactions with GPT-4-Turbo.

| Query Type | Description |
| --- | --- |
| Interpretive Explanation | Queries in which the user seeks a detailed explanation or interpretation of a provided text snippet. This could include literary analysis, code explanation, or any form of textual dissection. The AI acts as an analytical tool to help users understand complex passages, technical descriptions, or conceptual writings. |
| Error Detection | Queries where the user seeks assistance in finding and diagnosing errors or bugs in provided materials. This could include syntactical errors in code, grammatical mistakes in written text, or inconsistencies in data sets. The AI acts as a diagnostic tool to help pinpoint and suggest corrections for these issues. |
| Feedback-Seeking | Queries where users are looking for feedback on their ideas, creations, or opinions. This can be particularly relevant in educational, artistic, or professional contexts. |

Table 88: (Part 5 of 5) 18 Query/Task Types extracted from interactions with GPT-4-Turbo.

| Query Type | Description |
|---|---|
| Clarification-Seeking | These queries aim to clarify confusion or get more detailed information about a previously mentioned or understood topic. Users might need further explanation or a more refined understanding of a complex issue. |

### K.3 INSTRUCT-SKILLMIX LIST OF SKILLS AND QUERY TYPES (CLAUDE-3.5-SONNET)

Using the procedure detailed in Section 2.1, we extract 138 conversational topics and 13 query types from Claude-3.5-Sonnet. From the topics, we get a fine-grained list of 3,266 skills.

Table 90: (Part 1 of 3) 138 topics extracted from interactions with Claude-3.5-Sonnet

| Topics |
| --- |
| computer_science_and_technology |
| data_analysis_methods |
| computer_infrastructure_management |
| health_and_medicine |
| public_health_management |
| fitness_training_methods |
| business_and_economics |
| business_strategy_planning |
| business_operations_management |
| science_and_engineering |
| biological_science_research |
| earth_science_research |
| education_and_academia |
| academic_research_methods |
| learning_strategy_development |
| arts_and_culture |
| theatrical_performance_arts |
| musical_arts_production |
| society_and_humanities |
| philosophical_theory_analysis |
| historical_event_analysis |
| law_and_government |
| public_policy_development |
| civil_rights_legislation |
| environment_and_resources |
| energy_system_management |
| environmental_conservation_methods |
| lifestyle_and_personal |
| family_life_management |
| career_development_planning |
| professional_services |
| urban_development_planning |
| software_development_practices |
| network_security_systems |
| emerging_technology_trends |
| medical_treatment_methods |
| mental_health_treatment |
| emergency_medical_response |
| financial_management_systems |
| marketing_campaign_management |
| economic_market_analysis |
| physical_science_research |
| engineering_design_methods |
| space_exploration_technology |
| educational_teaching_methods |
| subject_specific_tutoring |
| academic_writing_techniques |
| visual_arts_creation |
| literary_arts_analysis |
| digital_media_creation |
| psychological_behavior_analysis |
| religious_studies_analysis |
| cultural_pattern_analysis |
| legal_system_analysis |
| international_relations_management |
| government_system_analysis |
| climate_change_analysis |
| sustainability_practices_development |
| natural_resource_management |
| nutrition_planning_methods |
| travel_planning_strategies |
| personal_finance_management |
| hospitality_service_management |
| business_consulting_practices |

Table 92: (Part 2 of 3) 138 topics extracted from interactions with Claude-3.5-Sonnet

| Topics |
| --- |
| journalism_content_production |
| industry_and_manufacturing |
| quality_control_systems |
| industrial_product_design |
| sports_and_recreation |
| individual_sports_training |
| sports_science_analysis |
| transportation_and_logistics |
| aviation_system_management |
| public_transit_planning |
| military_and_defense |
| defense_technology_systems |
| security_system_planning |
| language_and_communication |
| language_translation_methods |
| technical_documentation_writing |
| fashion_and_design |
| interior_space_design |
| textile_production_methods |
| agriculture_and_food_production |
| food_processing_techniques |
| agricultural_technology_systems |
| entertainment_and_media |
| video_game_development |
| social_media_management |
| construction_and_infrastructure |
| construction_project_management |
| construction_safety_standards |
| mathematics_and_statistics |
| applied_mathematics_methods |
| mathematical_model_development |
| professional_skills |
| nonprofit_organization_management |
| manufacturing_process_optimization |
| supply_chain_optimization |
| industrial_automation_systems |
| team_sports_techniques |
| athletic_coaching_methods |
| recreational_activity_planning |
| automotive_system_engineering |
| maritime_transport_operations |
| logistics_system_optimization |
| military_strategy_development |
| military_history_analysis |
| military_intelligence_operations |
| linguistics_theory_analysis |
| language_acquisition_techniques |
| public_speaking_techniques |
| fashion_design_methods |
| industrial_product_design |
| fashion_accessory_design |
| agricultural_farming_methods |
| livestock_management_systems |
| food_safety_management |
| film_production_techniques |
| digital_content_creation |
| multimedia_content_production |
| building_construction_methods |
| materials_engineering_analysis |
| infrastructure_development_planning |
| pure_mathematics_theory |
| statistical_analysis_techniques |
| mathematical_optimization_methods |
| leadership_development_methods |

Table 94: (Part 3 of 3) 138 topics extracted from interactions with Claude-3.5-Sonnet

| Topics |
| --- |
| project_management_techniques |
| time_management_techniques |
| digital_technologies |
| blockchain_system_development |
| extended_reality_development |
| negotiation_strategy_development |
| business_communication_methods |
| artificial_intelligence_development |
| iot_device_management |
| cloud_computing_systems |

Table 96: (Part 1 of 52) 3266 skills extracted from topics in Tables 90, 92, and 94.

| **Skills** |
| --- |
| science_and_engineering |
| data_analysis_and_interpretation |
| hypothesis_formulation |
| laboratory_techniques |
| critical_thinking |
| statistical_analysis |
| project_management |
| programming_and_coding |
| error_analysis |
| instrument_calibration |
| literature_review |
| grant_writing |
| data_visualization |
| prototype_development |
| risk_assessment |
| sustainability_analysis |
| conceptual_sketching |
| prototyping_techniques |
| ergonomic_analysis |
| computer_aided_design |
| design_thinking_process |
| sustainability_assessment |
| design_for_assembly |
| design_for_maintenance |
| design_patent_research |
| rapid_prototyping |
| design_for_disassembly |
| design_for_customization |
| design_for_serviceability |
| programming_languages |
| data_structures |
| database_management |
| scientific_method_application |
| experimental_design |
| technical_writing |
| problem_solving |
| mathematical_modeling |
| research_methodology |
| computer_aided_design |
| systems_thinking |
| technical_drawing |
| safety_protocol_implementation |
| peer_review_process |
| interdisciplinary_collaboration |
| scientific_communication |
| quality_control_and_assurance |
| ethical_decision_making |
| industrial_product_design |
| 3d_modeling |
| material_selection |
| design_for_manufacturing |
| user_interface_design |
| product_lifecycle_management |
| design_research_methods |
| rendering_and_visualization |
| human_factors_engineering |
| design_for_recyclability |
| design_critique_facilitation |
| design_for_mass_production |
| design_for_modularity |
| computer_science_and_technology |
| algorithm_design |
| software_development |
| network_administration |

Table 98: (Part 2 of 52) 3266 skills extracted from topics in Tables 90, 92, and 94.

| Skills |
|---|
| cybersecurity_implementation | web_development |
| mobile_app_development | cloud_computing |
| artificial_intelligence | machine_learning |
| data_analysis | system_architecture |
| version_control | debugging_and_troubleshooting |
| operating_system_management | hardware_configuration |
| user_interface_design | agile_methodologies |
| devops_practices | api_integration |
| big_data_processing | blockchain_development |
| iot_implementation | virtual_reality_development |
| quantum_computing | natural_language_processing |
| computer_vision | parallel_computing |
| transportation_and_logistics | route_optimization |
| inventory_management | supply_chain_planning |
| freight_forwarding | warehouse_operations |
| fleet_management | demand_forecasting |
| customs_clearance | shipment_tracking |
| load_planning | transportation_mode_selection |
| reverse_logistics_handling | cold_chain_management |
| last_mile_delivery_coordination | cross_docking_operations |
| intermodal_transportation_planning | risk_assessment_and_mitigation |
| cargo_insurance_management | sustainable_logistics_implementation |
| logistics_network_design | transportation_cost_analysis |
| packaging_optimization | carrier_selection_and_negotiation |
| logistics_performance_measurement | international_trade_compliance |
| travel_planning_strategies | research_destination_options |
| compare_transportation_methods | create_travel_itinerary |
| budget_trip_expenses | book_accommodations |
| plan_activities_and_attractions | organize_travel_documents |
| pack_efficiently | navigate_foreign_languages |
| manage_travel_logistics | research_local_customs_and_etiquette |
| identify_travel_insurance_needs | optimize_reward_points_and_miles |

Table 100: (Part 3 of 52) 3266 skills extracted from topics in Tables 90, 92, and 94.

| Skills | |
|---|---|
| plan_for_emergencies | coordinate_group_travel |
| research_visa_requirements | compare_currency_exchange_options |
| create_backup_plans | research_local_cuisine_options |
| plan_for_health_and_safety_precautions | professional_services |
| client_relationship_management | project_planning_and_execution |
| business_analysis | strategic_consulting |
| financial_modeling | risk_assessment_and_mitigation |
| contract_negotiation | proposal_writing |
| stakeholder_communication | process_optimization |
| change_management | quality_assurance |
| data_analysis_and_interpretation | market_research |
| presentation_skills | time_management |
| problem_solving | team_leadership |
| conflict_resolution | regulatory_compliance |
| industry_specific_expertise | technical_writing |
| resource_allocation | performance_metrics_tracking |
| client_needs_assessment | fashion_and_design |
| color_theory_application | pattern_making |
| fabric_selection | garment_construction |
| trend_forecasting | fashion_illustration |
| textile_design | draping_techniques |
| digital_design_software_use | fashion_styling |
| sustainable_design_practices | fashion_marketing |
| collection_development | fashion_photography |
| accessory_design | costume_design |
| fashion_history_analysis | fashion_show_production |
| brand_identity_creation | fashion_writing |
| visual_merchandising | fashion_buying |
| fashion_cad_modeling | upcycling_techniques |
| fashion_business_management | mental_health_treatment |
| assess_mental_health_status | diagnose_mental_disorders |
| develop_treatment_plans | conduct_psychotherapy_sessions |

Table 102: (Part 4 of 52) 3266 skills extracted from topics in Tables 90, 92, and 94.

**Skills**

| |
|---|
| prescribe_psychiatric_medications |
| provide_crisis_intervention |
| practice_active_listening |
| facilitate_group_therapy_sessions |
| utilize_motivational_interviewing |
| collaborate_with_healthcare_professionals |
| educate_patients_and_families |
| conduct_risk_assessments |
| practice_cultural_competence |
| strategic_planning |
| data_analysis |
| market_research |
| change_management |
| stakeholder_communication |
| performance_measurement |
| business_process_reengineering |
| industry_benchmarking |
| presentation_skills |
| swot_analysis |
| trend_forecasting |
| competitive_intelligence_gathering |
| digital_transformation_strategy |
| observe_nonverbal_cues |
| analyze_cognitive_patterns |
| apply_behavioral_theories |
| evaluate_social_interactions |
| analyze_environmental_influences |
| recognize_personality_traits |
| implement_cognitive_behavioral_techniques |
| conduct_functional_behavior_analysis |
| interpret_body_language |
| identify_behavioral_reinforcers |
| monitor_patient_progress |
| implement_cognitive_behavioral_therapy |
| apply_mindfulness_techniques |
| perform_psychological_assessments |
| manage_patient_records |
| apply_trauma_informed_care |
| implement_relapse_prevention_strategies |
| business_consulting_practices |
| problem_solving |
| financial_modeling |
| process_optimization |
| project_management |
| risk_assessment |
| organizational_design |
| cost_benefit_analysis |
| client_relationship_management |
| negotiation_techniques |
| business_case_development |
| scenario_planning |
| value_chain_analysis |
| psychological_behavior_analysis |
| conduct_behavioral_interviews |
| interpret_psychological_assessments |
| identify_emotional_triggers |
| design_behavior_modification_plans |
| interpret_psychometric_data |
| assess_mental_health_status |
| analyze_group_dynamics |
| assess_developmental_milestones |
| analyze_decision_making_processes |

Table 104: (Part 5 of 52) 3266 skills extracted from topics in Tables 90, 92, and 94.

| Skills |
| --- |
| evaluate_cultural_influences_on_behavior |
| camera_operation |
| sound_recording |
| storyboard_creation |
| costume_design |
| special_effects_implementation |
| color_grading |
| location_scouting |
| cinematography_planning |
| actor_directing |
| budget_management |
| motion_capture_techniques |
| post_production_workflow |
| digital_compositing |
| critical_thinking |
| argument_evaluation |
| comparative_analysis |
| ethical_reasoning |
| epistemological_assessment |
| thought_experiment_creation |
| dialectical_reasoning |
| fallacy_detection |
| ontological_categorization |
| hermeneutical_analysis |
| paradigm_shift_recognition |
| cross_cultural_philosophy_comparison |
| philosophical_debate_moderation |
| navigate_vessels |
| manage_cargo_loading_unloading |
| interpret_weather_patterns |
| conduct_vessel_maintenance |
| handle_maritime_emergencies |
| film_production_techniques |
| lighting_setup |
| script_writing |
| set_design |
| makeup_application |
| video_editing |
| sound_mixing |
| prop_management |
| shot_composition |
| production_scheduling |
| green_screen_filming |
| aerial_drone_filming |
| foley_sound_creation |
| philosophical_theory_analysis |
| logical_reasoning |
| conceptual_analysis |
| historical_contextualization |
| metaphysical_inquiry |
| textual_interpretation |
| philosophical_writing |
| premise_identification |
| syllogism_construction |
| phenomenological_description |
| axiological_evaluation |
| philosophical_system_mapping |
| philosophical_jargon_translation |
| maritime_transport_operations |
| operate_ship_systems |
| plan_voyage_routes |
| implement_maritime_safety_protocols |
| manage_crew_operations |
| comply_with_maritime_regulations |

Table 106: (Part 6 of 52) 3266 skills extracted from topics in Tables 90, 92, and 94.

**Skills**

| |
|---|
| operate_communication_equipment |
| perform_risk_assessments |
| execute_docking_procedures |
| implement_environmental_protection_measures |
| manage_maritime_documentation |
| conduct_maritime_security_checks |
| operate_precision_farming_equipment |
| implement_crop_monitoring_systems |
| interpret_satellite_imagery_for_crop_health |
| calibrate_agricultural_sensors |
| troubleshoot_smart_farming_devices |
| optimize_greenhouse_climate_control |
| maintain_automated_harvesting_machinery |
| configure_gps_guided_tractors |
| analyze_agricultural_drone_data |
| optimize_precision_fertilizer_application |
| implement_blockchain_for_supply_chain_tracking |
| design_automated_pest_control_systems |
| problem_formulation |
| variable_identification |
| equation_derivation |
| model_simplification |
| numerical_methods_implementation |
| sensitivity_analysis |
| error_analysis |
| statistical_inference |
| data_visualization |
| interdisciplinary_collaboration |
| model_documentation |
| model_calibration |
| model_comparison |
| design_workout_programs |
| manage_fuel_consumption |
| coordinate_port_operations |
| monitor_vessel_performance |
| operate_navigation_instruments |
| agricultural_technology_systems |
| analyze_soil_composition_data |
| manage_irrigation_automation |
| program_farm_management_software |
| design_hydroponics_systems |
| integrate_iot_devices_in_agriculture |
| develop_predictive_crop_yield_models |
| implement_vertical_farming_techniques |
| manage_livestock_monitoring_systems |
| mathematical_model_development |
| data_collection_and_analysis |
| assumption_making |
| parameter_estimation |
| dimensional_analysis |
| simulation_design |
| model_validation |
| optimization_techniques |
| computer_programming |
| literature_review |
| technical_writing |
| uncertainty_quantification |
| scenario_analysis |
| fitness_training_methods |
| perform_strength_exercises |

Table 108: (Part 7 of 52) 3266 skills extracted from topics in Tables 90, 92, and 94.

| Skills |
| --- |
| execute_cardiovascular_routines |
| adjust_training_intensity |
| track_fitness_progress |
| teach_flexibility_exercises |
| conduct_fitness_assessments |
| operate_gym_equipment |
| modify_exercises_for_injuries |
| explain_nutrition_principles |
| provide_motivational_coaching |
| society_and_humanities |
| cultural_analysis |
| historical_research |
| qualitative_data_analysis |
| philosophical_argumentation |
| anthropological_fieldwork |
| political_system_analysis |
| art_history_interpretation |
| demographic_trend_analysis |
| social_policy_evaluation |
| oral_history_collection |
| conflict_resolution |
| media_literacy |
| military_and_defense |
| weapon_handling |
| threat_assessment |
| logistics_management |
| physical_fitness_training |
| crisis_management |
| military_law_interpretation |
| emergency_medical_response |
| aerial_combat_maneuvering |
| counterterrorism_tactics |
| demonstrate_proper_form |
| monitor_heart_rate |
| implement_periodization_techniques |
| apply_plyometric_training |
| utilize_resistance_bands |
| practice_interval_training |
| incorporate_bodyweight_exercises |
| lead_group_fitness_classes |
| analyze_movement_patterns |
| critical_thinking |
| ethical_reasoning |
| social_theory_application |
| cross_cultural_communication |
| literary_interpretation |
| sociological_observation |
| comparative_religion_study |
| psychological_assessment |
| human_rights_advocacy |
| archival_research |
| ethnographic_writing |
| gender_studies_application |
| linguistic_analysis |
| tactical_planning |
| combat_strategy |
| military_intelligence_analysis |
| battlefield_communication |
| leadership_and_command |
| cyber_warfare_operations |
| reconnaissance_and_surveillance |
| military_equipment_maintenance |
| naval_navigation |
| military_diplomacy |

Table 110: (Part 8 of 52) 3266 skills extracted from topics in Tables 90, 92, and 94.

| Skills |
| --- |
| psychological_warfare_techniques |
| defense_policy_formulation |
| military_technology_development |
| financial_management_systems |
| financial_data_analysis |
| cash_flow_monitoring |
| investment_portfolio_management |
| financial_software_implementation |
| accounts_payable_and_receivable_management |
| financial_performance_metrics_tracking |
| tax_planning_and_compliance |
| financial_process_automation |
| internal_control_implementation |
| treasury_management |
| leadership_development_methods |
| effective_communication |
| decision_making |
| team_building |
| goal_setting |
| coaching_and_mentoring |
| time_management |
| motivating_others |
| ethical_decision_making |
| stress_management |
| cross_cultural_communication |
| organizational_skills |
| source_evaluation |
| cause_and_effect_analysis |
| comparative_analysis |
| primary_source_examination |
| historical_data_visualization |
| historiographical_analysis |
| military_budget_allocation |
| military_recruitment_and_training |
| military_history_analysis |
| budget_planning_and_forecasting |
| risk_assessment_and_management |
| financial_reporting_and_compliance |
| cost_control_and_optimization |
| audit_preparation_and_coordination |
| financial_statement_preparation |
| capital_structure_optimization |
| financial_modeling_and_scenario_analysis |
| financial_decision_support |
| financial_system_integration |
| active_listening |
| strategic_thinking |
| conflict_resolution |
| emotional_intelligence |
| performance_evaluation |
| change_management |
| problem_solving |
| critical_thinking |
| public_speaking |
| vision_casting |
| feedback_delivery |
| historical_event_analysis |
| chronological_ordering |
| contextual_interpretation |
| bias_identification |
| secondary_source_interpretation |
| cross_cultural_comparison |
| historical_argument_construction |

Table 112: (Part 9 of 52) 3266 skills extracted from topics in Tables 90, 92, and 94.

**Skills**

| | |
|---|---|
| evidence_based_reasoning | historical_narrative_creation |
| interdisciplinary_connection_making | long_term_impact_assessment |
| historical_trend_identification | critical_thinking_application |
| historical_methodology_implementation | historical_perspective_taking |
| medical_treatment_methods | diagnose_medical_conditions |
| develop_treatment_plans | administer_medications |
| perform_surgical_procedures | interpret_medical_test_results |
| monitor_patient_vital_signs | provide_emergency_medical_care |
| manage_chronic_diseases | prescribe_appropriate_medications |
| conduct_physical_examinations | apply_wound_care_techniques |
| implement_pain_management_strategies | perform_non_invasive_procedures |
| educate_patients_on_treatment_options | coordinate_multidisciplinary_care |
| assess_treatment_effectiveness | adjust_treatment_plans_as_needed |
| utilize_medical_imaging_technologies | apply_infection_control_measures |
| perform_cardiopulmonary_resuscitation | engineering_design_methods |
| problem_definition | requirements_analysis |
| concept_generation | design_iteration |
| prototype_development | feasibility_assessment |
| technical_drawing | computer_aided_design |
| design_for_manufacturability | failure_mode_analysis |
| design_optimization | systems_thinking |
| project_management | design_documentation |
| stakeholder_communication | cost_estimation |
| risk_assessment | sustainability_analysis |
| ergonomic_design | design_validation_testing |
| material_selection | design_for_assembly |
| quality_control_implementation | design_review_facilitation |
| reverse_engineering | visual_arts_creation |
| color_theory_application | perspective_drawing |
| composition_arrangement | brush_technique_mastery |
| digital_painting_proficiency | sketching_and_ideation |
| texture_creation | light_and_shadow_rendering |

Table 114: (Part 10 of 52) 3266 skills extracted from topics in Tables 90, 92, and 94.

**Skills**

anatomy_understanding
art_history_interpretation
abstract_concept_visualization
negative_space_utilization
gesture_drawing
art_material_knowledge
visual_balance_creation
emotional_expression_through_art
creative_problem_solving
plc_programming
scada_system_configuration
motion_control_implementation
sensor_integration
process_control_optimization
machine_vision_integration
predictive_maintenance_planning
automated_quality_control_setup
fieldbus_protocol_implementation
industrial_cybersecurity_implementation
automated_material_handling_system_design
industrial_cloud_computing_implementation
machine_learning_for_process_optimization
industrial_database_management
public_transit_planning
demand_forecasting
infrastructure_assessment
stakeholder_engagement
accessibility_planning
transit_scheduling
multimodal_integration
population_density_mapping
ridership_analysis

mixed_media_integration
visual_storytelling
proportion_and_scale_management
color_palette_selection
digital_software_navigation
stylization_and_abstraction
symbolism_incorporation
art_critique_and_analysis
industrial_automation_systems
hmi_design
industrial_network_setup
robot_programming
data_acquisition_system_design
safety_system_implementation
industrial_iot_implementation
energy_management_system_design
distributed_control_system_configuration
opc_ua_integration
digital_twin_development

industrial_edge_computing_setup
route_optimization
traffic_flow_analysis
budget_management
environmental_impact_evaluation
data_visualization
fare_system_design
land_use_analysis
transit_oriented_development_planning
service_frequency_optimization

Table 116: (Part 11 of 52) 3266 skills extracted from topics in Tables 90, 92, and 94.

| Skills |
| --- |
| transit_technology_evaluation |
| transit_policy_development |
| activity_research_and_discovery |
| time_management_and_scheduling |
| group_dynamics_management |
| location_scouting_and_evaluation |
| transportation_logistics_coordination |
| dietary_requirements_accommodation |
| permit_and_reservation_acquisition |
| environmental_impact_minimization |
| first_aid_and_safety_protocol_implementation |
| activity_customization_for_diverse_groups |
| post_activity_evaluation_and_feedback_collection |
| conflict_resolution_and_mediation |
| prioritize_tasks |
| set_smart_goals |
| implement_pomodoro_technique |
| eliminate_distractions |
| utilize_productivity_tools |
| apply_eisenhower_matrix |
| learn_to_say_no |
| identify_peak_productivity_hours |
| break_large_projects_into_smaller_tasks |
| set_deadlines_and_stick_to_them |
| public_health_management |
| health_policy_development |
| community_health_assessment |
| resource_allocation_optimization |
| outbreak_response_coordination |
| public_health_research_design |
| health_workforce_development |
| health_risk_assessment |
| public_outreach_coordination |
| recreational_activity_planning |
| budget_allocation_and_management |
| risk_assessment_and_mitigation |
| equipment_selection_and_preparation |
| weather_forecasting_interpretation |
| emergency_response_planning |
| accessibility_needs_assessment |
| team_building_exercise_design |
| cultural_sensitivity_awareness |
| time_management_techniques |
| create_to_do_lists |
| use_time_blocking |
| delegate_responsibilities |
| practice_effective_scheduling |
| conduct_time_audits |
| manage_email_efficiently |
| batch_similar_tasks |
| use_calendar_effectively |
| minimize_multitasking |
| practice_mindfulness_and_focus |
| epidemiological_data_analysis |
| disease_surveillance_implementation |
| health_program_planning |
| health_communication_strategy |
| health_equity_promotion |
| health_information_system_management |
| health_budget_management |
| health_education_program_design |

Table 118: (Part 12 of 52) 3266 skills extracted from topics in Tables 90, 92, and 94.

**Skills**

environmental_health_monitoring
health_data_visualization
cross_sector_collaboration_facilitation
population_health_trend_analysis
public_health_ethics_application
marketing_campaign_management
target_audience_identification
budget_planning_and_allocation
channel_selection_and_optimization
data_driven_decision_making
customer_segmentation_techniques
roi_calculation_and_analysis
campaign_timeline_management
brand_messaging_consistency
email_marketing_campaign_design
campaign_kpi_setting_and_monitoring
crisis_management_in_campaigns
campaign_compliance_and_legal_review
quality_control_systems
root_cause_analysis
quality_management_software_operation
failure_mode_and_effects_analysis
lean_six_sigma_implementation
corrective_action_planning
control_chart_interpretation
risk_assessment_and_mitigation
quality_documentation_management
nonconformity_identification_and_reporting
quality_cost_analysis
nonprofit_organization_management
fundraising_and_grant_writing
financial_management_for_nonprofits

health_emergency_preparedness
health_impact_evaluation
health_legislation_interpretation
health_technology_assessment
global_health_diplomacy
campaign_strategy_development
market_research_analysis
creative_content_creation
campaign_performance_tracking
a_b_testing_implementation
marketing_automation_tool_usage
cross_channel_campaign_coordination
stakeholder_communication_and_reporting
social_media_campaign_execution
conversion_rate_optimization
competitor_analysis_and_benchmarking
influencer_partnership_management
post_campaign_analysis_and_reporting
statistical_process_control
data_collection_and_analysis
process_mapping
design_of_experiments
quality_auditing
measurement_system_analysis
supplier_quality_management
continuous_improvement_facilitation
product_inspection_techniques

customer_feedback_analysis
strategic_planning
volunteer_management
board_governance

Table 120: (Part 13 of 52) 3266 skills extracted from topics in Tables 90, 92, and 94.

**Skills**

| |
|---|
| program_development_and_evaluation |
| nonprofit_marketing_and_communications |
| community_outreach_and_engagement |
| performance_measurement_and_impact_assessment |
| resource_allocation_and_budgeting |
| nonprofit_technology_implementation |
| conflict_resolution_and_negotiation |
| project_management_for_nonprofits |
| sports_and_recreation |
| team_coordination |
| equipment_maintenance |
| sports_rules_interpretation |
| nutrition_management_for_athletes |
| coaching_and_mentoring |
| outdoor_survival_skills |
| sports_medicine_first_aid |
| sports_data_analysis |
| sports_equipment_selection |
| sports_marketing_and_promotion |
| ecosystem_assessment |
| conservation_planning |
| environmental_impact_analysis |
| wildlife_population_monitoring |
| soil_conservation_practices |
| fisheries_stock_assessment |
| biodiversity_monitoring |
| invasive_species_control |
| environmental_policy_development |
| stakeholder_engagement |
| remote_sensing_interpretation |
| conflict_resolution_in_resource_management |
| environmental_law_compliance |
| donor_relations_and_stewardship |
| legal_compliance_and_risk_management |
| nonprofit_leadership_and_decision_making |
| stakeholder_relationship_management |
| ethical_decision_making_in_nonprofits |
| nonprofit_human_resources_management |
| advocacy_and_policy_influence |
| physical_fitness_training |
| strategic_game_planning |
| injury_prevention_techniques |
| athletic_performance_analysis |
| sports_psychology_application |
| sports_event_organization |
| sports_facility_management |
| sports_broadcasting_and_commentary |
| recreational_activity_planning |
| sports_officiating_and_refereeing |
| natural_resource_management |
| resource_inventory_management |
| sustainable_harvesting_techniques |
| habitat_restoration |
| water_quality_management |
| forest_management_planning |
| land_use_planning |
| climate_change_adaptation_strategies |
| watershed_management |
| resource_allocation_optimization |
| gis_mapping_and_analysis |
| ecological_modeling |
| sustainable_tourism_planning |

Table 122: (Part 14 of 52) 3266 skills extracted from topics in Tables 90, 92, and 94.

**Skills**

economic_market_analysis
statistical_analysis
trend_identification
competitive_landscape_assessment
supply_chain_analysis
risk_assessment_and_mitigation
macroeconomic_indicator_interpretation
market_entry_strategy_formulation
data_visualization_and_reporting
regulatory_impact_assessment
market_opportunity_identification
economic_policy_analysis
growth_rate_calculation
business_operations_management
supply_chain_management
quality_assurance
performance_measurement
risk_assessment
capacity_planning
workflow_analysis
continuous_improvement
stakeholder_management
change_management
team_leadership
problem_solving
logistics_coordination
musical_arts_production
music_composition
digital_audio_workstation_operation
instrument_performance
music_theory_application
live_sound_reinforcement

data_collection_and_management
econometric_modeling
market_segmentation
demand_forecasting
pricing_strategy_development
financial_ratio_analysis
industry_specific_knowledge_application
consumer_behavior_analysis
scenario_planning_and_simulation
swot_analysis_execution
investment_potential_evaluation
market_size_estimation
market_share_analysis
process_optimization
inventory_control
resource_allocation
project_management
lean_manufacturing
demand_forecasting
cost_reduction_strategies
data_driven_decision_making
operational_budgeting
time_management
strategic_planning
vendor_relationship_management
productivity_analysis
audio_engineering
sound_design
mixing_and_mastering
vocal_recording
arrangement_and_orchestration
music_production_software_proficiency

Table 124: (Part 15 of 52) 3266 skills extracted from topics in Tables 90, 92, and 94.

| Skills |
|---|
| midi_programming |
| acoustic_treatment_implementation |
| collaborative_project_coordination |
| studio_equipment_maintenance |
| music_marketing_and_promotion |
| animal_health_monitoring |
| breeding_program_implementation |
| disease_prevention_protocols |
| livestock_record_keeping |
| waste_management_implementation |
| animal_welfare_compliance |
| livestock_transportation_logistics |
| nutrition_plan_development |
| livestock_facility_design |
| sustainable_farming_practices_integration |
| livestock_product_quality_control |
| create_project_charter |
| develop_work_breakdown_structure |
| create_project_schedule |
| manage_stakeholder_expectations |
| monitor_project_progress |
| conduct_project_meetings |
| use_project_management_software |
| create_gantt_charts |
| resolve_project_conflicts |
| conduct_project_post_mortems |
| statistical_analysis_techniques |
| descriptive_statistics_calculation |
| correlation_analysis |
| variance_analysis |
| sampling_technique_selection |
| data_visualization |
| sample_creation_and_manipulation |
| music_business_management |
| music_copyright_understanding |
| audio_signal_processing |
| livestock_management_systems |
| feed_management_optimization |
| herd_productivity_analysis |
| pasture_rotation_planning |
| environmental_impact_assessment |
| genetic_improvement_strategies |
| market_trend_analysis |
| biosecurity_measures_implementation |
| automated_milking_system_operation |
| animal_behavior_assessment |
| project_management_techniques |
| define_project_scope |
| estimate_project_costs |
| identify_project_risks |
| allocate_resources_effectively |
| implement_change_control_processes |
| perform_critical_path_analysis |
| apply_agile_methodologies |
| manage_project_budget |
| perform_earned_value_analysis |
| implement_quality_control_measures |
| data_cleaning |
| hypothesis_testing |
| regression_modeling |
| probability_distribution_identification |
| outlier_detection |
| confidence_interval_calculation |

Table 126: (Part 16 of 52) 3266 skills extracted from topics in Tables 90, 92, and 94.

| Skills |
| --- |
| statistical_software_usage |
| multivariate_analysis |
| factor_analysis |
| power_analysis |
| bayesian_inference |
| principal_component_analysis |
| statistical_report_writing |
| construction_and_infrastructure |
| project_management |
| cost_estimation |
| material_selection |
| heavy_equipment_operation |
| welding_and_metalwork |
| electrical_wiring |
| building_code_compliance |
| risk_assessment_and_mitigation |
| soil_analysis |
| scaffolding_assembly |
| drainage_system_design |
| bridge_engineering |
| road_paving |
| waste_management_planning |
| seismic_retrofitting |
| sketching_garment_designs |
| pattern_making |
| textile_selection |
| trend_forecasting |
| digital_design_software_use |
| technical_flat_drawing |
| silhouette_development |
| fashion_history_analysis |
| garment_fitting_and_alteration |
| experimental_design |
| time_series_analysis |
| cluster_analysis |
| nonparametric_test_application |
| meta_analysis |
| survival_analysis |
| data_interpretation |
| structural_design |
| blueprint_reading |
| site_surveying |
| safety_protocol_implementation |
| concrete_pouring_and_finishing |
| plumbing_installation |
| hvac_system_installation |
| sustainable_design_implementation |
| quality_control_inspection |
| foundation_laying |
| load_calculation |
| traffic_flow_analysis |
| tunnel_construction |
| environmental_impact_assessment |
| urban_planning |
| fashion_design_methods |
| creating_moodboards |
| draping_fabric |
| color_theory_application |
| garment_construction |
| fashion_illustration |
| fabric_manipulation_techniques |
| collection_planning |
| sustainable_design_practices |
| fashion_marketing_integration |

Table 128: (Part 17 of 52) 3266 skills extracted from topics in Tables 90, 92, and 94.

| Skills |
| --- |
| accessory_design |
| building_construction_methods |
| blueprint_reading_interpretation |
| construction_site_management |
| cost_estimation_budgeting |
| foundation_laying_techniques |
| plumbing_system_installation |
| hvac_system_integration |
| masonry_techniques_application |
| welding_and_metalwork_execution |
| drywall_installation_finishing |
| sustainable_building_practices_implementation |
| building_code_compliance_assessment |
| construction_equipment_operation |
| building_information_modeling_bim |
| numerical_analysis |
| optimization_techniques |
| linear_algebra_application |
| mathematical_modeling |
| graph_theory_application |
| discrete_mathematics_utilization |
| fourier_analysis_application |
| matrix_computation |
| mathematical_software_usage |
| operations_research_techniques |
| signal_processing_methods |
| time_series_analysis |
| game_theory_application |
| configure_iot_devices |
| update_firmware_remotely |
| implement_security_protocols |
| troubleshoot_connectivity_issues |
| fashion_show_production |
| structural_design_analysis |
| material_selection_optimization |
| safety_protocol_implementation |
| project_scheduling_coordination |
| framing_and_roofing_execution |
| electrical_wiring_implementation |
| concrete_pouring_finishing |
| carpentry_skills_utilization |
| insulation_installation_methods |
| painting_and_coating_application |
| quality_control_inspection |
| waste_management_recycling |
| applied_mathematics_methods |
| differential_equation_solving |
| statistical_modeling |
| data_analysis_and_interpretation |
| computational_geometry |
| probability_theory_implementation |
| complex_analysis_techniques |
| stochastic_process_modeling |
| algorithm_design_and_analysis |
| dynamical_systems_analysis |
| cryptography_implementation |
| finite_element_analysis |
| machine_learning_algorithm_development |
| iot_device_management |
| monitor_device_health |
| manage_device_lifecycle |
| analyze_device_data |
| optimize_power_consumption |

Table 130: (Part 18 of 52) 3266 skills extracted from topics in Tables 90, 92, and 94.

**Skills**

| | |
|---|---|
| scale_iot_infrastructure | integrate_iot_platforms |
| manage_device_inventory | implement_over_the_air_updates |
| configure_device_authentication | monitor_network_performance |
| manage_device_groups | implement_data_encryption |
| create_device_dashboards | automate_device_onboarding |
| manage_device_permissions | analyze_device_logs |
| implement_edge_computing | optimize_data_transmission |
| manage_device_alerts | implement_device_diagnostics |
| ensure_regulatory_compliance | climate_change_analysis |
| data_collection_and_management | statistical_analysis |
| climate_modeling | greenhouse_gas_inventory_assessment |
| remote_sensing_interpretation | time_series_analysis |
| geospatial_data_analysis | scientific_report_writing |
| policy_impact_assessment | carbon_footprint_calculation |
| climate_scenario_development | adaptation_strategy_formulation |
| mitigation_measure_evaluation | interdisciplinary_collaboration |
| data_visualization | uncertainty_analysis |
| peer_reviewed_literature_synthesis | climate_risk_assessment |
| stakeholder_engagement | environmental_impact_assessment |
| renewable_energy_potential_analysis | ecosystem_vulnerability_assessment |
| climate_finance_analysis | sustainable_development_goal_alignment |
| climate_communication_and_outreach | government_system_analysis |
| comparative_government_analysis | political_structure_evaluation |
| policy_impact_assessment | legislative_process_mapping |
| electoral_system_comparison | constitutional_interpretation |
| bureaucratic_efficiency_measurement | power_distribution_analysis |
| governance_model_classification | political_ideology_identification |
| intergovernmental_relations_assessment | public_administration_evaluation |
| democratic_index_calculation | government_transparency_measurement |
| checks_and_balances_examination | federalism_vs_centralism_analysis |
| political_party_system_evaluation | government_budget_analysis |
| public_policy_formulation | civic_engagement_assessment |

Table 132: (Part 19 of 52) 3266 skills extracted from topics in Tables 90, 92, and 94.

| Skills |
| --- |
| government_accountability_measurement |
| regime_type_classification |
| international_governance_comparison |
| trend_analysis |
| color_theory_application |
| 3d_modeling_for_accessories |
| computer_aided_design |
| brand_identity_development |
| manufacturing_process_understanding |
| cost_estimation |
| client_communication |
| ergonomic_design_principles |
| pattern_making |
| fashion_forecasting |
| sourcing_materials |
| collaborative_design_skills |
| air_traffic_control_coordination |
| aircraft_maintenance_planning |
| aviation_safety_protocol_implementation |
| crew_resource_management |
| aviation_regulatory_compliance |
| passenger_flow_optimization |
| aviation_weather_interpretation |
| aviation_data_analysis |
| aviation_security_implementation |
| aviation_environmental_impact_assessment |
| construction_project_management |
| budget_estimation_and_cost_control |
| resource_allocation_and_management |
| quality_assurance_and_control |
| safety_protocol_implementation |
| progress_monitoring_and_reporting |
| political_risk_analysis |
| government_reform_strategy_development |
| fashion_accessory_design |
| sketching_techniques |
| material_selection |
| prototype_creation |
| market_research |
| sustainable_design_practices |
| quality_control_assessment |
| presentation_skills |
| fashion_history_knowledge |
| textile_manipulation |
| digital_rendering |
| technical_specification_writing |
| production_timeline_management |
| aviation_system_management |
| flight_scheduling_optimization |
| runway_capacity_management |
| fuel_efficiency_analysis |
| airport_operations_oversight |
| emergency_response_coordination |
| baggage_handling_system_management |
| aircraft_fleet_management |
| airspace_design_and_planning |
| ground_handling_operations_management |
| aviation_technology_integration |
| project_planning_and_scheduling |
| risk_assessment_and_mitigation |
| contract_negotiation_and_administration |
| stakeholder_communication_and_management |
| change_order_management |
| subcontractor_coordination |

Table 134: (Part 20 of 52) 3266 skills extracted from topics in Tables 90, 92, and 94.

**Skills**

```
building_code_compliance
value_engineering
procurement_and_logistics_management
site_selection_and_analysis
team_leadership_and_motivation
assess_nutritional_needs
create_balanced_meal_plans
design_personalized_diet_plans
interpret_nutrition_labels
adjust_meal_plans_for_dietary_restrictions
incorporate_supplements_into_diet_plans
plan_hydration_strategies
monitor_body_composition_changes
integrate_cultural_food_preferences
address_nutrient_deficiencies
create_budget_friendly_meal_plans
video_editing
audio_production
3d_modeling
color_grading
typography_design
social_media_content_optimization
user_interface_design
sound_design
web_design
augmented_reality_development
digital_storytelling
data_visualization
video_compression_techniques
environmental_impact_assessment
sustainable_design_principles_application
lifecycle_assessment_implementation

construction_document_interpretation
conflict_resolution
environmental_impact_assessment
construction_technology_implementation
nutrition_planning_methods
calculate_caloric_requirements
analyze_macro_nutrient_ratios
evaluate_food_quality_and_sources
track_daily_nutrient_intake
optimize_meal_timing_for_performance
educate_clients_on_nutrition_basics
adapt_plans_for_different_life_stages
manage_weight_loss_or_gain_programs
develop_sports_nutrition_strategies
digital_media_creation
graphic_design
motion_graphics_creation
animation_techniques
photo_manipulation
storyboard_creation
digital_illustration
video_compositing
digital_photography
virtual_reality_content_creation
live_streaming_management
interactive_media_design
digital_painting
sustainability_practices_development
resource_efficiency_analysis
waste_reduction_strategy_development
```

Table 136: (Part 21 of 52) 3266 skills extracted from topics in Tables 90, 92, and 94.

| Skills |
| --- |
| renewable_energy_integration |
| green_supply_chain_management |
| stakeholder_engagement_facilitation |
| sustainable_materials_selection |
| energy_audit_conducting |
| sustainable_agriculture_practices_implementation |
| corporate_social_responsibility_program_development |
| green_building_design_and_certification |
| eco_labeling_and_certification_management |
| logistics_system_optimization |
| inventory_management |
| demand_forecasting |
| transportation_mode_selection |
| reverse_logistics_management |
| cost_benefit_analysis |
| cross_docking_optimization |
| fleet_management |
| risk_assessment_and_mitigation |
| logistics_software_implementation |
| vendor_relationship_management |
| customs_and_compliance_management |
| logistics_process_automation |
| software_development_practices |
| code_review_execution |
| test_driven_development |
| code_refactoring |
| software_architecture_planning |
| debugging_and_troubleshooting |
| security_best_practices_implementation |
| requirements_gathering_and_analysis |
| pair_programming_facilitation |
| database_design_and_optimization |
| carbon_footprint_calculation |
| sustainability_reporting_and_disclosure |
| circular_economy_model_implementation |
| water_conservation_planning |
| biodiversity_impact_mitigation |
| sustainable_transportation_planning |
| supply_chain_analysis |
| route_optimization |
| warehouse_layout_design |
| order_fulfillment_process_improvement |
| data_driven_decision_making |
| lean_logistics_implementation |
| last_mile_delivery_planning |
| logistics_network_design |
| performance_metrics_development |
| sustainability_integration |
| capacity_planning |
| real_time_tracking_implementation |
| multi_modal_transportation_planning |
| version_control_management |
| agile_methodology_implementation |
| continuous_integration_setup |
| design_pattern_application |
| technical_documentation_writing |
| performance_optimization |
| code_quality_assessment |
| sprint_planning_and_execution |
| code_commenting_and_self_documentation |
| api_development_and_integration |

Table 138: (Part 22 of 52) 3266 skills extracted from topics in Tables 90, 92, and 94.

| Skills |
|---|
| devops_practices_implementation |
| code_modularity_and_reusability_improvement |
| unit_testing_implementation |
| scalability_planning_and_implementation |
| critical_thinking |
| academic_writing |
| literature_review |
| curriculum_development |
| peer_review |
| public_speaking |
| educational_technology_integration |
| scholarly_publishing |
| experiment_design |
| citation_management |
| academic_integrity_enforcement |
| educational_policy_analysis |
| academic_program_evaluation |
| meal_planning_and_preparation |
| conflict_resolution |
| effective_communication |
| child_rearing_techniques |
| work_life_balance_maintenance |
| emergency_preparedness |
| relationship_nurturing |
| financial_planning |
| family_tradition_cultivation |
| family_meeting_facilitation |
| digital_content_creation |
| writing_for_digital_platforms |
| social_media_management |
| graphic_design_for_digital_media |
| web_content_management |
| code_versioning_and_release_management |
| cross_platform_development |
| education_and_academia |
| research_methodology |
| data_analysis |
| classroom_management |
| student_assessment |
| grant_writing |
| teaching_methods |
| academic_advising |
| statistical_analysis |
| thesis_development |
| interdisciplinary_collaboration |
| learning_theories_application |
| instructional_design |
| family_life_management |
| household_budgeting |
| time_management |
| stress_management |
| home_organization |
| family_activity_coordination |
| household_chore_delegation |
| health_and_wellness_promotion |
| educational_support |
| technology_management_in_home |
| energy_conservation_practices |
| content_strategy_development |
| search_engine_optimization |
| video_production_and_editing |
| podcast_creation_and_hosting |
| email_marketing_campaign_creation |

Table 140: (Part 23 of 52) 3266 skills extracted from topics in Tables 90, 92, and 94.

**Skills**

| |
|---|
| data_analysis_for_content_performance |
| content_curation_and_aggregation |
| interactive_content_development |
| content_localization_and_translation |
| content_monetization_strategies |
| multimedia_content_production |
| augmented_reality_content_creation |
| chatbot_script_writing |
| arts_and_culture |
| cultural_analysis |
| artistic_technique_application |
| art_criticism_writing |
| artistic_style_identification |
| cultural_event_planning |
| cultural_heritage_preservation |
| cross_cultural_communication |
| cultural_policy_development |
| cultural_anthropology_research |
| cultural_diplomacy_implementation |
| cultural_symbolism_interpretation |
| cultural_trend_forecasting |
| environmental_conservation_methods |
| recycling_program_management |
| water_conservation_planning |
| sustainable_agriculture_practices |
| renewable_energy_system_design |
| ecosystem_monitoring_and_analysis |
| biodiversity_preservation_strategies |
| pollution_prevention_and_control |
| climate_change_mitigation_planning |
| environmental_education_program_design |
| sustainable_resource_management |
| user_experience_design |
| digital_storytelling |
| mobile_content_optimization |
| digital_asset_management |
| audience_segmentation_and_targeting |
| live_streaming_and_webinar_hosting |
| virtual_reality_experience_design |
| infographic_design_and_creation |
| art_appreciation |
| historical_context_interpretation |
| museum_curation |
| cultural_sensitivity_practice |
| art_restoration_techniques |
| art_collection_management |
| artistic_movement_analysis |
| art_market_evaluation |
| artistic_medium_experimentation |
| art_education_curriculum_design |
| artistic_composition_creation |
| art_exhibition_design |
| artistic_collaboration_facilitation |
| waste_reduction_implementation |
| energy_efficiency_assessment |
| habitat_restoration_techniques |
| carbon_footprint_calculation |
| environmental_impact_assessment |
| green_building_design_and_construction |
| sustainable_forestry_management |
| environmental_policy_development |
| sustainable_transportation_planning |
| wildlife_conservation_techniques |
| social_media_management |

Table 142: (Part 24 of 52) 3266 skills extracted from topics in Tables 90, 92, and 94.

**Skills**

| |
|---|
| content_creation |
| social_listening |
| trend_identification |
| community_building |
| influencer_collaboration |
| brand_voice_development |
| scheduling_and_automation |
| customer_service_handling |
| campaign_planning |
| social_media_policy_development |
| lifestyle_and_personal |
| stress_reduction |
| healthy_meal_preparation |
| goal_setting_and_achievement |
| work_life_balance_maintenance |
| habit_formation_and_breaking |
| relationship_building |
| personal_style_development |
| sleep_optimization |
| productivity_enhancement |
| personal_branding |
| decision_making |
| meditation_and_relaxation_techniques |
| law_and_government |
| policy_analysis |
| constitutional_interpretation |
| case_law_application |
| courtroom_advocacy |
| public_speaking |
| diplomatic_relations |
| election_management |
| administrative_law_application |
| audience_engagement |
| analytics_interpretation |
| crisis_management |
| hashtag_strategy |
| paid_advertising_management |
| visual_content_design |
| cross_platform_integration |
| performance_reporting |
| user_generated_content_curation |
| competitor_analysis |
| time_management |
| personal_finance_planning |
| effective_communication |
| mindfulness_practice |
| self_care_routines |
| decluttering_and_organization |
| conflict_resolution |
| fitness_program_design |
| emotional_intelligence_cultivation |
| public_speaking |
| networking_and_social_skills |
| creative_problem_solving |
| sustainable_living_practices |
| legal_research |
| legislative_drafting |
| regulatory_compliance |
| legal_writing |
| negotiation_and_mediation |
| government_budgeting |
| parliamentary_procedure |
| public_policy_implementation |
| ethics_and_integrity_management |

Table 144: (Part 25 of 52) 3266 skills extracted from topics in Tables 90, 92, and 94.

| Skills |
| --- |
| crisis_management |
| civic_engagement_facilitation |
| legislative_process_navigation |
| law_enforcement_administration |
| journalism_content_production |
| interview_techniques |
| storytelling_and_narrative_structure |
| headline_crafting |
| data_journalism |
| ethical_reporting |
| deadline_management |
| investigative_journalism_techniques |
| news_analysis |
| breaking_news_coverage |
| audience_engagement |
| fact_based_storytelling |
| visual_storytelling |
| analyze_source_text |
| apply_grammar_rules |
| research_cultural_context |
| implement_machine_learning_algorithms |
| adapt_idiomatic_expressions |
| utilize_computer_assisted_translation_tools |
| perform_localization |
| apply_natural_language_processing |
| implement_statistical_machine_translation |
| optimize_neural_machine_translation |
| perform_post_editing_machine_translations |
| integrate_translation_apis |
| customer_service_excellence |
| reservation_management |
| housekeeping_coordination |
| intergovernmental_coordination |
| human_rights_protection |
| judicial_decision_making |
| international_treaty_negotiation |
| research_and_fact_checking |
| news_writing |
| copy_editing |
| source_verification |
| multimedia_content_creation |
| news_judgment |
| social_media_reporting |
| content_curation |
| feature_writing |
| cross_platform_adaptation |
| search_engine_optimization |
| news_literacy |
| language_translation_methods |
| identify_linguistic_patterns |
| use_translation_software |
| maintain_translation_memory |
| evaluate_translation_quality |
| manage_terminology_databases |
| conduct_parallel_text_analysis |
| interpret_source_language_nuances |
| develop_custom_translation_algorithms |
| hospitality_service_management |
| conflict_resolution |
| front_desk_operations |
| food_and_beverage_service |

Table 146: (Part 26 of 52) 3266 skills extracted from topics in Tables 90, 92, and 94.

| Skills |
| --- |
| event_planning_and_execution |
| inventory_control |
| guest_satisfaction_monitoring |
| cultural_sensitivity |
| complaint_handling |
| team_leadership |
| financial_budgeting |
| quality_assurance_implementation |
| technology_systems_operation |
| problem_solving |
| diplomatic_negotiation |
| geopolitical_analysis |
| treaty_drafting |
| foreign_policy_formulation |
| economic_sanctions_management |
| international_organization_coordination |
| foreign_aid_administration |
| global_security_assessment |
| protocol_management |
| international_project_management |
| international_public_speaking |
| international_event_planning |
| international_media_relations |
| soil_preparation_techniques |
| irrigation_system_management |
| fertilizer_application_methods |
| seed_selection_and_planting |
| farm_equipment_operation |
| organic_farming_techniques |
| hydroponics_system_setup |
| soil_nutrient_analysis |
| precision_agriculture_implementation |
| revenue_management |
| staff_training_and_development |
| crisis_management |
| upselling_techniques |
| time_management |
| health_and_safety_compliance |
| marketing_and_promotion |
| sustainability_practices |
| multitasking |
| international_relations_management |
| cross_cultural_communication |
| conflict_resolution |
| multilateral_cooperation |
| international_law_interpretation |
| global_crisis_management |
| public_diplomacy_execution |
| international_trade_facilitation |
| international_mediation |
| foreign_language_proficiency |
| global_risk_assessment |
| diplomatic_etiquette_application |
| global_alliance_building |
| agricultural_farming_methods |
| crop_rotation_planning |
| pest_control_implementation |
| harvest_timing_optimization |
| sustainable_farming_practices |
| weather_pattern_analysis |
| greenhouse_management |
| crop_disease_identification |
| water_conservation_methods |
| livestock_integration_planning |

Table 148: (Part 27 of 52) 3266 skills extracted from topics in Tables 90, 92, and 94.

**Skills**

| |
|---|
| composting_and_soil_amendment |
| crop_yield_forecasting |
| vertical_farming_techniques |
| post_harvest_handling_and_storage |
| hazard_analysis_critical_control_point_implementation |
| sanitation_standard_operating_procedures_development |
| food_safety_risk_assessment |
| temperature_control_monitoring |
| microbial_contamination_prevention |
| personal_hygiene_training |
| food_storage_best_practices_implementation |
| cross_contamination_prevention |
| food_safety_culture_promotion |
| pest_control_management |
| food_safety_incident_investigation |
| food_safety_data_analysis_and_reporting |
| comparative_religion_analysis |
| religious_symbolism_decoding |
| religious_history_contextualization |
| religious_demographics_assessment |
| religious_ethics_evaluation |
| religious_art_interpretation |
| religious_sociology_application |
| religious_conflict_resolution |
| religious_anthropology_research |
| religious_law_interpretation |
| blockchain_system_development |
| distributed_ledger_design |
| cryptography_application |
| decentralized_application_development |
| blockchain_security_auditing |
| token_economics_modeling |
| farm_data_management_and_analysis |
| agricultural_drone_operation |
| agroforestry_implementation |
| food_safety_management |
| food_allergen_management |
| food_safety_audit_conducting |
| food_recall_protocol_execution |
| food_safety_legislation_compliance |
| food_safety_documentation_management |
| food_traceability_system_implementation |
| food_safety_equipment_maintenance |
| food_safety_training_program_development |
| religious_studies_analysis |
| sacred_text_interpretation |
| theological_concept_explanation |
| ritual_practice_examination |
| interfaith_dialogue_facilitation |
| myth_and_folklore_analysis |
| spiritual_experience_categorization |
| religious_philosophy_critique |
| religious_terminology_clarification |
| religious_psychology_assessment |
| religious_syncretism_identification |
| smart_contract_programming |
| consensus_mechanism_implementation |
| blockchain_architecture_planning |
| network_protocol_optimization |
| scalability_solution_implementation |
| interoperability_protocol_design |

Table 150: (Part 28 of 52) 3266 skills extracted from topics in Tables 90, 92, and 94.

| Skills |
|---|
| blockchain_governance_system_development |
| peer_to_peer_network_management |
| transaction_validation_algorithm_design |
| digital_signature_implementation |
| blockchain_integration_with_existing_systems |
| business_strategy_planning |
| competitive_intelligence_gathering |
| goal_setting_and_prioritization |
| risk_assessment_and_mitigation |
| stakeholder_management |
| business_model_innovation |
| kpi_development_and_tracking |
| strategic_partnership_development |
| value_chain_analysis |
| blue_ocean_strategy_formulation |
| strategic_communication |
| stage_presence |
| character_development |
| movement_and_physicality |
| memorization_techniques |
| ensemble_collaboration |
| prop_handling |
| accent_and_dialect_mastery |
| set_design_understanding |
| audience_engagement |
| warm_up_routines |
| blocking_and_staging |
| audition_preparation |
| energy_system_management |
| renewable_energy_integration |
| demand_response_management |
| smart_meter_data_analysis |
| data_structure_optimization |
| blockchain_performance_tuning |
| blockchain_api_development |
| blockchain_testing_and_debugging |
| market_analysis |
| swot_analysis_creation |
| financial_forecasting |
| resource_allocation_optimization |
| strategic_decision_making |
| scenario_planning |
| change_management_implementation |
| long_term_vision_creation |
| portfolio_management |
| balanced_scorecard_implementation |
| theatrical_performance_arts |
| vocal_projection |
| script_analysis |
| emotional_expression |
| timing_and_pacing |
| costume_management |
| stage_combat |
| makeup_application |
| lighting_cue_awareness |
| backstage_etiquette |
| voice_modulation |
| quick_change_techniques |
| rehearsal_discipline |
| power_grid_optimization |
| load_forecasting |
| energy_storage_planning |
| microgrid_design_and_control |

Table 152: (Part 29 of 52) 3266 skills extracted from topics in Tables 90, 92, and 94.

| Skills |
| --- |
| energy_efficiency_assessment |
| fault_detection_and_diagnosis |
| energy_market_analysis |
| voltage_regulation |
| energy_policy_implementation |
| energy_auditing |
| energy_cost_optimization |
| metacognitive_awareness |
| time_management |
| active_reading_strategies |
| memory_enhancement_techniques |
| self_assessment_and_reflection |
| resource_evaluation_and_selection |
| test_taking_strategies |
| motivation_maintenance |
| collaborative_learning_facilitation |
| continuous_improvement_mindset_cultivation |
| feedback_interpretation_and_application |
| deploy_cloud_infrastructure |
| implement_cloud_security_measures |
| design_scalable_architectures |
| implement_disaster_recovery_plans |
| monitor_cloud_performance |
| implement_containerization_technologies |
| configure_cloud_networking |
| implement_serverless_computing |
| integrate_cloud_services |
| configure_cloud_identity_management |
| sports_science_analysis |
| physiological_data_interpretation |
| injury_risk_assessment |
| training_program_design |
| power_quality_monitoring |
| peak_load_management |
| transmission_loss_reduction |
| distributed_generation_management |
| carbon_footprint_calculation |
| power_system_stability_analysis |
| learning_strategy_development |
| goal_setting_and_planning |
| note_taking_techniques |
| information_organization |
| critical_thinking_application |
| learning_style_identification |
| study_environment_optimization |
| stress_management_for_learning |
| attention_and_focus_improvement |
| technology_integration_for_learning |
| cloud_computing_systems |
| manage_virtual_machines |
| optimize_cloud_resource_allocation |
| configure_load_balancing |
| automate_cloud_deployments |
| manage_cloud_storage_solutions |
| orchestrate_microservices |
| optimize_cloud_costs |
| manage_multi_cloud_environments |
| implement_cloud_data_backup_strategies |
| optimize_cloud_application_performance |
| biomechanical_movement_analysis |
| performance_metrics_evaluation |
| nutrition_plan_development |
| recovery_strategy_optimization |

Table 154: (Part 30 of 52) 3266 skills extracted from topics in Tables 90, 92, and 94.

| Skills |
| --- |
| sports_equipment_analysis |
| athlete_monitoring_techniques |
| sports_psychology_implementation |
| fatigue_management_assessment |
| periodization_planning |
| sports_technology_integration |
| research_methodology_application |
| operate_spinning_machinery |
| perform_fabric_dyeing |
| maintain_textile_equipment |
| analyze_fabric_properties |
| operate_knitting_machines |
| troubleshoot_production_issues |
| handle_raw_materials |
| implement_sustainable_production_practices |
| operate_computerized_embroidery_machines |
| conduct_textile_testing |
| apply_textile_coating_techniques |
| effective_communication |
| problem_solving |
| teamwork_collaboration |
| conflict_resolution |
| project_management |
| emotional_intelligence |
| presentation_skills |
| stress_management |
| strategic_planning |
| data_analysis |
| digital_literacy |
| self_motivation |
| professional_etiquette |
| research_methodology |
| statistical_data_analysis |
| exercise_physiology_application |
| motion_capture_data_interpretation |
| sports_specific_testing_protocols |
| strength_and_conditioning_programming |
| performance_enhancement_strategies |
| textile_production_methods |
| set_up_weaving_looms |
| execute_screen_printing |
| implement_quality_control_measures |
| design_textile_patterns |
| apply_fabric_finishing_techniques |
| optimize_production_efficiency |
| manage_textile_inventory |
| perform_fabric_cutting_operations |
| operate_nonwoven_fabric_machinery |
| professional_skills |
| time_management |
| critical_thinking |
| leadership_ability |
| adaptability_flexibility |
| decision_making |
| networking_relationship_building |
| negotiation_skills |
| active_listening |
| creative_thinking |
| cultural_awareness |
| customer_service |
| organizational_skills |
| academic_writing_techniques |
| critical_thinking |

Table 156: (Part 31 of 52) 3266 skills extracted from topics in Tables 90, 92, and 94.

**Skills**

| |
|---|
| thesis_statement_formulation |
| literature_review_synthesis |
| academic_tone_maintenance |
| evidence_evaluation |
| abstract_writing |
| plagiarism_avoidance |
| data_analysis_interpretation |
| outline_development |
| revision_process_management |
| food_processing_techniques |
| implement_food_preservation_methods |
| maintain_food_safety_standards |
| execute_batch_processing |
| optimize_production_efficiency |
| implement_packaging_solutions |
| analyze_nutritional_content |
| troubleshoot_processing_issues |
| manage_waste_reduction_strategies |
| operate_automated_processing_lines |
| conduct_shelf_life_testing |
| self_assessment |
| career_research |
| interview_preparation |
| time_management |
| professional_development_planning |
| adaptability_and_flexibility |
| mentorship_seeking |
| career_path_mapping |
| leadership_development |
| negotiation_skills |
| stress_management |
| personal_swot_analysis |
| argument_construction |
| citation_management |
| paragraph_structuring |
| logical_flow_creation |
| proofreading_and_editing |
| academic_vocabulary_usage |
| scholarly_source_identification |
| peer_review_participation |
| academic_style_guide_adherence |
| apply_heat_treatment |
| operate_food_processing_equipment |
| perform_quality_control_checks |
| manage_ingredient_inventory |
| apply_fermentation_techniques |
| conduct_sensory_evaluation |
| develop_new_food_products |
| apply_food_chemistry_principles |
| implement_food_traceability_systems |
| apply_food_microbiology_knowledge |
| career_development_planning |
| goal_setting |
| resume_writing |
| personal_branding |
| skill_gap_analysis |
| decision_making |
| continuous_learning |
| work_life_balance_management |
| industry_trend_analysis |
| communication_skills_enhancement |
| conflict_resolution |
| financial_planning_for_career_growth |
| professional_etiquette |

Table 158: (Part 32 of 52) 3266 skills extracted from topics in Tables 90, 92, and 94.

**Skills**

```
legal_system_analysis
case_law_interpretation
legal_writing_and_drafting
legal_precedent_application
legal_system_evaluation
legal_policy_assessment
legal_reform_proposal_development
legal_risk_assessment
legal_trend_forecasting
cross_jurisdictional_analysis
legal_stakeholder_analysis
create_budget
set_financial_goals
build_emergency_fund
understand_compound_interest
negotiate_salary
calculate_net_worth
plan_for_retirement
research_financial_products
balance_checkbook
reduce_unnecessary_expenses
create_passive_income_streams
manage_risk_tolerance
use_budgeting_software
game_design
character_modeling
animation_rigging
game_engine_proficiency
narrative_writing
playtesting_and_debugging
version_control_management
shader_programming
```

```
legal_research_and_analysis
statutory_interpretation
comparative_law_analysis
legal_argument_construction
judicial_decision_analysis
legal_ethics_application
legal_data_visualization
legal_process_mapping
legal_system_efficiency_evaluation
legal_technology_integration
personal_finance_management
track_expenses
manage_debt
invest_wisely
optimize_tax_strategies
analyze_financial_statements
improve_credit_score
diversify_investment_portfolio
automate_bill_payments
compare_insurance_policies
maximize_employee_benefits
understand_mortgage_options
read_financial_news
video_game_development
level_creation
texture_mapping
scripting_and_programming
sound_design
user_interface_design
performance_optimization
asset_pipeline_development
artificial_intelligence_implementation
```

Table 160: (Part 33 of 52) 3266 skills extracted from topics in Tables 90, 92, and 94.

| Skills |
| --- |
| multiplayer_networking |
| project_management |
| health_and_medicine |
| perform_physical_examinations |
| administer_medications |
| develop_treatment_plans |
| provide_emergency_medical_care |
| counsel_patients_on_health_issues |
| sterilize_medical_equipment |
| prescribe_appropriate_medications |
| educate_patients_on_preventive_care |
| perform_health_risk_assessments |
| administer_vaccinations |
| conduct_mental_health_evaluations |
| analyze_nutritional_needs |
| infrastructure_development_planning |
| project_management |
| cost_benefit_analysis |
| resource_allocation |
| land_use_planning |
| utility_systems_planning |
| geographic_information_system_usage |
| sustainable_design_implementation |
| public-private-partnership-facilitation |
| demographic_trend_analysis |
| climate_resilience_planning |
| interdisciplinary_collaboration |
| construction_phasing_and_sequencing |
| defense_technology_systems |
| electronic_warfare_management |
| missile_guidance_programming |
| unmanned_vehicle_piloting |
| game_balancing |
| quality_assurance_testing |
| diagnose_medical_conditions |
| interpret_lab_results |
| conduct_surgical_procedures |
| monitor_patient_vital_signs |
| analyze_medical_imaging |
| maintain_patient_records |
| perform_cpr_and_first_aid |
| conduct_medical_research |
| manage_chronic_diseases |
| interpret_ecg_readings |
| assist_in_childbirth |
| perform_physical_therapy_exercises |
| operate_medical_devices |
| urban_planning |
| environmental_impact_assessment |
| stakeholder_engagement |
| risk_assessment_and_mitigation |
| transportation_network_design |
| data_analysis_and_interpretation |
| regulatory_compliance_management |
| budget_forecasting |
| infrastructure_lifecycle_management |
| economic_feasibility_studies |
| community_needs_assessment |
| long-term_strategic_planning |
| infrastructure_capacity_modeling |
| radar_system_operation |
| cybersecurity_implementation |
| satellite_communication_coordination |
| cryptography_application |

Table 162: (Part 34 of 52) 3266 skills extracted from topics in Tables 90, 92, and 94.

| Skills |
| --- |
| battlefield_network_administration |
| threat_detection_analysis |
| sensor_fusion_implementation |
| electromagnetic_spectrum_management |
| data_link_system_configuration |
| directed_energy_weapon_operation |
| quantum_computing_utilization |
| perform_statistical_analysis |
| apply_probability_theory |
| conduct_hypothesis_testing |
| use_mathematical_modeling |
| implement_regression_analysis |
| perform_multivariate_analysis |
| conduct_time_series_analysis |
| perform_numerical_simulations |
| apply_differential_equations |
| implement_machine_learning_algorithms |
| use_statistical_software_packages |
| conduct_monte_carlo_simulations |
| perform_factor_analysis |
| experimental_design |
| statistical_analysis |
| scientific_writing |
| hypothesis_formulation |
| instrument_calibration |
| peer_review_process |
| research_ethics_compliance |
| collaborative_research |
| sample_preparation |
| data_management |
| problem_solving |
| scientific_method_application |
| weapons_system_integration |
| command_and_control_optimization |
| ballistic_missile_defense_operation |
| combat_simulation_modeling |
| stealth_technology_application |
| artificial_intelligence_integration |
| mathematics_and_statistics |
| solve_complex_equations |
| interpret_data_visualizations |
| design_experimental_studies |
| calculate_descriptive_statistics |
| apply_linear_algebra |
| use_calculus_for_optimization |
| apply_bayesian_inference |
| use_matrix_operations |
| conduct_cluster_analysis |
| perform_data_cleaning_and_preprocessing |
| apply_game_theory |
| interpret_confidence_intervals |
| physical_science_research |
| data_collection |
| laboratory_techniques |
| literature_review |
| data_visualization |
| error_analysis |
| grant_proposal_writing |
| scientific_presentation |
| computational_modeling |
| safety_protocol_implementation |
| critical_thinking |
| equipment_troubleshooting |
| interdisciplinary_communication |

Table 164: (Part 35 of 52) 3266 skills extracted from topics in Tables 90, 92, and 94.

| Skills |
| --- |
| research_project_management |
| risk_assessment_and_hazard_identification |
| personal_protective_equipment_selection |
| osha_regulation_compliance |
| scaffolding_safety_inspection |
| confined_space_entry_procedures |
| safety_training_program_design |
| workplace_ergonomics_assessment |
| fire_prevention_and_protection |
| first_aid_and_cpr_administration |
| safety_signage_implementation |
| site_security_management |
| sustainable_resource_management |
| waste_reduction_and_recycling |
| water_conservation_techniques |
| carbon_footprint_calculation |
| environmental_policy_analysis |
| biodiversity_conservation |
| soil_contamination_remediation |
| sustainable_agriculture_practices |
| environmental_education_and_outreach |
| environmental_risk_management |
| space_exploration_technology |
| spacecraft_systems_engineering |
| mission_planning_and_analysis |
| life_support_systems_development |
| zero_gravity_adaptation_training |
| space_debris_tracking_and_mitigation |
| space_habitat_construction |
| radiation_shielding_design |
| thermal_control_systems_management |
| microgravity_experiment_design |
| construction_safety_standards |
| safety_plan_development |
| fall_protection_system_implementation |
| heavy_machinery_operation_safety |
| emergency_response_planning |
| accident_investigation_and_reporting |
| hazardous_material_handling |
| electrical_safety_management |
| noise_level_monitoring_and_control |
| tool_and_equipment_safety_inspection |
| environment_and_resources |
| environmental_impact_assessment |
| renewable_energy_implementation |
| ecosystem_restoration |
| green_building_design |
| natural_disaster_mitigation |
| air_quality_monitoring |
| environmental_data_analysis |
| climate_change_adaptation_planning |
| ecological_footprint_assessment |
| sustainable_transportation_planning |
| rocket_propulsion_design |
| orbital_mechanics_calculation |
| space_navigation_programming |
| space_suit_engineering |
| satellite_communication_implementation |
| interplanetary_trajectory_planning |
| extraterrestrial_resource_utilization |
| space_robotics_programming |
| space_telescope_operation |
| space_weather_forecasting |

Table 166: (Part 36 of 52) 3266 skills extracted from topics in Tables 90, 92, and 94.

| **Skills** |
| --- |
| launch_vehicle_integration |
| data_cleaning |
| data_visualization |
| regression_analysis |
| cluster_analysis |
| exploratory_data_analysis |
| data_interpretation |
| outlier_detection |
| data_transformation |
| text_mining |
| machine_learning_application |
| data_normalization |
| bayesian_analysis |
| experimental_design |
| deliver_clear_message |
| structure_speech_effectively |
| manage_stage_presence |
| handle_q_and_a_sessions |
| utilize_visual_aids |
| manage_speech_timing |
| use_rhetorical_devices |
| research_and_prepare_content |
| use_humor_appropriately |
| project_confidence |
| multimedia_content_production |
| audio_mixing |
| storyboard_creation |
| camera_operation |
| sound_recording |
| motion_graphics_creation |
| animation_techniques |
| live_streaming_management |
| data_analysis_methods |
| statistical_analysis |
| hypothesis_testing |
| time_series_analysis |
| principal_component_analysis |
| predictive_modeling |
| feature_selection |
| correlation_analysis |
| dimensionality_reduction |
| sentiment_analysis |
| data_sampling_techniques |
| multivariate_analysis |
| data_imputation |
| public_speaking_techniques |
| engage_audience |
| use_body_language |
| control_vocal_tone_and_pitch |
| incorporate_storytelling |
| adapt_to_different_audiences |
| overcome_stage_fright |
| improvise_and_think_on_feet |
| create_memorable_openings_and_closings |
| maintain_eye_contact |
| handle_technical_difficulties |
| video_editing |
| graphic_design |
| script_writing |
| lighting_setup |
| color_grading |
| 3d_modeling |
| special_effects_implementation |
| content_management_system_operation |

Table 168: (Part 37 of 52) 3266 skills extracted from topics in Tables 90, 92, and 94.

| Skills |
| --- |
| social_media_optimization |
| video_compression_techniques |
| user_experience_design |
| copyright_law_understanding |
| content_strategy_development |
| negotiation_strategy_development |
| conflict_resolution |
| persuasive_communication |
| stakeholder_analysis |
| critical_thinking |
| goal_setting |
| body_language_interpretation |
| power_dynamics_understanding |
| concession_management |
| alternative_option_generation |
| value_creation_identification |
| information_gathering |
| assertiveness_balancing |
| extended_reality_development |
| virtual_environment_creation |
| motion_tracking_implementation |
| user_interface_design_for_xr |
| haptic_feedback_integration |
| immersive_storytelling_techniques |
| cross-platform_xr_development |
| performance_optimization_for_xr |
| volumetric_capture_and_processing |
| xr_content_creation |
| xr_user_experience_design |
| xr_networking_and_multiplayer_development |
| computer_vision_for_xr |
| xr_testing_and_quality_assurance |
| digital_asset_management |
| interactive_media_development |
| project_management |
| audience_analysis |
| cross-platform_adaptation |
| active_listening |
| emotional_intelligence |
| strategic_planning |
| problem_solving |
| data_analysis |
| rapport_building |
| cultural_awareness |
| win_win_solution_creation |
| deadline_pressure_handling |
| risk_assessment |
| negotiation_style_adaptation |
| patience_and_timing_management |
| ethical_decision_making |
| 3d_modeling_and_design |
| augmented_reality_programming |
| spatial_audio_integration |
| real_time_rendering_optimization |
| gesture_recognition_development |
| xr_hardware_integration |
| shader_programming_for_xr |
| xr_interaction_design |
| photogrammetry_for_xr_assets |
| mixed_reality_application_development |
| spatial_computing_implementation |
| xr_prototyping_and_rapid_development |
| team_sports_techniques |

Table 170: (Part 38 of 52) 3266 skills extracted from topics in Tables 90, 92, and 94.

| Skills |
| --- |
| execute_passing_techniques |
| apply_offensive_strategies |
| coordinate_team_movements |
| communicate_effectively_on_field |
| execute_set_plays |
| perform_physical_conditioning |
| develop_spatial_awareness |
| execute_feinting_maneuvers |
| perform_tactical_substitutions |
| execute_scoring_techniques |
| entertainment_and_media |
| script_writing |
| audio_production |
| digital_marketing |
| graphic_design |
| podcast_hosting |
| voice_over_narration |
| audience_engagement |
| media_analysis |
| content_curation |
| interview_conducting |
| sound_design |
| emerging_technology_trends |
| evaluate_technological_feasibility |
| assess_societal_impact |
| develop_adoption_strategies |
| perform_competitive_analysis |
| evaluate_ethical_implications |
| integrate_emerging_technologies |
| analyze_patent_landscapes |
| assess_cybersecurity_risks |
| facilitate_cross_disciplinary_collaboration |
| implement_defensive_formations |
| perform_ball_handling_drills |
| analyze_opponent_tactics |
| maintain_proper_positioning |
| adapt_to_game_situations |
| practice_teamwork_exercises |
| master_sport_specific_rules |
| implement_zonal_marking |
| utilize_equipment_effectively |
| practice_injury_prevention_methods |
| content_creation |
| video_editing |
| social_media_management |
| public_speaking |
| live_streaming |
| music_composition |
| special_effects_creation |
| brand_development |
| trend_forecasting |
| event_planning |
| news_reporting |
| media_law_compliance |
| analyze_market_trends |
| predict_future_innovations |
| identify_disruptive_technologies |
| conduct_technology_forecasting |
| create_technology_roadmaps |
| assess_scalability_potential |
| manage_innovation_portfolios |
| develop_proof_of_concepts |
| evaluate_regulatory_compliance |

Table 172: (Part 39 of 52) 3266 skills extracted from topics in Tables 90, 92, and 94.

| Skills |
| --- |
| conduct_technology_due_diligence |
| security_system_planning |
| threat_analysis |
| security_policy_development |
| surveillance_system_planning |
| budget_allocation |
| perimeter_security_design |
| data_protection_strategy |
| compliance_regulation_interpretation |
| incident_response_planning |
| physical_security_layout_design |
| security_staff_coordination |
| write_professional_emails |
| conduct_productive_meetings |
| negotiate_business_deals |
| manage_conflict_resolution |
| craft_clear_business_reports |
| use_nonverbal_communication |
| give_constructive_feedback |
| utilize_digital_communication_tools |
| implement_internal_communication_strategies |
| master_public_speaking |
| employ_active_questioning_techniques |
| assess_student_knowledge |
| create_personalized_lesson_plans |
| adapt_teaching_methods |
| track_student_progress |
| motivate_and_engage_students |
| address_individual_learning_styles |
| implement_assessment_techniques |
| foster_critical_thinking_skills |
| communicate_effectively_with_students |
| communicate_complex_technical_concepts |
| risk_assessment |
| vulnerability_identification |
| access_control_design |
| emergency_response_planning |
| security_technology_evaluation |
| personnel_security_management |
| security_audit_conducting |
| security_awareness_training |
| security_equipment_selection |
| cybersecurity_integration |
| business_communication_methods |
| deliver_effective_presentations |
| active_listening |
| create_persuasive_proposals |
| facilitate_group_discussions |
| develop_interpersonal_relationships |
| adapt_communication_style |
| practice_cross_cultural_communication |
| create_effective_visual_aids |
| write_concise_business_memos |
| subject_specific_tutoring |
| identify_learning_gaps |
| explain_complex_concepts_simply |
| provide_constructive_feedback |
| use_subject_specific_resources |
| manage_tutoring_time_effectively |
| develop_practice_exercises |
| build_student_confidence |
| utilize_educational_technology |
| set_achievable_learning_goals |

Table 174: (Part 40 of 52) 3266 skills extracted from topics in Tables 90, 92, and 94.

| Skills |
| --- |
| provide_study_strategies |
| urban_development_planning |
| land_use_planning |
| infrastructure_needs_assessment |
| public_transportation_planning |
| community_engagement_facilitation |
| historic_preservation_integration |
| traffic_flow_optimization |
| economic_development_planning |
| smart_city_technology_integration |
| mixed_use_development_design |
| urban_density_management |
| conduct_field_surveys |
| operate_remote_sensing_equipment |
| model_climate_systems |
| collect_sediment_cores |
| map_geological_formations |
| conduct_hydrological_assessments |
| operate_ground_penetrating_radar |
| analyze_ocean_current_patterns |
| conduct_soil_analysis |
| use_geographic_information_systems |
| conduct_environmental_impact_assessments |
| analyze_tree_ring_data |
| conduct_geophysical_surveys |
| close_reading |
| textual_interpretation |
| character_analysis |
| theme_extraction |
| narrative_perspective_analysis |
| historical_context_evaluation |
| stylistic_analysis |
| create_supportive_learning_environment |
| zoning_regulation_analysis |
| demographic_trend_forecasting |
| environmental_impact_evaluation |
| sustainable_design_implementation |
| urban_growth_modeling |
| affordable_housing_strategy_development |
| green_space_allocation |
| urban_revitalization_strategy_creation |
| urban_resilience_planning |
| walkability_assessment |
| earth_science_research |
| analyze_geological_samples |
| interpret_satellite_imagery |
| perform_geochemical_analysis |
| analyze_seismic_data |
| measure_atmospheric_composition |
| analyze_ice_core_samples |
| perform_radiometric_dating |
| monitor_volcanic_activity |
| interpret_paleoclimate_data |
| analyze_rock_formations |
| operate_mass_spectrometers |
| model_plate_tectonics |
| literary_arts_analysis |
| critical_thinking |
| literary_device_identification |
| plot_structure_analysis |
| symbolism_recognition |
| genre_classification |
| comparative_literature_analysis |
| rhetorical_device_recognition |

Table 176: (Part 41 of 52) 3266 skills extracted from topics in Tables 90, 92, and 94.

| Skills |
| --- |
| intertextuality_identification |
| literary_theory_application |
| figurative_language_interpretation |
| literary_criticism_writing |
| citation_and_referencing |
| literary_adaptation_analysis |
| financial_analysis |
| strategic_planning |
| project_management |
| budget_forecasting |
| supply_chain_optimization |
| business_valuation |
| negotiation_and_deal_making |
| business_process_improvement |
| pricing_strategy_development |
| economic_policy_analysis |
| international_trade_management |
| entrepreneurship_and_innovation |
| marketing_strategy_development |
| analyze_historical_context |
| research_case_law |
| advocate_for_equality |
| communicate_complex_ideas |
| negotiate_legislative_compromises |
| organize_grassroots_movements |
| implement_compliance_measures |
| mediate_conflicts |
| analyze_voting_patterns |
| educate_public_on_rights |
| technical_documentation_writing |
| audience_analysis |
| clear_writing |
| cultural_context_interpretation |
| authorial_intent_analysis |
| literary_movement_categorization |
| research_methodology |
| literary_period_identification |
| business_and_economics |
| market_research |
| risk_management |
| data_interpretation |
| investment_portfolio_management |
| cost_benefit_analysis |
| economic_modeling |
| financial_statement_preparation |
| competitive_analysis |
| business_plan_writing |
| mergers_and_acquisitions_planning |
| business_ethics_implementation |
| organizational_leadership |
| civil_rights_legislation |
| interpret_legal_documents |
| draft_policy_proposals |
| identify_discrimination_patterns |
| build_coalitions |
| assess_societal_impact |
| debate_constitutional_issues |
| evaluate_policy_effectiveness |
| conduct_public_hearings |
| litigate_civil_rights_cases |
| investigate_rights_violations |
| information_gathering |
| content_organization |
| technical_jargon_translation |

Table 178: (Part 42 of 52) 3266 skills extracted from topics in Tables 90, 92, and 94.

| Skills |
| --- |
| document_structuring |
| version_control_management |
| proofreading_and_editing |
| metadata_tagging |
| user_testing_coordination |
| technical_research |
| api_documentation |
| localization_preparation |
| index_creation |
| template_design |
| revision_tracking |
| project_management |
| quality_assurance_testing |
| policy_analysis |
| data_interpretation |
| impact_assessment |
| problem_identification |
| policy_evaluation |
| public_speaking |
| budget_allocation |
| regulatory_framework_design |
| cross_sector_collaboration |
| risk_assessment |
| conflict_resolution |
| scenario_planning |
| political_landscape_analysis |
| web_development |
| data_analysis |
| cybersecurity_implementation |
| database_management |
| ui_ux_design |
| search_engine_optimization |
| style_guide_adherence |
| collaborative_editing |
| visual_aid_creation |
| accessibility_implementation |
| feedback_incorporation |
| software_tool_proficiency |
| troubleshooting_guide_creation |
| cross_referencing |
| glossary_compilation |
| document_formatting |
| stakeholder_communication |
| deadline_adherence |
| public_policy_development |
| stakeholder_engagement |
| legislative_drafting |
| cost_benefit_analysis |
| research_methodology |
| strategic_planning |
| negotiation_and_consensus_building |
| project_management |
| policy_implementation_planning |
| evidence_based_decision_making |
| policy_communication |
| performance_measurement |
| ethical_reasoning |
| digital_technologies |
| mobile_app_creation |
| cloud_computing_management |
| network_administration |
| software_engineering |
| digital_marketing |
| social_media_management |

Table 180: (Part 43 of 52) 3266 skills extracted from topics in Tables 90, 92, and 94.

| Skills |
| --- |
| content_management_system_usage |
| machine_learning_implementation |
| internet_of_things_integration |
| augmented_reality_creation |
| digital_project_management |
| api_integration |
| responsive_web_design |
| implement_firewalls |
| perform_vulnerability_assessments |
| deploy_virtual_private_networks |
| conduct_penetration_testing |
| manage_security_information_and_event_management |
| perform_log_analysis |
| implement_multi-factor_authentication |
| patch_management_and_updates |
| implement_data_loss_prevention_strategies |
| manage_security_policies |
| perform_malware_analysis |
| configure_network_security_appliances |
| implement_endpoint_protection_solutions |
| perform_threat_hunting |
| literature_review_conducting |
| hypothesis_development |
| data_collection_methods_selection |
| interview_technique_application |
| qualitative_data_analysis |
| academic_writing_and_citation |
| research_proposal_drafting |
| data_visualization_creation |
| sampling_technique_application |
| research_software_utilization |
| systematic_review_performing |
| artificial_intelligence_programming |
| blockchain_development |
| virtual_reality_development |
| data_visualization |
| it_support_and_troubleshooting |
| version_control_system_usage |
| network_security_systems |
| configure_intrusion_detection_systems |
| manage_access_control_lists |
| monitor_network_traffic |
| implement_encryption_protocols |
| configure_network_segmentation |
| conduct_security_audits |
| design_secure_network_architectures |
| respond_to_security_incidents |
| implement_secure_wireless_networks |
| conduct_risk_assessments |
| manage_security_compliance |
| academic_research_methods |
| research_question_formulation |
| experimental_design_planning |
| survey_design_and_implementation |
| statistical_analysis_performing |
| research_ethics_compliance |
| peer_review_process_navigation |
| grant_application_writing |
| research_methodology_selection |
| validity_and_reliability_assessment |
| meta_analysis_conducting |
| research_presentation_skills |

Table 182: (Part 44 of 52) 3266 skills extracted from topics in Tables 90, 92, and 94.

| Skills |
| --- |
| academic_publication_process_understanding |
| interdisciplinary_research_collaboration |
| educational_teaching_methods |
| classroom_management |
| differentiated_instruction |
| educational_technology_integration |
| formative_feedback_provision |
| collaborative_learning_implementation |
| adaptive_teaching_strategies |
| behavior_management |
| active_listening_and_communication |
| inclusive_education_practices |
| student_motivation_techniques |
| flipped_classroom_implementation |
| problem_based_learning_facilitation |
| individual_sports_training |
| analyze_performance_metrics |
| develop_mental_toughness |
| improve_technique_efficiency |
| manage_training_intensity |
| enhance_flexibility_and_mobility |
| adapt_training_to_environmental_conditions |
| utilize_sports_specific_equipment |
| incorporate_cross_training_methods |
| develop_sport_specific_skills |
| optimize_sleep_patterns |
| athletic_coaching_methods |
| performance_analysis |
| athlete_motivation_strategies |
| goal_setting_and_planning |
| strength_and_conditioning_programming |
| team_building_exercises |
| research_time_management |
| lesson_planning |
| student_assessment |
| curriculum_development |
| student_engagement_techniques |
| inquiry_based_learning_facilitation |
| project_based_learning_design |
| learning_styles_accommodation |
| critical_thinking_promotion |
| educational_data_analysis |
| multicultural_teaching_approaches |
| peer_tutoring_facilitation |
| blended_learning_design |
| educational_game_design |
| create_personalized_training_plan |
| optimize_nutrition_intake |
| implement_recovery_strategies |
| set_realistic_goals |
| prevent_sports_injuries |
| track_progress_systematically |
| balance_strength_and_endurance_training |
| maintain_proper_form_and_posture |
| manage_pre_competition_anxiety |
| analyze_and_improve_biomechanics |
| training_program_design |
| injury_prevention_techniques |
| communication_with_athletes |
| sport_specific_technique_instruction |
| nutrition_guidance |
| psychological_preparation |

Table 184: (Part 45 of 52) 3266 skills extracted from topics in Tables 90, 92, and 94.

| Skills |
| --- |
| recovery_and_regeneration_methods |
| video_analysis_and_feedback |
| competition_strategy_development |
| stress_management_techniques |
| sports_ethics_and_fair_play_promotion |
| spatial_planning |
| furniture_arrangement |
| material_selection |
| 3d_modeling_and_visualization |
| budget_management |
| trend_analysis |
| architectural_drawing_interpretation |
| accessory_styling |
| building_code_compliance |
| universal_design_implementation |
| moodboard_creation |
| computer_aided_design_software_proficiency |
| vendor_and_contractor_coordination |
| linguistics_theory_analysis |
| compare_language_typologies |
| identify_morphological_patterns |
| interpret_semantic_relationships |
| conduct_discourse_analysis |
| formulate_linguistic_hypotheses |
| analyze_corpus_data |
| critique_linguistic_theories |
| explain_language_acquisition_processes |
| apply_psycholinguistic_principles |
| evaluate_language_universals |
| material_property_characterization |
| failure_mode_identification |
| x_ray_diffraction_analysis |
| biomechanical_analysis |
| periodization_planning |
| leadership_skill_development |
| athlete_assessment_and_evaluation |
| interior_space_design |
| color_theory_application |
| lighting_design |
| ergonomic_design |
| sustainable_design_implementation |
| client_communication |
| space_optimization |
| textile_and_fabric_selection |
| project_management |
| historical_style_adaptation |
| acoustics_management |
| scale_and_proportion_balancing |
| presentation_skills |
| analyze_linguistic_structures |
| apply_phonological_rules |
| evaluate_syntactic_theories |
| assess_pragmatic_contexts |
| examine_historical_language_changes |
| design_linguistic_experiments |
| apply_computational_linguistics_methods |
| synthesize_cross_linguistic_data |
| investigate_sociolinguistic_variations |
| analyze_phonetic_transcriptions |
| materials_engineering_analysis |
| microstructure_analysis |
| stress_strain_curve_interpretation |
| scanning_electron_microscopy |

Table 186: (Part 46 of 52) 3266 skills extracted from topics in Tables 90, 92, and 94.

**Skills**

| Skills |
| --- |
| thermal_analysis_techniques |
| corrosion_resistance_evaluation |
| phase_diagram_interpretation |
| finite_element_analysis |
| non_destructive_testing_methods |
| polymer_degradation_assessment |
| fatigue_life_prediction |
| military_intelligence_operations |
| data_analysis |
| covert_surveillance |
| cryptography_and_decryption |
| human_intelligence_collection |
| strategic_planning |
| source_cultivation |
| report_writing |
| cyber_intelligence_gathering |
| cultural_awareness |
| intelligence_briefing_delivery |
| electronic_warfare_techniques |
| imagery_interpretation |
| industry_and_manufacturing |
| quality_control_management |
| lean_manufacturing_implementation |
| inventory_management |
| equipment_maintenance_scheduling |
| product_lifecycle_management |
| manufacturing_cost_analysis |
| assembly_line_balancing |
| just_in_time_inventory_control |
| material_requirements_planning |
| industrial_robotics_programming |
| assess_patient_condition |
| mechanical_testing_procedures |
| composite_material_design |
| material_selection_optimization |
| fracture_mechanics_assessment |
| surface_treatment_analysis |
| crystallography_interpretation |
| material_performance_modeling |
| information_gathering |
| threat_assessment |
| signal_interception |
| geospatial_intelligence_interpretation |
| counterintelligence_operations |
| risk_mitigation |
| pattern_recognition |
| operational_security_maintenance |
| foreign_language_proficiency |
| tactical_decision_making |
| asset_management |
| open_source_intelligence_analysis |
| network_analysis |
| production_line_optimization |
| supply_chain_coordination |
| industrial_safety_compliance |
| process_automation_design |
| resource_allocation_planning |
| industrial_waste_reduction |
| production_capacity_planning |
| industrial_ergonomics_application |
| factory_layout_design |
| statistical_process_control |
| emergency_medical_response |
| perform_cpr |

Table 188: (Part 47 of 52) 3266 skills extracted from topics in Tables 90, 92, and 94.

| Skills |
| --- |
| control_bleeding |
| use_automated_external_defibrillator |
| treat_shock |
| handle_burn_injuries |
| manage_cardiac_emergencies |
| triage_multiple_casualties |
| operate_emergency_vehicles |
| perform_basic_life_support |
| handle_hazardous_materials_exposure |
| supply_chain_optimization |
| inventory_management |
| warehouse_layout_design |
| logistics_network_planning |
| data_analysis_and_visualization |
| just_in_time_inventory_control |
| supply_chain_modeling |
| performance_metrics_tracking |
| negotiation_and_contract_management |
| quality_control_implementation |
| supply_chain_software_utilization |
| cross_functional_team_coordination |
| supply_chain_resilience_planning |
| manufacturing_process_optimization |
| lean_manufacturing_implementation |
| supply_chain_optimization |
| process_flow_mapping |
| root_cause_analysis |
| equipment_efficiency_assessment |
| production_scheduling_optimization |
| continuous_improvement_facilitation |
| automation_integration |
| cost_benefit_analysis |
| administer_first_aid |
| manage_airway_obstruction |
| immobilize_fractures |
| recognize_stroke_symptoms |
| administer_emergency_medications |
| communicate_with_emergency_services |
| use_personal_protective_equipment |
| manage_environmental_emergencies |
| provide_emotional_support_to_patients |
| demand_forecasting |
| route_optimization |
| supplier_relationship_management |
| risk_assessment_and_mitigation |
| lean_manufacturing_implementation |
| reverse_logistics_management |
| cost_benefit_analysis |
| process_improvement_techniques |
| sustainability_assessment |
| transportation_mode_selection |
| blockchain_integration_for_transparency |
| global_trade_compliance_management |
| last_mile_delivery_optimization |
| production_line_analysis |
| quality_control_management |
| inventory_management |
| statistical_process_control |
| six_sigma_methodology_application |
| waste_reduction_techniques |
| bottleneck_identification_and_resolution |
| data_driven_decision_making |
| ergonomic_workplace_design |
| resource_allocation_optimization |

Table 190: (Part 48 of 52) 3266 skills extracted from topics in Tables 90, 92, and 94.

| **Skills** |
| --- |
| performance_metrics_development |
| value_stream_mapping |
| change_management_implementation |
| language_and_communication |
| public_speaking |
| nonverbal_communication |
| persuasive_writing |
| empathetic_communication |
| proofreading_and_editing |
| translation_and_interpretation |
| effective_questioning |
| speech_writing |
| audience_analysis |
| facilitation_skills |
| social_media_communication |
| pure_mathematics_theory |
| logical_deduction |
| mathematical_modeling |
| axiom_formulation |
| set_theory_application |
| topology_conceptualization |
| mathematical_induction_application |
| group_theory_implementation |
| category_theory_application |
| functional_analysis_implementation |
| graph_theory_application |
| abstract_algebra_manipulation |
| mathematical_proof_writing |
| cultural_pattern_analysis |
| identify_cultural_symbols |
| interpret_cultural_rituals |
| document_ethnographic_data |
| predictive_maintenance_planning |
| cycle_time_reduction |
| cross_functional_team_leadership |
| active_listening |
| written_communication |
| cross_cultural_communication |
| conflict_resolution |
| technical_writing |
| foreign_language_proficiency |
| body_language_interpretation |
| presentation_skills |
| tone_and_voice_modulation |
| interpersonal_communication |
| business_writing |
| emotional_intelligence |
| abstract_reasoning |
| theorem_proving |
| formal_proof_construction |
| mathematical_notation_usage |
| number_theory_analysis |
| algebraic_structure_manipulation |
| complex_analysis_techniques |
| differential_geometry_interpretation |
| measure_theory_utilization |
| combinatorics_problem_solving |
| mathematical_logic_reasoning |
| real_analysis_techniques |
| mathematical_conjecture_formulation |
| observe_cultural_behaviors |
| analyze_social_structures |
| compare_cross_cultural_practices |
| conduct_qualitative_interviews |

Table 192: (Part 49 of 52) 3266 skills extracted from topics in Tables 90, 92, and 94.

**Skills**

| |
|---|
| apply_anthropological_theories |
| map_cultural_networks |
| interpret_nonverbal_communication |
| examine_historical_contexts |
| identify_cultural_trends |
| interpret_cultural_narratives |
| apply_cultural_relativism |
| formulate_optimization_problems |
| implement_nonlinear_optimization_algorithms |
| use_gradient_descent_methods |
| analyze_convex_optimization_problems |
| utilize_dynamic_programming_approaches |
| develop_multi-objective_optimization_models |
| perform_sensitivity_analysis |
| use_quadratic_programming_techniques |
| solve_network_optimization_problems |
| develop_particle_swarm_optimization_algorithms |
| utilize_simulated_annealing_techniques |
| implement_branch_and_bound_algorithms |
| tactical_analysis |
| resource_allocation |
| logistics_planning |
| scenario_modeling |
| communication_coordination |
| force_deployment_optimization |
| psychological_warfare_planning |
| geopolitical_assessment |
| contingency_planning |
| strategic_forecasting |
| cyber_warfare_strategy |
| diplomatic_negotiation |
| morale_management |
| recognize_cultural_biases |
| analyze_linguistic_patterns |
| assess_cultural_values |
| synthesize_cultural_information |
| analyze_material_culture |
| evaluate_cultural_adaptation |
| mathematical_optimization_methods |
| apply_linear_programming_techniques |
| solve_constrained_optimization_problems |
| apply_integer_programming_techniques |
| implement_metaheuristic_algorithms |
| apply_stochastic_optimization_methods |
| implement_genetic_algorithms |
| apply_interior_point_methods |
| perform_lagrangian_relaxation |
| military_strategy_development |
| threat_assessment |
| terrain_evaluation |
| intelligence_gathering |
| risk_management |
| decision_making_under_pressure |
| technology_integration |
| historical_battle_analysis |
| alliance_building |
| war_game_simulation |
| operational_planning |
| asymmetric_warfare_tactics |
| supply_chain_management |
| military_history_analysis |

Table 194: (Part 50 of 52) 3266 skills extracted from topics in Tables 90, 92, and 94.

| Skills |
| --- |
| analyze_battle_tactics |
| interpret_historical_documents |
| assess_logistical_challenges |
| analyze_geopolitical_influences |
| evaluate_military_intelligence_operations |
| analyze_war_economics |
| assess_cultural_impacts_on_warfare |
| analyze_military_organizational_structures |
| evaluate_weapons_effectiveness |
| assess_psychological_warfare_techniques |
| evaluate_military_training_methods |
| language_acquisition_techniques |
| vocabulary_memorization |
| pronunciation_improvement |
| writing_proficiency_development |
| language_immersion_implementation |
| spaced_repetition_application |
| accent_reduction_techniques |
| translation_exercise_completion |
| audio_visual_material_utilization |
| goal_setting_and_tracking |
| metacognitive_strategy_application |
| machine_learning_implementation |
| data_preprocessing |
| model_evaluation_and_validation |
| computer_vision_development |
| deep_learning_framework_usage |
| data_visualization |
| ai_research_methodology |
| ai_system_integration |
| ai_model_deployment |
| ai_security_implementation |
| evaluate_strategic_decisions |
| compare_military_technologies |
| identify_key_military_leaders |
| reconstruct_historical_battles |
| interpret_military_maps |
| compare_military_doctrines |
| interpret_military_communications |
| analyze_military_alliances |
| interpret_military_symbolism |
| active_listening_practice |
| grammar_pattern_recognition |
| reading_comprehension_enhancement |
| conversation_practice_engagement |
| mnemonic_device_utilization |
| language_learning_app_navigation |
| idiomatic_expression_mastery |
| language_exchange_participation |
| self_assessment_conducting |
| cultural_context_understanding |
| artificial_intelligence_development |
| neural_network_design |
| feature_engineering |
| natural_language_processing |
| reinforcement_learning_application |
| algorithm_optimization |
| ethical_ai_implementation |
| ai_project_management |
| ai_performance_tuning |
| ai_explainability_techniques |
| ai_hardware_optimization |

Table 196: (Part 51 of 52) 3266 skills extracted from topics in Tables 90, 92, and 94.

| Skills |
| --- |
| ai_software_development |
| ai_documentation_and_reporting |
| ai_model_versioning_and_tracking |
| vehicle_architecture_design |
| electrical_system_optimization |
| automotive_safety_engineering |
| vehicle_diagnostics_implementation |
| vehicle_network_communication |
| fuel_efficiency_optimization |
| vehicle_performance_testing |
| ergonomics_and_human_factors_integration |
| automotive_manufacturing_process_design |
| automotive_systems_integration |
| automotive_cybersecurity_implementation |
| automotive_sensor_integration |
| automotive_quality_control_management |
| crop_rotation_planning |
| irrigation_system_design |
| harvest_timing_optimization |
| sustainable_farming_practices |
| farm_equipment_operation |
| organic_certification_compliance |
| food_processing_techniques |
| farm_financial_planning |
| precision_agriculture_implementation |
| vertical_farming_techniques |
| farm_to_table_logistics |
| climate_adaptive_farming |
| food_preservation_methods |
| network_configuration |
| data_center_management |
| cloud_infrastructure_deployment |
| ai_testing_and_quality_assurance |
| ai_data_collection_and_annotation |
| automotive_system_engineering |
| powertrain_integration |
| chassis_dynamics_analysis |
| thermal_management_design |
| automotive_software_development |
| noise_vibration_harshness_mitigation |
| emissions_control_system_design |
| automotive_materials_selection |
| vehicle_aerodynamics_optimization |
| vehicle_durability_testing |
| autonomous_driving_system_development |
| vehicle_weight_reduction_strategies |
| vehicle_simulation_and_modeling |
| agriculture_and_food_production |
| soil_nutrient_management |
| pest_identification_and_control |
| food_safety_protocol_implementation |
| livestock_health_monitoring |
| greenhouse_management |
| crop_yield_forecasting |
| agricultural_data_analysis |
| seed_selection_and_breeding |
| post_harvest_handling |
| aquaponics_system_management |
| agricultural_waste_management |
| agritourism_program_development |
| computer_infrastructure_management |
| server_administration |
| virtualization_implementation |
| system_monitoring |

Table 198: (Part 52 of 52) 3266 skills extracted from topics in Tables 90, 92, and 94.

| Skills |
| --- |
| performance_optimization |
| backup_and_restoration |
| patch_management |
| load_balancing_configuration |
| hardware_troubleshooting |
| infrastructure_documentation |
| cost_optimization |
| automation_scripting |
| power_management |
| inventory_tracking |
| service_level_agreement_monitoring |
| infrastructure_scalability_planning |
| biological_science_research |
| data_analysis |
| scientific_writing |
| cell_culture |
| pcr_amplification |
| bioinformatics_analysis |
| grant_writing |
| scientific_presentation |
| animal_handling |
| western_blotting |
| gene_expression_analysis |
| data_visualization |
| scientific_collaboration |
| disaster_recovery_planning |
| security_policy_enforcement |
| capacity_planning |
| storage_management |
| software_deployment |
| vendor_management |
| compliance_management |
| remote_access_configuration |
| cable_management |
| change_management_implementation |
| incident_response_coordination |
| lifecycle_management |
| experimental_design |
| laboratory_techniques |
| literature_review |
| dna_sequencing |
| protein_purification |
| statistical_modeling |
| research_ethics_compliance |
| sample_preparation |
| molecular_cloning |
| flow_cytometry |
| hypothesis_formulation |
| lab_safety_protocols |

Table 200: (Part 1 of 2) 13 Query/Task Types extracted from interactions with Claude-3.5-Sonnet

| Query Type | Description |
|---|---|
| Information-seeking | Queries aimed at acquiring factual knowledge or understanding about a specific topic |
| Help/Problem-solving | Queries seeking assistance with specific issues or challenges that need resolution |
| Verification/Validation | Queries asking to check the accuracy, correctness, or appropriateness of something |
| Opinion/Recommendation | Queries seeking subjective advice or suggestions based on preferences or criteria |
| Clarification | Queries asking for explanation of concepts, terms, or distinctions between related things |
| Exploratory/Learning | Open-ended queries driven by curiosity without a specific information target |
| Creative/Generative | Queries requesting the creation of new content or ideas |

Table 202: (Part 2 of 2) 13 Query/Task Types extracted from interactions with Claude-3.5-Sonnet

| Query Type | Description |
| --- | --- |
| Analysis | Queries asking for detailed examination or evaluation of content, data, or situations |
| Social/Emotional | Queries seeking conversation, emotional support, or human-like interaction |
| Hypothetical/Scenario Analysis | Queries exploring potential outcomes or consequences of theoretical situations |
| Comparative | Queries seeking evaluation of differences and similarities between options |
| Procedural/How-to | Queries seeking step-by-step instructions or methodical guidance |
| Planning/Strategy | Queries seeking structured approaches to achieve specific goals |

# L  SKILL EXTRACTION PROMPTS

## L.1  PROMPT FOR SKILL EXTRACTION (INSTRUCT-SKILLMIX-D)

```
Consider the following question.  Label this question with a skill that would be required
to solve the question.  Basically, you should be able to use the skill as a dictionary
key in python.  The skill name should be lower case letters only.  The skill name should
be very descriptive and you may use multiple words to describe the skills required in the
question.  If you do use multiple words per question, then join them by an underscore.
{text}
Your answer should be as follows:
<name of the skill>, reason:  <reason for the skill>
```

## L.2  PROMPT FOR SEMANTIC CLUSTERING (INSTRUCT-SKILLMIX-D)

```
Given the list of skills required to solve various questions, your task is to categorize
these skills into descriptive and specific groups.  Each category should not only capture
the essence of the skills it includes but also reflect clear, distinct areas of expertise
or application.  Use terminology that is precise and specific to the tasks those skills
accomplish.  Categories should be narrow enough to provide meaningful insight into the
specialization they represent.  Format category names in lowercase, joining multiple
words with underscores.
For each category you create, provide a comprehensive rationale explaining:
- Why these particular skills are grouped together.
- How the category name specifically reflects the specialization and distinctiveness of
the skills.

### Skills List:
{skills_joined_str}

### Examples:
- Category name:  `web_development`
- Included skills:  html_css_design, javascript_interactivity, web_performance_optimization
- Rationale:  This category focuses specifically on the skills required to build and
optimize web interfaces, distinguishing it from broader software development.

- Category name:  `machine_learning_techniques`
- Included skills:  machine_learning_model_creation, understanding_algorithmic_fairness,
machine_learning_application_in_finance
- Rationale:  These skills are grouped under 'machine_learning_techniques' because
they directly relate to the development and application of machine learning models,
emphasizing specific use cases and ethical considerations, which are distinct from
general programming skills.

Please ensure your category names are informative, specific, and reflective of the skills
grouped under them.  This will aid in their practical utility and clarity.
```

## L.3  Prompt for Creating Synthetic Data (Instruct-SkillMix-D)

**Prompt 1**

```
I am creating a dataset of engaging human-AI interactions that demonstrate the
capabilities and value of AI models across diverse skills.  Please help me generate
high-quality examples for this dataset.

I will first give you a set of {num_skills} skills that are valuable to humans.

Next, you would create a query that a human might plausibly ask whose answer involves
using those {num_skills} skills.

For this query you will generate a good length answer using your best helpfulness and
wisdom.  Please try to avoid generic sound queries and answers.  People often have
concrete problems and would give important details.  The answer should also give some
concrete details and examples, but not get verbose.

Before I give you the set of {num_skills} skills, I will provide some examples involving 2
skills.
Example 1:
### Skills:  self_reflection_skills, linguistics
### Instruction:  I often find myself in conflicts at work due to miscommunication and
differing perspectives.  How can I improve my conflict resolution skills by incorporating
self-reflection and effective communication techniques into my daily interactions with
colleagues?
### Response:  To enhance your conflict resolution skills in the workplace, integrate
these self-reflection and linguistic strategies into your daily routine:
Self-Reflection Techniques:
End-of-Day Review:  Spend 10 minutes reflecting on interactions, noting successes and
areas for improvement.  For example, Ï successfully de-escalated a tense situation
with a client by actively listening to their concerns and finding a mutually beneficial
solution.¨
Conflict Log:  Record details of conflicts, your responses, and outcomes in a dedicated
journal.  Analyze patterns monthly to identify growth opportunities.  For instance, Ï
noticed I tend to become defensive when criticized.  I will practice pause and respond
techniques to improve my reactions.¨
Linguistic Techniques:
Empathy Statements:  Use phrases that demonstrate understanding and validate others'
perspectives, such as Ï can see why you feel that wayör Ïhat must be challenging
for you.Ïn a recent meeting, saying Ï understand your frustration with the project
timelineḧelped a coworker feel heard and open to finding a solution together.
Reflective Listening:  Paraphrase the other person's key points before responding to
ensure clarity and show you're listening.  For example, Šo, what I'm hearing is that
you're concerned about the budget allocation.  Let's discuss some options to address
that.¨
Integration into Routine:
Weekly Practice Sessions:  Dedicate an hour each week to role-play conflict scenarios
with a trusted colleague, focusing on applying self-reflection insights and linguistic
techniques.  Provide each other with constructive feedback for continuous improvement.
Real-Time Application:  When a conflict arises, take a moment to mentally review your
phrase bank and select an appropriate response.  After the interaction, briefly reflect
on what worked well and what could be improved next time.
By consistently practicing these strategies, you'll build stronger habits and enhance
your overall conflict resolution skills in the workplace.
Example 2:
### Skills:  psychological_understanding, data_handling_and_analysis
### Instruction:  I recently started working as a data analyst for a meal kit delivery
service.  We've noticed a significant increase (20%) in customer churn over the past
quarter.  How can I leverage data analysis and psychological principles to identify the
root causes of churn and develop targeted strategies to improve customer retention?
### Response:  To address the 20% increase in customer churn for your meal kit service,
employ a combination of data analysis and psychological insights:
Data Analysis:
Customer Segmentation:  Analyze churn rates by customer segments such as age, location,
and dietary preferences.  For example, if you find that vegetarian customers have a
higher churn rate, you can focus on improving vegetarian meal options.
Churn Timing Analysis:  Identify common timeframes when churn occurs, such as after the
first month or after specific promotions end.  If churn spikes after the first month,
consider implementing a more robust onboarding process or extending promotional offers.
Psychological Insights:
Exit Surveys:  Conduct exit surveys to gather direct feedback from churning customers.
Include open-ended questions to uncover deeper insights.  For instance, Ẅhat was the
primary reason for canceling your subscription?änd Ẅhat could we have done better to keep
you as a customer?¨
Customer Journey Mapping:  Map out the customer journey to identify pain points and
opportunities for improvement.  Apply psychological principles like the Peak-End Rule
to prioritize the most impactful moments.  Focus on creating a positive onboarding
experience and ensuring a satisfying last interaction before cancellation.
```

```
Actionable Strategies:
Personalized Retention Campaigns:  Develop targeted email campaigns for at-risk segments,
addressing their specific concerns.  For example, send vegetarian customers an email
highlighting new plant-based recipes and offering a discount on their next box.
```

```
Flexible Subscription Options:  Introduce more flexible subscription plans, such as the
ability to skip weeks or customize delivery frequency.  This addresses the psychological
need for autonomy and control, reducing the likelihood of churn.
By leveraging data-driven insights and psychological principles, you can create a
targeted approach to reduce churn and improve customer satisfaction in your meal kit
service.
Example 3:
### Skills:  food_related_knowledge_and_skills, internet_technologies
### Instruction:  I'm a food truck owner looking to expand my online presence and attract
more customers.  I want to create a website that allows customers to customize their
meals, track my truck's real-time location, and place orders for pickup.  What web
technologies should I consider to build a user-friendly and feature-rich website for
my food truck business?
### Response:  To create a robust and interactive food truck website with meal
customization and real-time location tracking, consider the following web technologies:
Front-End:
React:  Use React, a popular JavaScript library, for building a dynamic and responsive
user interface.  React's component-based architecture allows for easy management of
real-time updates, such as displaying the current location of the food truck.
HTML5 Geolocation API: Leverage the HTML5 Geolocation API to capture the user's location
(with their permission) and display nearby food truck locations on an interactive map.
Leaflet or Google Maps:  Integrate Leaflet, an open-source mapping library, or Google
Maps to display food truck locations and enable users to get directions.  Customize the
map markers to show additional information like menu items or special offers.
Back-End:
Node.js and Express:  Use Node.js, a server-side JavaScript runtime, and the Express
web application framework to handle server-side logic and API routing.  Node.js's
non-blocking I/O model is well-suited for real-time applications like location tracking.
WebSocket:  Implement WebSocket communication to enable real-time updates between the
server and the client.  When a food truck's location changes, the server can push the
updated coordinates to connected clients, ensuring users always see the most current
location.
Firebase Realtime Database:  Store and sync food truck location data in real-time using
Firebase Realtime Database.  This NoSQL cloud database allows for efficient storage and
retrieval of location data, enabling seamless real-time updates across multiple devices.
User Experience:
Progressive Web App (PWA): Develop the website as a PWA to provide a native app-like
experience.  PWAs offer features like offline accessibility, push notifications, and home
screen installation, enhancing user engagement and convenience.
Responsive Design:  Ensure the website is fully responsive and optimized for various
screen sizes, particularly mobile devices.  Use CSS media queries and flexible layouts to
create a seamless experience across desktops, tablets, and smartphones.
Accessibility:  Follow web accessibility guidelines (WCAG) to ensure the website is
usable by people with disabilities.  Implement proper headings, alt text for images, and
keyboard navigation to create an inclusive user experience.
By combining these web technologies, you can create a feature-rich and user-friendly food
truck website that offers meal customization and real-time location tracking, providing a
seamless experience for your customers.
Example 4:
### Skills:  algorithmic_and_programming_skills, digital_marketing
### Instruction:  As a software engineer, I've developed a sophisticated recommendation
engine that I believe could greatly benefit e-commerce businesses.  I want to showcase
my work through a technical blog post and live demo to attract potential clients and
demonstrate the value of my solution.  How can I structure my blog post and demo to
effectively highlight the technical aspects and business benefits of my recommendation
engine?
### Response:  To effectively showcase your recommendation engine and attract potential
e-commerce clients, structure your technical blog post and live demo as follows:
Blog Post Structure:
Attention-Grabbing Introduction:  Start with a compelling hook that highlights the value
of personalized recommendations in e-commerce.  Use statistics or real-world examples to
demonstrate the impact on sales and customer satisfaction.
Problem Statement:  Clearly define the challenges e-commerce businesses face in providing
relevant product recommendations at scale.  Discuss common pain points like data sparsity,
cold-start problems, and real-time updates.
Technical Deep Dive:  Explain the core components of your recommendation engine, such as
collaborative filtering, content-based filtering, or hybrid approaches.  Use diagrams
and code snippets to illustrate your architecture and key algorithms.  Highlight any
innovative techniques you've employed, such as deep learning or reinforcement learning.
Performance Metrics:  Present quantitative results that showcase the effectiveness of
your recommendation engine.  Include metrics like precision, recall, F1 score, and mean
average precision.  Compare your results to industry benchmarks or popular open-source
recommendation libraries to demonstrate your engine's superiority.
Scalability and Efficiency:  Discuss how your recommendation engine handles large-scale
```

data and real-time updates. Explain your strategies for efficient data processing, such as parallel computing or incremental updates. Provide performance benchmarks to highlight the speed and scalability of your solution.
Live Demo: E-commerce Store Integration: Create a mock e-commerce store that seamlessly integrates your recommendation engine. Showcase personalized product recommendations based on user interactions, such as viewed items, purchases, or ratings.

Real-Time Recommendations: Demonstrate how your engine adapts in real-time as users navigate the store. For example, show how the recommendations update dynamically based on the user's browsing history or cart contents.
Explanations and Transparency: Provide clear explanations for each recommendation, such as C̈ustomers who bought this item also bought...ör R̈ecommended based on your recent searches.T̈his transparency builds trust and helps users understand the relevance of the recommendations.
A/B Testing: Implement an A/B testing feature that allows potential clients to compare the performance of your recommendation engine against a default or random recommendation system. Visualize the results through metrics like click-through rates or conversion rates to highlight the tangible benefits of your solution.
Promotion and Engagement:
Targeted Social Media Ads: Create compelling social media ads that highlight the key features and benefits of your recommendation engine. Target decision-makers in the e-commerce industry, such as marketers, product managers, and CTOs. Use eye-catching visuals and clear calls-to-action to drive traffic to your blog post and live demo.
LinkedIn Thought Leadership: Share your blog post on LinkedIn and engage in discussions related to e-commerce personalization. Participate in relevant groups and forums to establish yourself as a thought leader in the field. Offer valuable insights and answer questions to build trust and credibility with potential clients.
Webinars and Workshops: Host webinars or workshops that provide a hands-on experience with your recommendation engine. Walk participants through the setup process and demonstrate how to integrate your solution into their existing e-commerce platforms. Offer Q&A sessions to address specific concerns and showcase your expertise.
By structuring your technical blog post and live demo in this manner, you can effectively showcase the capabilities and benefits of your recommendation engine. Combine this with targeted digital marketing techniques to reach and engage potential e-commerce clients, ultimately driving interest and adoption of your solution. Example 5:
### Skills: geographical_and_historical_knowledge, analytical_thinking
### Instruction: I'm a history enthusiast planning a two-week trip to Eastern Europe with my family. We're particularly interested in exploring medieval architecture and learning about the region's experiences during the Cold War era. How can I create an itinerary that combines visits to historical sites, museums, and cultural experiences to gain a deeper understanding of Eastern Europe's past and present?
### Response: Here's a curated itinerary for a historically rich two-week trip through Eastern Europe, focusing on medieval and Cold War sites:
Week 1: Poland and Czech Republic
Krakow (3 days):
Wawel Castle: Explore this 14th-century castle complex, which served as the residence of Polish kings. Marvel at the Gothic and Renaissance architecture, and visit the Dragon's Den, a legendary cave beneath the castle.
Oskar Schindler's Factory Museum: Learn about the Holocaust and Oskar Schindler's efforts to save Jewish workers during World War II. The museum offers a immersive experience, recreating the wartime atmosphere of Krakow.
Nowa Huta: Take a guided tour of this planned socialist city, built during the Cold War era. Visit the iconic Lord's Ark Church, which became a symbol of resistance against the communist regime.
Warsaw (2 days):
Old Town: Stroll through the meticulously reconstructed Old Town, which was destroyed during World War II. Visit the Market Square, the Warsaw Barbican, and St. John's Cathedral to admire the Gothic and Renaissance architecture.
Warsaw Uprising Museum: Discover the story of the 1944 Warsaw Uprising, a major resistance operation against Nazi occupation. The museum features interactive exhibits, original artifacts, and a 3D movie that brings history to life.
Palace of Culture and Science: Explore this imposing Stalinist-era skyscraper, which remains the tallest building in Poland. Take an elevator to the observation deck for panoramic views of Warsaw.
Prague (2 days):
Prague Castle: Visit the world's largest ancient castle complex, dating back to the 9th century. Explore the Gothic St. Vitus Cathedral, the Romanesque St. George's Basilica, and the Golden Lane, a picturesque row of colorful houses.
Charles Bridge: Walk across this iconic 14th-century stone bridge, lined with baroque statues. Enjoy street musicians and artists, and take in the stunning views of the Vltava River and the Old Town.
Nuclear Bunker Tour: Experience the chilling reality of the Cold War by visiting a former Soviet nuclear bunker. Learn about the strategies and technology employed during this tense period in history.
Week 2: Hungary and Romania
Budapest (3 days):
Buda Castle: Discover this 13th-century castle, which has been rebuilt and expanded over the centuries. Visit the Hungarian National Gallery, the Budapest History Museum, and the Matthias Church, known for its colorful tiled roof.

```
House of Terror Museum:  Explore this powerful museum dedicated to the victims of
the fascist and communist regimes in Hungary.  The exhibits are housed in the former
headquarters of the Arrow Cross Party and the communist secret police.
Memento Park:  Visit this open-air museum showcasing monumental statues from the Soviet
era.  Learn about the propaganda and ideology behind these imposing sculptures.
Bucharest (2 days):
Palace of Parliament:  Tour the world's largest civilian building, constructed during the
communist era under the rule of Nicolae Ceaușescu.
```

```
Marvel at the opulent interiors and learn about the controversial history of this massive
structure.
Old Town:  Explore the charming streets of Bucharest's Old Town, lined with historical
buildings, churches, and cafes.  Visit the ruins of the Old Princely Court, which served
as the residence of Wallachian princes.
Revolution Square:  Pay tribute to the heroes of the 1989 Romanian Revolution at this
significant square.  See the Memorial of Rebirth, which honors those who lost their lives
fighting against the communist regime.
Travel Tips:
Book guided tours with knowledgeable local guides to gain deeper insights into the
historical context and personal stories behind each site.
Stay in centrally located accommodation to minimize travel time and maximize your
exploration of each city.
Use public transportation or ride-sharing services to navigate within cities, as parking
can be challenging and expensive in historical areas.
By following this itinerary, you'll embark on a fascinating journey through Eastern
Europe's medieval and Cold War past, gaining a profound appreciation for the region's
rich history and resilience.

Now, it is your turn! Given the guidelines and examples above, please create a query
that a human might plausibly ask whose answer involves using all skills below:
### Skills: {skills_str}
```

**Prompt 2**

```
Thanks!  It looks like the response may have been cut off.  Please rewrite the entire
response, making sure it fits within the length constraint.  No need to include verbose
or extraneous information.  Thanks!
```

**Prompt 3**

```
Pretend you are the human asking the question, and make a list of weaknesses and
strengths of the response.  Currently the response looks a bit generic.  Perhaps it
could benefit from some concrete examples and details.
```

**Prompt 4**

```
Now, please refine your instruction and response (keep the strong parts and work on the
weak parts).
```

**Prompt 5**

```
It looks like the improved response may have been cut off.  Please rewrite the entire
improved response, making sure it fits within the length constraint.  No need to include
verbose or extraneous information.  Thanks again!
```

### L.4   PROMPT FOR EXTRAPOLATING SKILLS (INSTRUCT-SKILLMIX)

```
First, consider the following definitions:
- Knowledge:  A set of information, facts, descriptions, and theories acquired through
education or experience.
- Topic:  A subset of Knowledge that focuses on a particular area or subject.
- Skill:  A function that transforms Knowledge into actions or performances that achieve
specific outcomes.

Now, for the topic of topic, please provide me with multiple skills (in snake case) that
could be relevant to the topic.
```

## L.5 PROMPT FOR CREATING SYNTHETIC DATA (INSTRUCT-SKILLMIX)

**Prompt 1**

```
I am creating a dataset of engaging human-AI interactions that demonstrate the
capabilities and value of AI models across diverse skills.  Please help me generate
high-quality examples for this dataset.

I will first give you a set of {num_skills} skills that are valuable to humans.

Next, you would create a query that a human might plausibly ask whose answer involves
using those {num_skills} skills.

For this query you will generate a good length answer using your best helpfulness and
wisdom.  Please try to avoid generic sound queries and answers.  The answer should also
give some concrete details and examples, but not get verbose.

Before I give you the set of {num_skills} skills, I will provide some examples involving 2
skills.
Example 1:
### Task Type:  Information-Seeking
### Skills:  psychological_understanding, data_handling_and_analysis
### Instruction:  What are the primary psychological theories that explain the impact
of social media on adolescent self-esteem, and what evidence supports these theories?
Additionally, what statistical techniques have been used in the literature to dissect
these relationships, and how do these techniques help in understanding the nuances of
psychological data?
### Response:  The impact of social media on adolescent self-esteem is a widely studied
topic within psychology.  Various theories and statistical techniques are employed to
understand and measure these effects.  Below, I'll outline some of the primary theories,
the supporting evidence, and the statistical techniques used in this research.
### Response:  The impact of social media on adolescent self-esteem is a widely studied
topic within psychology.  Various theories and statistical techniques are employed to
understand and measure these effects.  Below, I'll outline some of the primary theories,
the supporting evidence, and the statistical techniques used in this research.

**Psychological Theories and Evidence**

Social Comparison Theory:  Adolescents often assess their worth by comparing themselves
with peers on social media.  A study by Vogel et al.  (2014) found that Facebook usage
was associated with negative self-perceptions when users compared themselves to others
they perceived as better off.  This suggests a direct correlation between the frequency
of social media use and increased feelings of inferiority.

Self-Presentation Theory:  Adolescents feel pressure to present themselves in an
idealized way on platforms like Instagram and Snapchat.  Research by Chou and Edge (2012)
highlights that the need for positive feedback and fear of negative evaluations can lead
to cycles of anxiety and lowered self-esteem when these expectations are not met.

Media Richness Theory:  Different types of social media platforms impact self-esteem
differently.  A study by Sherman et al.  (2016) compared text-based versus image-based
platforms, finding that image-based platforms (like Instagram) can exacerbate body image
concerns and anxiety more than text-based platforms due to the intense focus on physical
appearance.

Statistical Techniques and Applications

Regression Analysis:  Used extensively to quantify the relationship between social
media usage and self-esteem.  For instance, studies might control for variables like
age, gender, and socioeconomic status to isolate the effect of social media usage.

Structural Equation Modeling (SEM): SEM enables researchers to construct a comprehensive
model of how different factors like peer influence, media consumption, and personal
traits interact to affect self-esteem.  An example is Kross et al.'s (2013) study, which
modeled the pathways through which social media affects mood and self-esteem over time.

Longitudinal Studies and Growth Curve Modeling:  These techniques provide insights
into how self-esteem trajectories change over time with social media use.  For instance,
longitudinal research might track self-esteem changes from early to late adolescence,
revealing critical periods where impacts are most pronounced.

Mediation and Moderation Analyses:  These are crucial for exploring underlying mechanisms
and boundary conditions.  For example, studies might explore how the negative impact of
social media on self-esteem can be mediated by social support or moderated by individual
resilience.

**Contradictory Evidence**

While many studies suggest a negative impact of social media on self-esteem, some
```

```
research offers a different perspective. For instance, Valkenburg et al. (2006)
found that online interactions can enhance well-being by providing social support and
a platform for self-expression. These findings suggest that the impact of social media
on self-esteem is not universally negative and can vary based on usage patterns and
individual differences.

**Practical Implications**

Understanding these dynamics is crucial for developing targeted interventions. For
instance, educational programs that teach critical media literacy can help adolescents
navigate social media healthily. Moreover, knowing which aspects of social media use
trigger negative self-comparisons can guide the development of features that promote
positive interaction and reduce the emphasis on appearance.

Example 2:
### Task Type: Help-Seeking
### Skills: self_reflection_skills, linguistics
### Instruction: I often find myself in conflicts at work due to miscommunication and
differing perspectives. How can I improve my conflict resolution skills by incorporating
self-reflection and effective communication techniques into my daily interactions with
colleagues?
### Response: To enhance your conflict resolution skills in the workplace, integrate
these self-reflection and linguistic strategies into your daily routine:
Self-Reflection Techniques:
End-of-Day Review: Spend 10 minutes reflecting on interactions, noting successes and
areas for improvement. For example, Ï successfully de-escalated a tense situation
with a client by actively listening to their concerns and finding a mutually beneficial
solution.¨
Conflict Log: Record details of conflicts, your responses, and outcomes in a dedicated
journal. Analyze patterns monthly to identify growth opportunities. For instance, Ï
noticed I tend to become defensive when criticized. I will practice pause and respond
techniques to improve my reactions.¨
Linguistic Techniques:
Empathy Statements: Use phrases that demonstrate understanding and validate others'
perspectives, such as Ï can see why you feel that wayör Ïhat must be challenging
for you.Ïn a recent meeting, saying Ï understand your frustration with the project
timelineḧelped a coworker feel heard and open to finding a solution together.
Reflective Listening: Paraphrase the other person's key points before responding to
ensure clarity and show you're listening. For example, S̈o, what I'm hearing is that
you're concerned about the budget allocation. Let's discuss some options to address
that.¨
Integration into Routine:
Weekly Practice Sessions: Dedicate an hour each week to role-play conflict scenarios
with a trusted colleague, focusing on applying self-reflection insights and linguistic
techniques. Provide each other with constructive feedback for continuous improvement.
Real-Time Application: When a conflict arises, take a moment to mentally review your
phrase bank and select an appropriate response. After the interaction, briefly reflect
on what worked well and what could be improved next time.
By consistently practicing these strategies, you'll build stronger habits and enhance
your overall conflict resolution skills in the workplace.

Example 3:
### Task Type: Information-Seeking
### Skills: geographical_and_historical_knowledge, analytical_thinking
### Instruction: What were the major geopolitical changes in Europe following World War
II, and how have these changes influenced current European political boundaries?
### Response: **Major Geopolitical Changes in Europe After World War II**

1. Division of Germany: The division of Germany into East and West Germany in 1949
was a critical symbol of the Cold War's onset. This division, marked physically by
the Berlin Wall from 1961, lasted until the reunification in 1990 and significantly
influenced European political dynamics.

2. Creation of the Iron Curtain: Winston Churchill's term, the Ïron Curtain,¨described
the ideological and physical boundary separating the Soviet-controlled Eastern Bloc
and the Western democracies. This division dictated political alignments and conflicts
during the Cold War, influencing policies and international relations.

3. Decolonization: Following the war, European powers like Britain and France began
a significant process of decolonization. This period saw the independence of nations
such as India in 1947 and the rapid decolonization of Africa in the 1960s, marking a
major shift in global dynamics and European foreign policies.

4. Formation of the European Union (EU): The EU's origins lie in the European Coal and
Steel Community in 1951, evolving into the European Economic Community by 1957. These
alliances, expanding to include more countries over the decades, aimed to foster economic
cooperation and prevent further wars in Europe, influencing both economic and political
policies within the continent.

5. NATO and the Warsaw Pact: The establishment of NATO in 1949 by Western countries
```

```
was a strategic move for collective security against the Soviet threat.  The Soviet
response, the Warsaw Pact in 1955, defined the military alliances in Europe, solidifying
the East-West divide.

**Influence on Current European Political Boundaries**

1.  German Reunification:  The fall of the Berlin Wall in 1989 and the subsequent
reunification of East and West Germany in 1990 reshaped Germany's role in Europe,
altering both its internal and external political boundaries.

2.  EU Expansion:  The EU's expansion has included many former Eastern Bloc countries,
fundamentally changing the political landscape of Europe.  The Schengen Agreement,
implemented in 1995, minimized the importance of national boundaries within the EU,
promoting free movement and economic integration.

3.  Breakup of Yugoslavia and the Soviet Union:  The disintegration of Yugoslavia into
seven successor states throughout the 1990s and the Soviet Union into 15 independent
countries in 1991 dramatically redrew political boundaries.  These events, rooted in
ethnic tensions and political upheavals, continue to influence regional stability and
alignments.

**Case Studies:  Key Treaties and Shifts in Alliances**

1.  Treaty of Paris (1951):  This treaty established the European Coal and Steel
Community, a foundational step towards European integration.  It set precedents for
future economic policies and cooperative frameworks within Europe, promoting peace and
economic stability across former wartime adversaries.

2.  Impact of NATO and the Warsaw Pact on Poland:  Poland's transition from a Warsaw Pact
member to a NATO member in 1999 exemplifies the dramatic shift in military and political
alliances.  This change significantly influenced Poland's foreign and security policies,
reflecting broader geopolitical shifts in Eastern Europe post-Cold War.

Now, it is your turn!  Given the guidelines and examples above, please create a query
that aligns with the following task type:
{query_type}

Include specific text snippets when they are essential to analyze and validate your query
effectively.  Make sure that the query is one that a human might plausibly ask whose
answer involves using all skills below:
### Task Type:  {query_str}
### Skills:  {skills_str}
```

## Prompt 2

```
Thanks!  It looks like the response may have been cut off.  Please rewrite the entire
response, making sure it fits within the length constraint.  No need to include verbose
or extraneous information.  Thanks!
```

## Prompt 3

```
Pretend you are the human asking the question, and make a list of weaknesses and
strengths of the response.  Currently the response looks a bit generic.  Perhaps it
could benefit from some concrete examples and details.
```

## Prompt 4

```
Now, please refine your instruction and response (keep the strong parts and work on the
weak parts).
```

## Prompt 5

```
It looks like the improved response may have been cut off.  Please rewrite the entire
improved response, making sure it fits within the length constraint.  No need to include
verbose or extraneous information.  Thanks again!
```

# M    COMPARISON OF RESPONSES

In this section, we show side-by-side comparisons of responses from various models to the following instruction from AlpacaEval.

> I'm trying to teach myself to have nicer handwriting.  Can you help?

Observe that responses produced by models finetuned on INSTRUCT-SKILLMIX tend to include concrete examples (whereas responses produced by the other models do not always).

### Response by LLaMA-3-8B-Base SFT on INSTRUCT-SKILLMIX(k=2)-4K

```
Absolutely!  Improving your handwriting involves both technique and practice.  Here's a
concise guide to help you achieve nicer handwriting:

**1.  Posture and Grip:**
- **Posture:** Sit upright with your paper at a 45-degree angle to reduce arm and hand
movement, which helps prevent smudging.
- **Grip:** Use a relaxed, comfortable grip with your pen or pencil resting lightly in
your fingers.  This promotes smoother movement and reduces pressure on the paper.

**2.  Pen Lifts:**
- **When to Lift:** Between words and sentences to create clear breaks, especially in
cursive.
- **Purpose:** Pen lifts help maintain the integrity of each letter, preventing them from
blending together and enhancing readability.

**3.  Slant and Direction:**
- **Natural Slant:** Write with a natural slant that's comfortable for your hand,
typically a slight upward slope helps legibility.
- **Direction:** Write continuously without lifting your pen too much, maintaining a fluid
motion can enhance the flow and elegance of your handwriting.

**4.  Pressure and Speed:**
- **Pressure:** Apply consistent pressure, avoiding excessive force which can cause
fatigue and messy handwriting.
- **Speed:** Write at a comfortable pace that allows your handwriting to remain clear and
legible.

**5.  Regular Practice:**
- **Routine:** Set aside time daily for handwriting practice, such as 10 minutes a day
writing sentences or paragraphs.
- **Variety:** Mix practice sessions with both structured exercises focusing on specific
letters and free writing to maintain fluidity.

**6.  Paper and Tools:**
- **Paper Choice:** Use smooth, medium-weight paper that allows for control without too
much slip.
- **Tool Variety:** Experiment with different writing tools|pens, pencils, markers|to find
which provides the best feel and outcome for your handwriting.

**7.  Evaluation and Adjustment:**
- **Self-Review:** Regularly assess your handwriting samples to note improvements and
areas needing more work.
- **Feedback:** Ask for feedback from others or enroll in a calligraphy class to gain
insights from experienced instructors.

By focusing on these specific elements and committing to regular practice, your
handwriting will improve.  Remember, small, consistent efforts can lead to significant
improvements over time.

This response is concise yet informative, providing specific, actionable advice for
each step of the handwriting improvement process.  It also includes practical tips and
suggestions for maintaining motivation and evaluating progress, making it a comprehensive
guide for anyone looking to refine their handwriting.
```

**Response by Mistral-7B-Base-v0.2 SFT on INSTRUCT-SKILLMIX(k=2)-4K**

```
Improving your handwriting is a rewarding skill to develop and can be both therapeutic
and practical.  Here's a structured guide with specific techniques and exercises to help
you achieve elegant, legible handwriting:

**Step 1:  Posture and Setup**
- **Posture**:  Sit with a straight spine and relax your shoulders to prevent tension.
- **Light**:  Use a desk lamp for adequate lighting.
- **Paper and Pen**:  Use smooth, lined paper and a pen grip that feels comfortable, such
as a Pilot G2 or a Pentel EnerGel.

**Step 2:  Warm-up and Mindfulness**
- **Warm-up**:  Write the alphabet three times to warm up muscles.
- **Breathing**:  Inhale for four counts, exhale for four counts; repeat until calm.  This
helps reduce writing tension.

**Step 3:  Basic Strokes**
- Practice basic strokes:  the upward stroke (v), downward stroke (∩), and circular stroke
(o).

**Step 4:  Letter Formation**
- **Uppercase Letters**:  Form each letter slowly, tracing them with a stencil for
accuracy.
- **Lowercase Letters**:  Practice looping 'l' and crossing 't'.

**Step 5:  Word Practice**
- Write simple sentences like \The quick brown fox jumps over the lazy dog." Focus on each
word.

**Step 6:  Cursive Writing**
- Start with connected uppercase letters, then move to fluid cursive lowercase.
- Try online tutorials for modeled cursive formation.

**Step 7:  Pressure and Speed**
- Practice on a tablet to see pressure applied; aim for consistent pressure.
- Slow down; writing fast increases errors.

**Step 8:  Style Inspirations**
- Explore calligraphy styles online, imitating specific fonts like Script or Gothic.

**Step 9:  Regular Practice**
- Write a journal entry daily, focusing on quality, not quantity.

**Step 10:  Challenges and Variety**
- Weekly challenges like writing with your non-dominant hand or different tools.
- Experiment with bullet journaling for creativity.

**Step 11:  Track Progress**
- Monthly before-and-after writing samples.
- Ask for feedback from a calligraphy expert or mentor.

**Step 12:  Enjoy the Process**
- Listen to calming music or an audiobook to enjoy the time writing.

**Additional Tips:**
1.  **Muscle Training**:  Use hand exercise tools or write with silverware for grip
strength.
2.  **Mindfulness**:  Meditate before writing to enhance focus.
3.  **Rewards**:  Treat yourself after achieving goals, like a new pen for milestones.

This guide not only sharpens your handwriting but also promotes mindfulness and
creativity, offering both functional and mental benefits.  Enjoy the journey to better
handwriting!
```

**Response by the Reference Model (GPT-4-Turbo 2023-11-06)**

> Certainly! Improving your handwriting involves patience, practice, and attention to technique. Here are some tips to help you develop nicer handwriting:
> 1. **Understand Your Current Style**: Before you can improve your handwriting, you need to understand where you're starting from. Write a paragraph and analyze it to see where you could make improvements. Look for consistent slant, letter size, spacing, and legibility.
> 2. **Choose the Right Tools**: Use a pen or pencil that feels comfortable in your hand. The right tool can make a significant difference in your writing. Some people find that using a fountain pen improves their writing because it encourages a more relaxed grip and has a smoother ink flow.
> 3. **Practice Basic Strokes**: Before focusing on letters, practice basic strokes. This includes straight lines, curves, and circles. These are the building blocks of letters, and getting comfortable with these will make your letters more consistent.
> 4. **Slow Down**: Speed can lead to sloppy writing. By slowing down, you can focus on the shape of each letter and ensure that you're writing them correctly.
> 5. **Maintain Proper Posture and Grip**: Sit up straight and hold your pen or pencil with a relaxed grip. Don't hold it too tightly, as this can make your handwriting appear more jagged and can also cause your hand to tire out more quickly.
> 6. **Form Letters Correctly**: Start and end each letter in the right spot to ensure consistency and flow. Practice writing each letter of the alphabet, focusing on the ones that give you the most trouble.
> 7. **Practice Consistent Letter Size and Spacing**: Consistency is key to neat handwriting. Work on keeping all of your letters the same size and your words evenly spaced. You could use lined paper as a guide.
> 8. **Use Handwriting Worksheets**: You can find handwriting practice worksheets online for both print and cursive writing. These worksheets can guide you through the proper strokes and help you practice consistency.
> 9. **Copy Neat Writing**: Find a sample of handwriting you admire and try to copy it. This can help you develop a sense of the style you're aiming for.
> 10. **Write Often**: Like any skill, handwriting improves with regular practice. Find reasons to write by hand every day, whether it's journaling, writing letters, or making to-do lists.
> 11. **Analyze and Adjust**: Regularly look back at your handwriting to see where you've improved and what still needs work. Make adjustments as needed.
> 12. **Be Patient and Persistent**: Don't get discouraged if progress is slow. Handwriting improvement takes time and dedication. Remember, the goal of improving your handwriting is to make it more legible and aesthetically pleasing, but it should also be comfortable for you to write. There's no one "correct" style of handwriting, so focus on developing a style that works for you."

**Response by UltraLM 13B**

```
Sure, I'd be happy to help!  Here are some tips to improve your handwriting:

1.  Practice regularly:  The more you practice, the better your handwriting will become.
Set aside some time every day to practice writing letters, words, and sentences.
2.  Start with the basics:  Begin by learning the basic strokes of each letter, such as
circles, lines, and curves.  Then practice writing each letter in isolation before moving
on to words and sentences.
3.  Use the right tools:  Make sure you have the right supplies, such as a good-quality
pen or pencil, lined paper, and an eraser.  Using the right tools can make a big
difference in the quality of your handwriting.
4.  Watch your posture:  Good posture is essential for good handwriting.  Sit up straight
with your feet flat on the floor and your arms and wrists relaxed.
5.  Relax your grip:  Hold your pen or pencil lightly and avoid gripping it too tightly.
This will help you write more smoothly and legibly.
6.  Take breaks:  It's important to take breaks while practicing to avoid fatigue and
strain.  Take a break every 20-30 minutes and stretch your hands and wrists.

Remember, improving your handwriting takes time and practice, so be patient with yourself.
Keep practicing and soon you'll have beautiful handwriting!
```

**Response by Mistral-7B-Base-v0.2 SFT on Alpaca-52K**

```
Sure! Here are some tips that can help you improve your handwriting:

1.  Practice regularly:  The more you practice, the better your handwriting will become.
Set aside some time each day to write out letters, words, and sentences to improve your
handwriting.

2.  Use the right tools:  Make sure you have a good pen or pencil that flows smoothly and
is comfortable to write with.  A good quality pen or pencil can make a big difference in
the quality of your writing.

3.  Keep your posture in mind:  Sit up straight and hold your pen or pencil correctly,
with your arm and wrist relaxed.  This will help you write more smoothly and avoid
cramping.

4.  Take your time:  Don't rush your writing.  Take your time to form each letter and
make sure it's legible.

5.  Use proper grip:  Hold your pen or pencil correctly, with your fingers and thumb
supporting it, and your index finger and middle finger guiding it.

6.  Keep your letters consistent:  Make sure your letters are the same size and have the
same slant.  This will make your writing look more uniform.

7.  Practice proper letter formation:  Make sure your letters are formed correctly, with
the right strokes and curves.  You can find examples of proper letter formation online or
in a handwriting guidebook.

8.  Keep your handwriting neat:  Keep your writing neat and tidy, with even spacing
between words and lines.

9.  Practice cursive writing:  Cursive writing can help improve your handwriting by
making your letters more connected and flowing.

10.  Experiment with different styles:  Try different styles of handwriting to find
one that suits you.  There are many styles to choose from, such as print, cursive, and
italic.

11.  Use guides:  You can use a ruler or a grid to help you keep your writing straight
and even.

12.  Get feedback:  Ask a friend or teacher to give you feedback on your writing and see
where you can improve.

13.  Keep it simple:  Avoid overly decorative or complicated writing, as it can be hard
to read.

14.  Keep it legible:  Make sure your writing is easy to read, even if it's not perfect.

15.  Be patient:  Improving your handwriting takes time and practice, so be patient with
yourself and keep practicing.

Remember, everyone's handwriting is unique, so don't be too hard on yourself if
it doesn't look exactly like someone else's.  Just keep practicing and you'll see
improvement over time.
```

**Response by Mistral-7B-Base-v0.2 SFT on Alpaca-1K Longest**

```
Sure!  Here are some tips that may help you improve your handwriting:

1.  Practice regularly:  Like any skill, improving your handwriting takes practice.  Set
aside some time each day to write out words, sentences or even whole passages to develop
fluency and consistency in your strokes.

2.  Use the correct tools:  Make sure you have a pen or pencil that feels comfortable in
your hand and that you sharpen it regularly to ensure a fine point.  Also, use paper that
is not too smooth or too rough, as this can affect the quality of your strokes.

3.  Posture:  Sit up straight and keep your writing surface at a comfortable height to
avoid straining your hands, wrists, or arms.  Your writing instrument should be held in a
relaxed grip, not too tightly.

4.  Start with proper formation:  Make sure you form each letter correctly from the
beginning strokes.  This means starting with the upward and downward strokes of the
letter at the right angle, and sizing them appropriately.

5.  Don't rush:  Take your time and write slowly and deliberately, focusing on the
movement of your hand and the pressure you apply to the paper.

6.  Keep it simple:  Avoid overly ornate or fancy writing, as this can be hard to read
and may slow you down.  Stick to basic, uniform strokes and practice until they become
second nature.

7.  Don't lift your pen:  Lifting your pen off the paper to change direction can cause
inconsistencies in your writing.  Practice making smooth, fluid transitions between
strokes without lifting your pen.

8.  Practice Cursive:  Cursive writing can help improve the flow and connectedness of
your letters, making your handwriting look neater.  Practice connecting the letters in
words and phrases.

9.  Pay attention to spacing:  Make sure there is enough space between words and letters
for legibility.

10.  Keep it clean:  A neat, uncluttered workspace can help you focus on your writing.
Keep your desk free of clutter and sharpen your pencils regularly to avoid smudges.

11.  Experiment:  Try different styles and techniques until you find one that feels
natural and looks good to you.  Don't be afraid to mix and match until you find what
works best for you.

12.  Don't give up:  Improving your handwriting takes time and patience, so don't
get discouraged if you don't see results immediately.  Keep practicing and you will
eventually see an improvement.

Remember, handwriting is a personal expression, so find a balance between what looks
nice and what comes naturally to you.  With practice, you can develop a neat and legible
handwriting that reflects your personality.
```

