# OpenReview forum: "Instruct-SkillMix: A Powerful Pipeline for LLM Instruction Tuning"
_ICLR.cc/2025/Conference — ICLR 2025 Poster_

### Official Review · Reviewer_jh2B · 2024-10-25

**Soundness:** 4
**Presentation:** 3
**Contribution:** 4
**Rating:** 6
**Confidence:** 4

**Summary:**

This paper focuses on constructing high-quality data for enhancing base LLMs’ instruction following capability. To achieve this goal, the authors propose a novel pipeline, called Instruct-SkillMix, which naturally combines diversity and difficulty in the data generation process. SFT with the resulting SFT dataset leads to very impressive performance gains on several established benchmarks. In particular, the data generation is fully automatic and the size of dataset can scale up easily.

**Strengths:**

- It finds that directly prompting a strong LLM to identify crucial skills achieves better performance than extracting skills from existing IFT datasets.
- The performance of using merely thousands of Instruct-SkillMix data is impressive.
- The data generation pipeline is fully automated and has nearly no human intervention.
- It conducts detailed ablation studies and shows the contributions of different components.
- It reveals that even a small amount of low-quality data greatly harms the instruction following performance.

**Weaknesses:**

- The type of queries and topics could be relevant to coverage of data. I think it might be worth to do ablation study on the query and topic types.

**Questions:**

Kindly refer to the weaknesses.

---

> ### Author Response · Authors · 2024-11-23
> **Response to Reviewer jh2B**
>
> > The type of queries and topics could be relevant to coverage of data. I think it might be worth to do ablation study on the query and topic types.
>
> We definitely agree with the reviewer. Part of the motivation behind the final version of the pipeline was that in Instruct-SkillMix-D, we noticed limited diversity in the types of Q&A produced (in particular, most Q&A pairs seemed to be of query type “help-seeking”). Thus, queries and topics (and extrapolated skills) would be important to coverage of data.
>
>
> That being said, while past open efforts invested significant human effort in ensuring high coverage of topics and scenarios to sufficiently equip the LLM for scenarios it might encounter at deployment time, we take a subtly different tack: accepting that pre-training is the dominant source of the LLM’s “inner knowledge,” we focus our instruction tuning on merely teaching the LLM skills to drawing upon that inner knowledge and present it nicely during conversations. Recent work by Zhao et al. (https://arxiv.org/abs/2409.19808) corroborates the effectiveness of our approach: in particular, they show that fine-tuning on k skills yields notable improvements when applied to both out-of-distribution combinations of k skills, and as well as more than k skills.

---

> ### Author Response · Authors · 2024-12-02
> **Response to Reviewer jh2B**
>
> We hope that most of your concerns have been addressed. If so, as the discussion period nears the end, we would appreciate it if you could reconsider your assessment. We’d be happy to engage in further discussions. We would be very happy to engage further in discussions to clarify any remaining concerns that you might havez

---

### Official Review · Reviewer_oHAp · 2024-11-03

**Soundness:** 3
**Presentation:** 3
**Contribution:** 2
**Rating:** 5
**Confidence:** 4

**Summary:**

The authors propose a new instruction tuning data generation pipeline, INSTRUCT-SkillsMIX. They prompt a strong LLM to identify some key instruction following skills. They then use these skills to produce useful instruction following data. They show strong results on instruction following benchmarks where an LLM is used as a judge.

**Strengths:**

- This paper shows very strong performance on benchmarks where LLMs are used as a judge.
- The InstructSkillMix framework is novel and interesting. Moreover, it does not require any seed data, which is beneficial.

**Weaknesses:**

- The baseline methods are not fair: the main comparison is to Alpaca 52K, which is really old and known to be a low quality dataset. I think the authors should try comparing their dataset to stronger datasets such as ShareGPT with the responses regenerated by GPT4-Turbo.
- In my opinion, section 1.1 is somewhat misleading. The authors (in line 70-75) say it is a mystery why public instruction tuning does not match the performance of proprietary instruct models. However, these proprietary models are trained in a variety of stages, and with distillation and RL techniques. It is not expected that instruction tuning alone can match the performance of proprietary models.
- Table 4 suggests that the main reason for the good performance of InstructSkillMix is that a stronger model is used for distillation compared to previous IFT datasets. With the same judge, Alpaca-1K longest performs similarly to InstructSkill Mix (although Alpaca 1K longest is a weak dataset in my opinion: instructions are created using text-davinci-003). Alpaca-1K longest does perform worse on the length-controlled benchmark, but this is not a fair comparison since Alpaca-1K longest is specifically biased to encourage longer responses.
- The authors claim that InstructSkillMix is a more performant data generation method than UltraChat and Vicuna (line 511). Although InstructSkillMix seems to be a strong method, my guess is that the primary reason that InstructSkillMix outperforms UltraChat and Vicuna is due to a stronger teacher model. If the authors want to make this claim, I think they should regenerate responses from UltraChat and ShareGPT using the same teacher model InstructSkillMix uses.
- In my opinion relying only on AlpacaEval 2.0 and MTBench is a bit limited. It would be beneficial for the authors to evaluate their models on other tasks, including mathematical reasoning (MATH, GSM8K), instruction following (IFEval), and knowledge benchmarks (MMLU).

**Questions:**

- Can you compare performance to ShareGPT with the responses regenerate with GPT4-Turbo?
- Can the authors discuss weakness 3?

---

> ### Author Response · Authors · 2024-11-23
> **Response to Reviewer oHAp**
>
> ## Response to Weaknesses
> > The baseline methods are not fair: the main comparison is to Alpaca 52K, which is really old and known to be a low quality dataset. I think the authors should try comparing their dataset to stronger datasets such as ShareGPT with the responses regenerated by GPT4-Turbo.
>
> Please see the following portion in our meta-comment: **Gains from pipeline vs. strong teacher model, Part 1: Rewriting responses to ShareGPT and UltraChat**.
>
> > In my opinion, section 1.1 is somewhat misleading. The authors (in line 70-75) say it is a mystery why public instruction tuning does not match the performance of proprietary instruct models. However, these proprietary models are trained in a variety of stages, and with distillation and RL techniques. It is not expected that instruction tuning alone can match the performance of proprietary models.
>
> Our paper *does* show that SFT on a tiny dataset can lead to better performance (on some evaluations)  than well-regarded frontier models. However, the reviewer is absolutely correct that the phrasing is inaccurate. Llama-3-8b-Instruct was developed with a combination of distillation and RL techniques. However, Mistral-7B-Instruct-v0.2 was developed **only with SFT** with datasets openly available on Huggingface. A correct statement would be: Unlike prior open efforts to SFT Mistral-7B-Base-v0.2, our pipeline is the only method (to our knowledge) that can result in high performance via SFT Mistral-7B-Base-v0.2 and even surpass Mistral-7B-Instruct-v0.2 on some common benchmarks.
> We will rewrite this more clearly in the final version of the paper.
>
> > Table 4 suggests that the main reason for the good performance of Instruct-SkillMix is that a stronger model is used for distillation compared to previous IFT datasets. With the same judge, Alpaca-1K longest performs similarly to Instruct-SkillMix (although Alpaca 1K longest is a weak dataset in my opinion: instructions are created using text-davinci-003). Alpaca-1K longest does perform worse on the length-controlled benchmark, but this is not a fair comparison since Alpaca-1K longest is specifically biased to encourage longer responses.
>
> We refer the reviewer to the following section in the meta-comment for additional benchmarks: **Gains from pipeline vs. strong teacher model, Part 1: Rewriting responses to ShareGPT and UltraChat**.
>
> > The authors claim that InstructSkillMix is a more performant data generation method than UltraChat and Vicuna (line 511). Although InstructSkillMix seems to be a strong method, my guess is that the primary reason that InstructSkillMix outperforms UltraChat and Vicuna is due to a stronger teacher model. If the authors want to make this claim, I think they should regenerate responses from UltraChat and ShareGPT using the same teacher model InstructSkillMix uses.
>
> We refer the reviewer to the following sections in the meta-comment:
> 1) **Gains from pipeline vs. strong teacher model, Part 1: Rewriting responses to ShareGPT and UltraChat**
> 2) **Gains from pipeline vs. strong teacher model, Part 2: Generalizability of pipeline with different teacher model**
>
> > In my opinion relying only on AlpacaEval 2.0 and MTBench is a bit limited. It would be beneficial for the authors to evaluate their models on other tasks, including mathematical reasoning (MATH, GSM8K), instruction following (IFEval), and knowledge benchmarks (MMLU).
>
> We kindly refer the reviewer to Table 8, where we report performance on GSM8K, MMLU (and other benchmarks). On these benchmarks, we do not see a drop in performance.
>
> We would like to highlight that we reported these results in the appendix (instead of the main paper) because we wanted to evaluate our method on general instruction-following abilities. IFEval primarily tests formatting related skills (number of lines, paras etc.), and our current pipeline does not teach those skills either.

---

> ### Author Response · Authors · 2024-11-23
> **Response to Reviewer oHAp Continued**
>
> ## Response to Questions
> > Can you compare performance to ShareGPT with the responses regenerate with GPT4-Turbo?
>
> We refer the reviewer to the following section in the meta-comment: **Gains from pipeline vs. strong teacher model, Part 1: Rewriting responses to ShareGPT and UltraChat**
>
> We set Alpaca 52K as our baseline because it is the most commonly studied dataset in previous literature. Also, there are some technical difficulties in comparing against ShareGPT:
> 1) The quality of the dataset is uneven and it is known that a lot of effort is needed to filter the dataset. The Vicuna team does not disclose the exact filtering rule they used for the final version of their dataset, nor did they release the dataset itself.
> 2) The dataset is multi-turn with some of the questions dependent on the model’s response in the previous turn. Regenerating the response in the first turn might make the user’s prompt in the subsequent turns obsolete, which may introduce confusion to the student model.
>
> We attempt to compare against ShareGPT with the following approach. We use the filtered version available at: https://huggingface.co/datasets/anon8231489123/ShareGPT_Vicuna_unfiltered/tree/main and randomly select 1000 examples and only take the first (user, assistant) interaction. From a preliminary experiment, SFT on just the first turn performs just as well as SFT on all turns. We then regenerate the assistant response with GPT-4-Turbo.
>
> Our pipeline still significantly outperforms regenerating responses to prompts from ShareGPT.
>
>
> > Can the authors discuss weakness 3?
>
> The reviewer brings up a valid concern. However, we would first like to point out that the numbers reported in Table 4 were from Instruct-SkillMix-D. When we include the results from Instruct-SkillMix (our main pipeline), even the WR is significantly higher for our pipeline.
>
> Nonetheless, to address the concern about length-controlled being an unfair metric for models trained on Alpaca 1K Longest, we regenerate responses to randomly selected 1000 examples from the Alpaca 52K dataset.
>
> We refer the reviewer to the following section in the meta-comment: **Gains from pipeline vs. strong teacher model, Part 1: Rewriting responses to ShareGPT and UltraChat**
>
> Although Alpaca 1K Random performs better on the length-controlled metric than Alpaca 1K Longest, it falls significantly under our pipeline. This shows the strong performance of the models is not entirely due to a strong teacher model, but can be attributed to our novel pipeline.

---

> ### Author Response · Authors · 2024-12-02
> **Response to Reviewer oHAp**
>
> We hope that most of your concerns have been addressed. If so, as the discussion period nears the end, we would appreciate it if you could reconsider your assessment. We’d be happy to engage in further discussions. We would be very happy to engage further in discussions to clarify any remaining concerns that you might havez

---

> > ### Comment · Reviewer_oHAp · 2024-12-02
> > **Response to Rebuttal**
> >
> > Thank you for the additional experiments. They make the methodological contribution seem more sound. However, I still believe that the claim of "We surpass all prior open SFT-only instruction-tuning efforts by a wide margin" cannot really be claimed as a contribution of this paper. The good results shown in this paper are likely due to the increase in performance of closed source models, rather than methodological improvements. This is highlighted by the fact that the performance when using Claude 3.5 Sonnet as a teacher model is much lower than the performance when GPT-4-Turbo is used as a teacher.

---

> > > ### Author Response · Authors · 2024-12-02
> > >
> > > We thank the reviewer for their response, and address their comments below:
> > >
> > > > Thank you for the additional experiments. They make the methodological contribution seem more sound. However, I still believe that the claim of "We surpass all prior open SFT-only instruction-tuning efforts by a wide margin" cannot really be claimed as a contribution of this paper. The good results shown in this paper are likely due to the increase in performance of closed source models, rather than methodological improvements.
> > >
> > > While we agree that perhaps “all” is too strong, we believe that the increase in performance _does_ come from our pipeline rather than solely from improvements in the closed source models. We highlight the following experiments that when using the Instruct-SkillMix pipeline vs. other pipelines with the same teacher model, **our methodology always outperforms all baselines we consider** (including the ones that the reviewer recommends).
> > >
> > > In Table 4, we had presented the ablations where we generate Instruct-SkillMix-D (our weaker pipeline) with a weaker version of GPT-4 to match the teacher model from Alpaca-1K-Longest, and we still saw that our methodology was superior.
> > >
> > > We would additionally like to present the following result, where we follow the Alpaca pipeline to generate 1000 (instruction, response) pairs (not just regenerating response to existing instructions).
> > >
> > > | Generation Model  | Dataset                                                | WR(%) | LC WR(%) | MT-Bench |
> > > |-------------------|------------------------------------------------------------------|---------|---------|--------|
> > > | GPT-4-Turbo | Alpaca 1K (from scratch)                                       |  6.18 | 12.53 | 6.43 |
> > >
> > >
> > > Note that even though we use GPT-4-Turbo, a strong teacher model, the downstream performance is abysmal. Similarly, we have provided multiple baseline methodologies which **do not benefit equally from the use of a stronger teacher model**. With the suggestion of the reviewer, we even added two more baseline methodologies (regenerating responses from ShareGPT and Ultrachat). Even though the reviewer claims that they should be a stronger dataset than Alpaca, we observe no significant difference in the downstream performance than regenerating responses to Alpaca-1K-Random.
> > >
> > > We believe that we have adequately shown that our methodology consistently outperforms all other methodologies known to us. If the reviewer would like to conclude that the good results shown in this paper are due to the increase in performance of closed source models and not methodological improvements, we would like to invite the reviewer to suggest other methodologies to compare against, which benefit equally or even more from the use of a strong teacher model than ours.
> > >
> > >
> > > > This is highlighted by the fact that the performance when using Claude 3.5 Sonnet as a teacher model is much lower than the performance when GPT-4-Turbo is used as a teacher.
> > >
> > > We respectfully disagree. The results only show that Claude 3.5 Sonnet is a worse teacher model than GPT-4-Turbo.
> > >
> > > We refer the reviewer to the following part of our meta-comment to all reviewers: **Gains from pipeline vs. strong teacher model, Part 2: Generalizability of pipeline to other teacher models**.
> > > Here, one can see that when fixing the teacher model as Claude 3.5 Sonnet, we also observe an improvement from using our methodology, compared to other methodologies (e.g., Alpaca and ShareGPT).

---

### Official Review · Reviewer_MpdQ · 2024-11-04

**Soundness:** 2
**Presentation:** 3
**Contribution:** 2
**Rating:** 3
**Confidence:** 4

**Summary:**

The paper introduces INSTRUCT-SKILLMIX, a pipeline for creating instruction-tuning datasets using large language models. The method involves two stages: (1) extracting instruction-following skills using an LLM's metacognitive abilities, and (2) generating synthetic (instruction, response) pairs using random combinations of these skills. Using just 4K examples, the authors demonstrate that base models fine-tuned on this data achieve competitive performance on instruction-following benchmarks compared to much larger models.

**Strengths:**

The paper presents a novel approach to synthetic data generation that achieves strong results with only 4K examples, suggesting an efficient path forward for instruction tuning. The empirical validation is well-designed, testing across multiple benchmarks and models while including careful ablation studies that isolate the effects of different components.

The method is cost-effective, requiring only about $600 compared to traditional human annotation approaches.

The authors provide some analysis of how low-quality data affects model performance, offering practical insights for dataset creation.

The paper also test both their preferred method and a seed-dataset dependent variant, providing comparative insights.

**Weaknesses:**

The paper's most significant limitation is the performance plateau at 4K examples, with no clear explanation or analysis of learning curves as dataset size increases. This is compounded by limited investigation of whether different architectures or model sizes might hit different ceilings.

The evaluation methodology relies heavily on AlpacaEval 2.0 and lacks assessment of long-form generation and multi-turn conversations. The use of both teacher and grader models from the same model family (GPT-4) raises concerns about potential systematic biases.

Also, the methodology lacks a principled approach for determining the optimal number of skills or combinations, and provides no systematic quality metrics for the generated data.

The paper provides limited investigation of how different teacher models might affect results. Lack of this raises questions about the method's generalizability.

The relationship between skills and model performance remains inadequately explored, with no clear metrics for assessing skill quality or coverage.

**Questions:**

Have you investigated whether different model architectures or sizes hit the 4K example ceiling at different points?

Could you explain the choice of k=2 for skill combinations? Have you explored other values?

How would the method perform with different teacher models (e.g., Claude, PaLM)?

Would it be possible that combining synthetic data with human annotations potentially break through the 4K example ceiling?

Could you elaborate on potential approaches for quality control in the data generation process?

Could you provide analysis of model performance on longer-form tasks and multi-turn conversations?

---

> ### Author Response · Authors · 2024-11-23
> **Response to Reviewer MpdQ**
>
> ## Response to Weaknesses
>
> > The paper's most significant limitation is the performance plateau at 4K examples, with no clear explanation or analysis of learning curves as dataset size increases. This is compounded by limited investigation of whether different architectures or model sizes might hit different ceilings.
>
> All methods plateau at some point, and no prior method had hit the performance level of our method despite using far more data (including efforts with humans in the loop such as Dolly and ShareGPT). Ultimately, an SFT cannot improve on the teacher, and in our experiments stronger models like Llama3 8B plateau pretty much when they achieve 50% win-rate against the teacher on held-out queries. The high performance on evals (e.g., on AlpacaEval 2.0) points to good OOD generalization. A weaker model or a weaker pipeline generalizes less well and even on training may not match the teacher’s performance on training data.
>
> For detailed learning curves on different dataset sizes (and different architectures), we refer the reviewer to the following parts of our initial submission: Table 7 (Appendix B) and Figures 3 and 4 (Appendix E.2). As shown in Figure 4, the trend appears to depend significantly on the model, with different architectures and sizes reaching different performance ceilings. Since our original submission, we have also trained Gemma2 2B on Instruct-SkillMix datasets of varying sizes (see table below). Taking these results into account along with Table 7, we still see that different architectures and sizes hit different plateaus. This suggests that the observed plateau is influenced by both dataset size and the specific model architecture and size.
>
> | Model  | Dataset              | WR(%) | LC WR(%) | MT-Bench |
> |---------------|---------------------------|---------|---------|--------|
> | Gemma2 2B | Instruct-SkillMix-1K | 11.89 | 11.17 | 5.98 |
> | Gemma2 2B | Instruct-SkillMix-2K | 10.92 | 10.62 | 6.04 |
> | Gemma2 2B | Instruct-SkillMix-4K | 11.37 | 11.11 | 6.18 |
>
> The strength of our paper is our simple and powerful pipeline. We hope it will cause a rethink of current approaches and motivate other efforts to improve the method; e.g. (a) better ways to extract skills (b) curating Q&A according to types of questions (e.g., essay questions, fixed-format questions as in IF-EVAL, or on a ranking of skills based upon how they contribute to downstream performance.
>
> > The evaluation methodology relies heavily on AlpacaEval 2.0 and lacks assessment of long-form generation and multi-turn conversations. The use of both teacher and grader models from the same model family (GPT-4) raises concerns about potential systematic biases.
>
>
> We thank the reviewer for bringing up a good point. However, we would like to highlight that we chose
> AlpacaEval 2.0 as an evaluation for instruction following (not necessarily long-form generation / multiturn conversations). Please also see the response to the last question (i.e. one could easily use the Instruct-SkillMix pipeline and specify answers of different forms and word-length).
>
> Regarding systematic biases due to teacher and grader models this was addressed in the paragraph **Effect of choice of grader** on page 8. Using Claude 3 Opus as a grader conserves the relative ranking between models, but favors our models even more than does GPT-4.
>
> We also refer reviewer to the meta-comment (**Gains from pipeline vs. strong teacher model, Part 2: Generalizability of pipeline with different teacher model**) where we use Claude-Sonnet-3.5 for the Instruct-SkillMix Pipeline, GPT-4 as the grader, and still see performance gains compared to baselines.
>
> > Also, the methodology lacks a principled approach for determining the optimal number of skills or combinations, and provides no systematic quality metrics for the generated data.
>
> Yes, the optimal set of skills as well as number of skills is left for future work. One issue is that we also lack good evaluations, and so our focus was on showing that our new ideas lead to strong OOD generalization, i.e. performance on current evaluations. The paper explores the performance of SFT on combinations of k skills for k=1 and k=2. Here, k=2 is not significantly better than k=1, but this is possibly due to limitations of the evaluations.  In a somewhat different setting (i.e. with different task and different list of skills), recent work by Zhao et al. (https://arxiv.org/abs/2409.19808) shows that fine-tuning on combinations of k skills yields notable improvements when applied to both out-of-distribution combinations of k skills, and as well as more than k skills. Potentially those ideas can yield better evaluations than AlpacaEval 2.0.

---

> ### Author Response · Authors · 2024-11-23
> **Response to Reviewer MpdQ Continued**
>
> > The paper provides limited investigation of how different teacher models might affect results. Lack of this raises questions about the method's generalizability.
>
> We refer the reviewer to the following sections in the meta-comment:
> 1) **Gains from pipeline vs. strong teacher model, Part 1: Rewriting responses to ShareGPT and UltraChat**
> 2) **Gains from pipeline vs. strong teacher model, Part 2: Generalizability of pipeline with different teacher model**
>
> > The relationship between skills and model performance remains inadequately explored, with no clear metrics for assessing skill quality or coverage.
>
> The question refers to “Quality and coverage,” which in prior approaches to instruction tuning (e.g., UltraChat) was decided by humans.  While there is no doubt that the set of skills matters, we present evidence that it may be simplistic to use humans to judge this.  We reported on two methods to come up with a list of skills and find that performance differs a lot, and especially with respect to brittleness to presence of “bad” data. (Instruct-SkillMix-D vs Instruct-SkillMix in Table 7, Appendix B).  This is mystifying. The experiment suggested it is best to use skills derived directly using the metacognition of the teacher model — rather than from making the model extract skills from pre-existing seed datasets.
>
> But interestingly, the performance of skills depends on *what other model* is used for generation. The best performance is achieved when the skills are sourced from the same model as the one generating the data (see table below). In other words, there may be no optimum set of skills, and the set may change with the teacher and even the student. Our paper opts to highlight such phenomena, which is an important contribution for design of instruction-tuning datasets.
>
> We will add the above discussion and thank the reviewer for bringing this up.
>
> | Generation Model  | Dataset                                                | WR(%) | LC WR(%) | MT-Bench |
> |-------------------|------------------------------------------------------------------|---------|---------|--------|
> | GPT-4-Turbo | Instruct-SkillMix-1K (skills from Claude)                | **43.22** | 31.98 | 7.20  |
> | GPT-4-Turbo | Instruct-SkillMix-1K                                                | 41.97 | **38.48** | **7.33**  |
> | Claude 3.5 Sonnet | Instruct-SkillMix-1K (skills from GPT)           | 21.32 | 23.91 | 6.87  |
> | Claude 3.5 Sonnet | Instruct-SkillMix-1K                                       | **25.74** | **25.54** | **6.88**  |

---

> ### Author Response · Authors · 2024-11-23
> **Response to Reviewer MpdQ Continued**
>
> ## Response to Questions
>
> > Have you investigated whether different model architectures or sizes hit the 4K example ceiling at different points?
>
> Please see our response to the first weakness.
>
> > Could you explain the choice of k=2 for skill combinations? Have you explored other values?
>
> As shown in Table 3, we explore k=1 and k=2. We did not explore larger values of k since we did not observe significant enough gains from k=1 to k=2.
>
>
> > How would the method perform with different teacher models (e.g., Claude, PaLM)?
>
> Please see the following portion in the meta-comment: **Gains from pipeline vs. strong teacher model, Part 2: Generalizability of pipeline with different teacher model**.
>
> > Would it be possible that combining synthetic data with human annotations potentially break through the 4K example ceiling?
>
> We would like to emphasize that the ceiling achieved using just 4K examples is already quite high (i.e., it has not been achieved by other SFT-only efforts. The performance surpasses SFT on ShareGPT (where responses are written by humans). Please see the performance of Vicuna-13B v1.5 in Table 7 (Appendix B). While it may be possible that human annotations could help, one of the main contributions of this paper was to minimize the need for human annotation / intervention so as to reduce the cost required for generating the dataset.
>
>
> > Could you elaborate on potential approaches for quality control in the data generation process?
>
> One of the main contributions of this paper was to minimize the need for human annotation / intervention so as to reduce the cost required for generating the dataset. As hinted above, we suspect that most human interventions may reduce performance.
>
>
> Please see Table 13 (Appendix G) and the following table (which reports the numbers for Instruct-SkillMix instead of Instruct-SkillMix-D). When we train on 4 different subsets of the same size from our dataset, the resulting models have similar performance across the benchmarks.
>
> However, Table 6 (also included below) shows the negative effect on the trained models when we intentionally introduce 20% of data that has uneven quality (shorter in length or poor quality). The difference between the two settings helps us deduce that the data generated from our pipeline has a uniform quality.
>
> | Model       | Dataset                        | WR(%) | LC WR(%) |
> |---------------|------------------------------------|---------|---------|
> | Mistral 7B | Instruct-SkillMix-1K-Split1 | 41.97 | 38.48 |
> | Mistral 7B | Instruct-SkillMix-1K-Split2 | 41.71 | 35.44 |
> | Mistral 7B | Instruct-SkillMix-1K-Split3 | 40.78 | 36.40 |
> | Mistral 7B | Instruct-SkillMix-1K-Split4 | 42.52 | 36.64 |
>
> | Model       | Dataset                        | WR(%) | LC WR(%) |
> |---------------|------------------------------------|---------|---------|
> | Mistral 7B | Instruct-SkillMix-2K           | 40.83 | 36.18 |
> | Mistral 7B | Instruct-SkillMix-2K-Brev  | 35.35 | 31.61 |
> | Mistral 7B | Instruct-SkillMix-2K-Junk  | 30.10 | 24.60 |
>
> > Could you provide analysis of model performance on longer-form tasks and multi-turn conversations?
>
> While we appreciate the reviewer bringing up this point, this is not in scope of the project, and all generated data was a single (instruction, response) pair. However, our pipeline is adaptable to other settings, and it is easy to inject human inductive bias if needed. By changing the prompt to specifically ask for skills and data for multi-turn conversations, we can generate data that are relevant to multi-turn conversations. See an example of a generated list of skills / and an example data point.
>
> ### EXAMPLE SKILLS
> itinerary_customization, problem_solving
>
> ### EXAMPLE DATA
> ###Human: I need to plan a trip to New York for a conference but I also want to squeeze in some sightseeing. Can you help me with that?\n### AI: Absolutely! When will you be going, and how many days will you be staying in New York?\n\n### Human: I'm attending the conference from April 10th to 12th, but I can stay until the 15th.\n### AI: Great! Would you prefer to do your sightseeing before or after the conference days? [rest omitted]

---

> ### Author Response · Authors · 2024-12-02
> **Response to Reviewer MpdQ**
>
> We hope that most of your concerns have been addressed. If so, as the discussion period nears the end, we would appreciate it if you could reconsider your assessment. We’d be happy to engage in further discussions. We would be very happy to engage further in discussions to clarify any remaining concerns that you might have

---

### Author Response · Authors · 2024-11-23
**General Response to All Reviewers**

We thank the reviewers for their thoughtful reviews on our work, and for recognizing the novelty of our pipeline, as well as the strong results achieved via SFT on the resulting dataset.

We first address issues raised by more than one reviewer, and mention updates to the results since submission.

## **Primary Contribution of Our Work**

We would like to emphasize the following points as the main contribution of our work
1. We surpass all prior open SFT-only instruction-tuning efforts by a *wide* margin.
- We enable small open models to achieve competitive performance to large proprietary models only via SFT. No RL.
2. Our method is extremely cost effective (estimated data cost $600)
- We avoid human intervention / annotation. The automated pipeline seems usable for other purposes.
- We use orders of magnitude smaller datasets (our 1K dataset beats older methods that used 50k to 1M examples)


## **Gains from pipeline vs. strong teacher model, Part 1: Rewriting responses to ShareGPT and UltraChat**

To disentangle the advantage of Instruct-SkillMix from the potential benefit of using a stronger teacher model, we initially compared the results of SFT on two datasets (generated by the same model): Alpaca-1K Longest and Instruct-SkillMix-D-1K (Table 4).

One reviewer found this insufficient and desired a similar comparison with UltraChat and ShareGPT. We regenerate responses to random 1K examples from UltraChat and ShareGPT using the same teacher model as used in Instruct-SkillMix. We present results in the table below, showing SFT on Instruct-SkillMix is superior to them, especially in the most important metric of LC win rate. We think the LC win rate is due to the high density of information (per word) elicited by our SkillMix idea for generating diverse and interesting queries.

| Generation Model  | Dataset                                                | WR(%) | LC WR(%) | MT-Bench |
|-------------------|------------------------------------------------------------------|---------|---------|--------|
| GPT-4-Turbo | Alpaca 1K Longest (response regenerated)          | 35.23 | 19.62 | 6.99 |
| GPT-4-Turbo | Alpaca 1K Random (response regenerated)         | 20.85 | 23.48  | 6.93 |
| GPT-4-Turbo | Ultrachat 1K (response regenerated)                    | 37.10 | 25.64 | **7.39** |
| GPT-4-Turbo | ShareGPT 1K (response regenerated)                  | 30.06 | 26.01 | 7.19 |
| GPT-4-Turbo | Instruct-SkillMix-D-1K                                            | 33.87 | 27.48 | 6.92  |
| GPT-4-Turbo | Instruct-SkillMix-1K                                                | **41.97** | **38.48** | 7.33  |


## **Gains from pipeline vs. strong teacher model, Part 2: Generalizability of pipeline to other teacher models**

Our pipeline indeed primarily utilizes models from the GPT-4 family. Below we report experiments with the same pipeline but using Claude-Sonnet-3.5 in all stages–- curating skills and query types, followed by Q&A generation. We report performance on AlpacaEval 2.0 (with GPT-4 as the grader) in the table below:


| Generation Model  | Dataset                                                | WR(%) | LC WR(%) | MT-Bench |
|-------------------|------------------------------------------------------------------|---------|---------|--------|
| Claude 3.5 Sonnet | Alpaca 1K Longest (response regenerated) | 22.10 | 19.12 | **7.13**  |
| Claude 3.5 Sonnet | ShareGPT 1K (response regenerated)         | 21.00 | 19.77 | 7.06  |
| Claude 3.5 Sonnet | Instruct-SkillMix-1K                                       | **25.74** | **25.54** | 6.88  |

Observe that for a fixed generator model, SFT on Instruct-SkillMix outperforms Alpaca-1K Longest and ShareGPT-1K. This affirms that the benefits observed via SFT on Instruct-SkillMix arise from our pipeline.

---

### Meta-Review · Area_Chair_uTmj · 2024-12-21

**Metareview:**

This paper introduces a method for creating synthetic instruction fine-tuning data using a teacher model. The method firstly instructs a strong teacher LLM to generate a list of skills useful for instruction following, and then ask the LLM to generate instructions/queries which require 1 or 2 randomly selected skills, after which the teacher LLM is again used to generate the response. The method achieved impressive instruction-following performance using merely 1k-4k data points. Various ablation studies are performed to verify that the proposed method for dataset construction (while controlling other factors such as the teacher model and the grader model) can indeed lead to significant performance improvement.

The reviewers generally agreed that the paper has a number of strengths: the proposed method is novel, the experiments and ablation studies are well designed, the method is cost-effective, the paper additionally offers insights as to how the addition of low-quality data degrades the performance, the pipeline is fully automated not requiring any human intervention.

In the initial review, the reviewers pointed out some concerns, mostly regarding the fairness of comparisons and whether the reported performance advantage indeed stems from the novel methodology.  After reading the paper and the discussions, I think the authors' rebuttal sufficiently addressed these concerns. Below I list some major concerns:
- The performance plateau at 4K examples (Reviewer MpdQ): The authors responded that all methods plateau eventually since its performance is upper-bounded by the teacher model, and different models can plateau at different points. Also, this plateau at 4K examples is already quite high, and the performance achieved by the proposed method is already impressive in various experiments (surpassing the baselines and the instruction-tuned models).
- Potential systematic bias since the teacher and grader are from the same family (Reviewer MpdQ): The authors pointed to experiments in which they used a different grader model and also added a new experiment during rebuttal. The proposed method still consistently performs better than the baseline methods in these experiments. I think this is sufficient to justify that the performance advantage of the proposed method is not significantly affected by this systematic bias.
- How the teacher model affects the performance (Reviewer MpdQ): The added results during rebuttal sufficiently demonstrated that the proposed method performs better than the baselines across different teacher models.
- Should compare with other synthetic data method (ShareGPT and Ultrachat) with regenerated responses using GPT-4 (Reviewer oHAp): The authors added the corresponding results requested by the reviewer, and the performance advantage of the proposed method is consistent in these new experiments.
- The performance improvement of the proposed pipeline is due to the use of stronger teacher model (Reviewer oHAp): The authors' response to this point also makes sense to me. The newly added experiments showed that even if the same strong teacher model is used to regenerate the responses for other synthetic data method, the proposed method still performs better.
- Lack of evaluation for long-form generation and multi-turn conversations (Reviewer MpdQ): I think since the proposed method for instruction dataset construction is novel, evaluating it in standard single-turn conversation benchmarks seems good enough to me (as long as fair comparison is ensured).

I also agree with the follow-up comment from Reviewer oHAp that the reported numbers in this paper are partially due to the improved capability of the GPT-4 teacher model. However, the authors have used the new experiments to demonstrate that even if the same teacher model is used, the proposed method still works better. I suggest the authors revise and soften some of the potential overclaims pointed out by Reviewer oHAp (such as "by a wide margin").

Overall, I think the paper makes a useful and important contribution to instruction fine-tuning based on teacher models, and the method could inspire future works in this area as well. Therefore, acceptance is recommended.

**Additional Comments On Reviewer Discussion:**

During rebuttal, only Reviewer oHAp responded with a follow-up comment. After reading the reviews and rebuttal, I think the authors have sufficiently addressed the concerns raised by the reviewers in their initial reviews.

---

### Decision · Program_Chairs · 2025-01-22

Accept (Poster)